# The implications of alternative splicing regulation for maximum lifespan

Wei Jiang[1], Sika Zheng [2,3] ✉ & Liang Chen [1] ✉

Mammalian maximum lifespan (MLS) varies over a hundred-fold, yet the molecular mechanisms underlying this diversity remain unclear. We present a cross-species analysis of alternative splicing (AS) across six tissues in 26 mammals, identifying hundreds of conserved AS events significantly associated with MLS, with the brain containing twice as many tissue-specific events as peripheral tissues. MLS-AS events are enriched in pathways related to mRNA processing, stress response, neuronal functions, and epigenetic regulation, and are largely distinct from genes whose expression correlates with MLS, indicating that AS captures unique lifespan-related signals. The brain exhibits certain associations divergent from peripheral tissues and reduced overlap with body mass (BM)-associated splicing; neither is observed at the gene expression level. While MLS- and age-associated AS events show limited overlap, the shared events are enriched in intrinsically disordered protein regions, suggesting a role in protein flexibility and stress adaptability. Furthermore, MLS-associated AS events display stronger RNA-binding protein (RBP) motif coordination than age-associated ones, highlighting a more genetically programmed adaptation for lifespan determination, in contrast to the more variable splicing changes seen with chronological aging. These findings suggest alternative splicing as a distinct, transcription-independent axis of lifespan regulation, offering insights into the molecular basis of longevity.

Aging is a dynamic, inevitable, and irreversible process characterized by the gradual deterioration of multiple body systems, leading to increased mortality risk[1]. Advances in medical science have substantially extended human life expectancy, which in turn has led to higher incidences of chronic age-associated diseases[2]. Currently, aging is considered the most important risk factor for most chronic diseases, including cardiovascular disease, atherosclerosis, cancer, and neurodegenerative diseases[3–5].

The maximum lifespan (MLS) is the upper limit of lifespan a species can achieve. Because human MLS is significantly limited by aging and chronic aging-associated disease[6], understanding the factors that regulate MLS is crucial. MLS studies will shed light on the

fundamental biology of aging and provide valuable insights into extending healthy human lifespans.

Comparative studies examining lifespan variability across species may provide a panoramic view on the biological underpinnings of lifespan limit, longevity, and aging[7,8]. Recent cross-species transcriptomic analyses by Lu et al. focused on gene-level expressions and identified thousands of genes whose expression levels correlate with species-specific MLS[9]. They further found that genes with expression negatively correlated with MLS are enriched in energy metabolism and inflammation pathways and are controlled by circadian regulatory factors, whereas positively correlated genes are enriched in DNA repair and are controlled by the pluripotency

[1]Department of Quantitative and Computational Biology, University of Southern California, Los Angeles, CA, USA. [2]Division of Biomedical Sciences, School of Medicine, University of California, Riverside, CA, USA. [3]Center for RNA Biology and Medicine, University of California, Riverside, CA, USA. ✉e-mail: sikaz@ucr.edu; liang.chen@usc.edu

network. These findings highlight distinct transcriptional programs associated with MLS regulation.

The dynamics of transcriptomic landscapes tied to the aging process are attributed to both transcriptional and post-transcriptional regulations[10]. Alternative splicing (AS) is a post-transcriptional or co-transcriptional regulatory mechanism by which a single gene generates multiple distinct mature transcript isoforms, leading to protein diversity in higher eukaryotes[11]. Up to 95% of multi-exon human genes undergo AS, often exhibiting tissue- or cell-type dependent regulation and dynamically controlled in distinct cellular processes[12]. Around 15% of human hereditary diseases and cancers are associated with AS anomalies[13,14]. The association between AS and the intertwined realms of aging and longevity remains unclear[15]. Age-associated splicing changes have been reported in different experimental systems, albeit in a piecemeal fashion, and alterations in certain splicing factor expressions in the spleen have been linked to differences in lifespan (ranging from 2.1 to 3.3 years, or about 1.5-fold differences) among six mouse strains[15,16]. These earlier findings hint that splicing regulation might impact lifespan, but a comprehensive comparative analysis of maximum lifespan as a species trait across species with widely varying maximum lifespans has not yet been conducted.

In this study, we systematically investigated MLS-associated AS events across multiple mammalian species spanning a broad range of maximum lifespans to uncover potential links between splicing and lifespan regulation. We first analyzed and integrated transcriptomic datasets of six tissues in 26 non-human mammals with a maximum lifespan ranging from 2.2 to 37 years (>16-fold differences) to identify conserved MLS-associated AS events. Additionally, we determine age-associated AS events in humans using data from the Genotype-Tissue Expression (GTEx) project[17]. By harnessing the power of comparative transcriptomics and comparing associated splicing across species and within humans, our study sheds light on the relationship between AS and the complex biology of longevity, enriching our understanding of the multifaceted aging process.

## Results

### The landscape of homologous alternative splicing events across 26 non-human mammalian species

To identify AS events associated with MLS using a comparative transcriptomics approach, it is essential to gather splicing information for homologous AS events from a wide range of species with substantial variation in maximum lifespan. Since alternative exons in general are less conserved than constitutive exons[18], our study focused on species within the related orders to enhance the precision of AS identification. The chosen RNA-seq transcriptome dataset encompasses six tissues from 26 species, whose lifespans range from 2.2 to 37 years. These species belong to the Rodentia and Eulipotyphla orders, which diverged approximately 90 million years ago[19].

To acquire homologous AS events across the 26 species, we employed an integrated pipeline involving transcriptome assembly and sequence comparison (Supplementary Fig. 1). Homologous AS events were discerned through sequence alignments with mouse sequences because the mouse genome annotation is the most complete (see Supplementary Fig. 2, details in Methods). Figure 1a shows that the number of homologous events correlated with the evolutionary distance from mice, ranging from 900 to over 3700 events. The Spearman correlation between the event count and evolutionary distances, calculated across 100 phylogenetic trees generated by PhyloT, yielded a median value of −0.51.

In total, we identified 1974 homologous AS events (distributed among 1267 genes) conserved in at least ten species, which we defined as conserved AS events. While all seven alternative splicing patterns were found across species, cassette exon (skipped exon) emerged as the most prevalent type (33.4%) and showed the strongest enrichment (Fig. 1b, c). Notably, cassette exon accounted for 80.8% of all

conserved events (Fig. 1c, top panel) and exhibited the highest rate of being conserved across different species (17.2% of all cassette exons, Fig. 1c, bottom panel). In contrast, although alternative first exon (AF) was the second most frequent AS event, averaging 22.8% in a single species, comparable to cassette exon (Fig. 1b), none of these were conserved (Fig. 1c). Similarly, while alternative last exon (average of 6.1%) was common within individual species, none met our conservation criteria (Fig.1b, c, and Supplementary Fig. 3). Mutually exclusive exons were the least common AS type overall (Fig. 1b), being the least frequent in 24 out of the 26 species (Supplementary Fig. 3a). Interestingly, despite their scarcity, mutually exclusive exon events were efficiently conserved; while they represented only 1.7% of the total conserved events (Fig. 1c, top panel), their intrinsic conservation rate of 4.66% was the second highest among all AS types (Fig. 1c, bottom panel).

### Alternative splicing events associated with maximum lifespan in non-human mammalian species

To explore the role of AS in MLS, we calculated the Spearman correlation between the PSI (percent spliced-in) values of an alternative exon in each tissue and the maximum lifespans of the species studied. Approximately 220–270 AS events per tissue were significantly associated with MLS (absolute Spearman correlation >0.4 and FDR < 0.05), with roughly half showing positive correlations (higher exon inclusion in long-lived species, Fig. 2a) and half negative (higher inclusion in shorter-lived species, Fig. 2b).

In total, 731 out of 1974 conserved AS events (37%) showed an MLS association in at least one of the six tissues, indicating that a substantial fraction of splicing events may be linked to lifespan differences. For comparison, only about 20% of genes showed expression levels correlated with MLS[9], suggesting that alternative splicing captures additional lifespan-related signals beyond gene expression alone. The majority of these AS events were cassette exon, and the distribution of AS types closely mirrored that of all homologous AS events considered, indicating that no specific AS type is disproportionately enriched among the MLS-associated events (Supplementary Fig. 4).

Because closely related species might have similar lifespans and splicing patterns due to common ancestry, we conducted an analysis of phylogenetically independent contrasts (PICs) on the original PSI values to ensure correlations were not driven by phylogenetic relationships. The evolutionary relatedness among species can introduce statistical dependencies, potentially leading to biased conclusions. PICs adjust for these dependencies by statistically accounting for phylogenetic relationships among species. After applying a threshold of absolute correlation greater than 0.2 to the corrected data, we observed that 83% of associations remained significant after correction. This underscores the robustness of our discovered MLS-AS associations.

To test the consistency of MLS-associations within the same gene, we examined genes harboring multiple MLS-associated AS events. We identified 34 such genes in the heart, 30 in the kidney, 24 in the lung, 23 in the brain, 22 in the skin, and 21 in the liver. Most of these genes (mean = 89%) showed consistent directionality of association with MLS. This high level of concordance strengthens the evidence for coordinated regulation of splicing in relation to lifespan. In a smaller subset of genes (mean = 11%), we observed discordant associations. These less common discordant patterns may reflect splicing-specific regulatory mechanisms or functional diversification of protein isoforms. These findings highlight the complexity of splicing regulation in lifespan determination.

### Relationship between alternative splicing events associated with body size and lifespan

Building on previous studies that link larger mammalian body size with longer lifespans[20], we sought to interrogate the potential role of

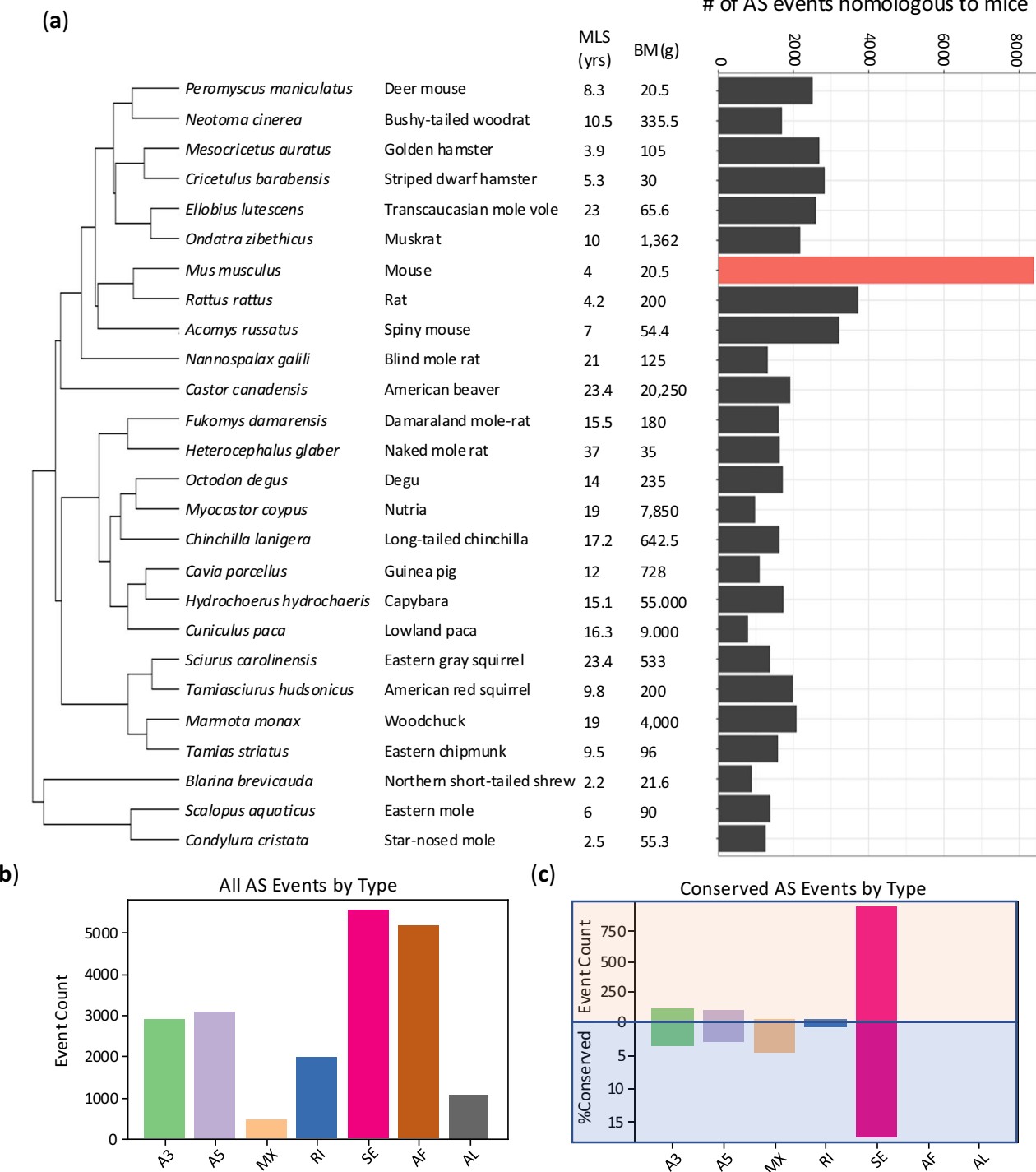

**Fig. 1 | Homologous alternative splicing events in Rodentia and Eutlipotyphyla species. a** The number of AS events homologous to the mouse reference. **b** Average distribution of seven types of AS events in a species, considering all AS events (including non-homologous ones), and (**c**) Conserved AS event types in at least 10 species, shown as both raw counts (positive y-axis) and conservation percentage within each type (negative y-axis).

alternative splicing in this relationship. By examining body mass (BM)-associated AS events across tissues, we identified, on average, 250 BM-associated AS events per tissue. Notably, these BM-associated events significantly overlapped with MLS-associated events (Fig. 2a, b, adjusted p-values < 10^{-4}, hypergeometric tests with FDR correction), supporting the interconnection between longevity and body mass.

Interestingly, the overlap between AS events associated with MLS and those linked to body mass was lowest in the brain, as demonstrated by the smallest Jaccard index values, a statistic measuring the

overlap between two sets (Fig. 2c, d). The enrichment for shared splicing events related to both MLS and BM is much weaker in the brain (adjusted $p = 2.87 \times 10^{-5}$) than in any other tissue (adjusted $p < 10^{-24}$ for each of the other tissues). This indicates that, in the brain, the way splicing correlates with lifespan operates more independently from body size, or the regulation of life span and body mass at the AS level is less interwoven in the brain than in other peripheral tissues. Intriguingly, when analyzing gene expression (instead of splicing) associated with MLS and BM, the brain did not show this distinction (Fig. 2e, f),

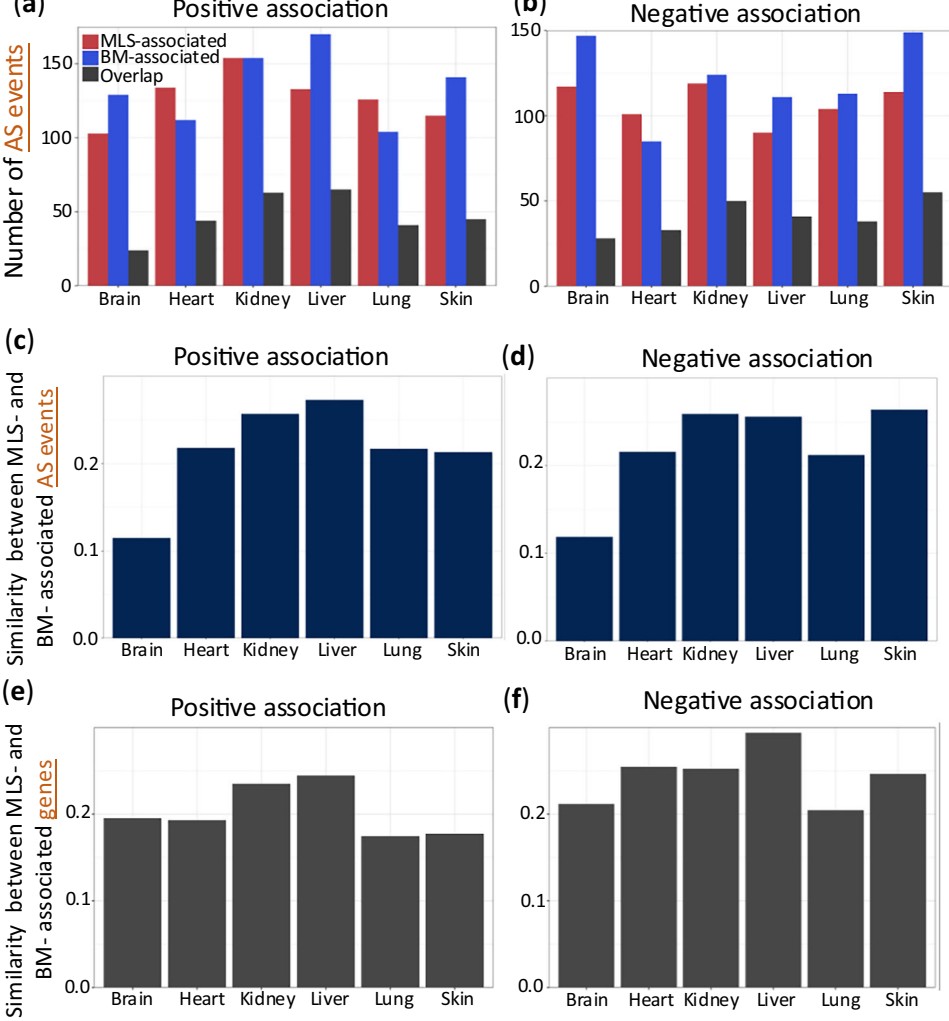

**Fig. 2 | MLS-associated alternative splicing events in mammals.** Bar plots illustrate the number of AS events where the percent spliced-in (PSI) level is positively (**a**) or negatively (**b**) correlated with MLS, body mass (BM), or both. Jaccard index values are calculated to show the similarity between BM-associated and MLS-associated AS events for (**c**) positive association, and (**d**) negative association. This index is determined by dividing the overlap of AS events between MLS-associated and BM-associated AS by the size of the union of MLS-associated and BM-associated AS events. Similarity between BM-associated and MLS-associated genes is calculated for (**e**) positive association and (**f**) negative association.

implying potential divergence in the regulatory functions of AS and gene expression during brain evolution.

Although BM and MLS are moderately correlated (Spearman correlation = 0.44), they represent distinct biological traits and are not redundant. To disentangle their individual effects, we applied the model MLS = $\beta_0 + \beta_1 \cdot BM + \beta_2 \cdot PSI$ for each identified MLS-AS event. Our results showed that PSI contributes independently to MLS in 59% of cases (where $p$-value for $\beta_2 < 0.05$ and $p$-value for $\beta_1 > 0.05$), while both PSI and BM jointly contribute to MLS in 35% of cases ($p$-values for both $\beta_2$ and $\beta_1 < 0.05$). Altogether, in 94% of MLS-associated AS events, PSI was a significant predictor even after accounting for BM, supporting the idea that splicing contributes independently to lifespan regulation.

Figure 3 illustrates two examples of AS events in genes that have been implicated in lifespan regulation. *Fn1* encodes fibronectin, a glycoprotein that binds to membrane-spanning receptor proteins integrins, and extracellular matrix proteins like collagen and fibrin[21–23]. An exon-skipping event in *Fn1* displayed a positive correlation with MLS (Fig. 3a). This observation aligns with prior evidence that dysregulation of *Fn1* AS can decrease both mean and maximum lifespans[24]. Figure 3b shows an exon-skipping event in *Arnt3* with a negative correlation with MLS. *Arnt3* encodes a circadian rhythm transcription factor (also known as *BMAL2*)[25,26], and *Arnt3* knockout mice show

premature aging phenotypes, such as shorter lifespan, sarcopenia, cataracts, decreased subcutaneous fat, and organ shrinkage[27]. Our analysis thus establishes an association between AS of *Arnt3* to MLS.

## Gene expression levels of MLS-associated alternative splicing events

Species with smaller effective population sizes tend to exhibit a higher average splicing error rate across their genomes[28]. This increase in splicing inaccuracy, largely manifesting in low-abundance isoforms or less functional splicing, results in greater AS. According to the molecular error hypothesis, much of the transcriptomic variations generated by AS, alternative transcription initiation, polyadenylation, or RNA editing may reflect random noise rather than finely tuned adaptation, and the diversity reduces with a gene's expression level[29–33]. This hypothesis suggests that genes with higher expression levels have a lower overall frequency of alternative splicing, as splicing errors are more costly for these genes.

Interestingly, effective population size is also, though not strongly, inversely correlated with longevity[28]. Building on both the drift barrier hypothesis and the molecular error hypothesis, if AS events negatively correlated with maximum lifespan (neg-MLS) are higher expressed genes whereas those positively correlated (pos-

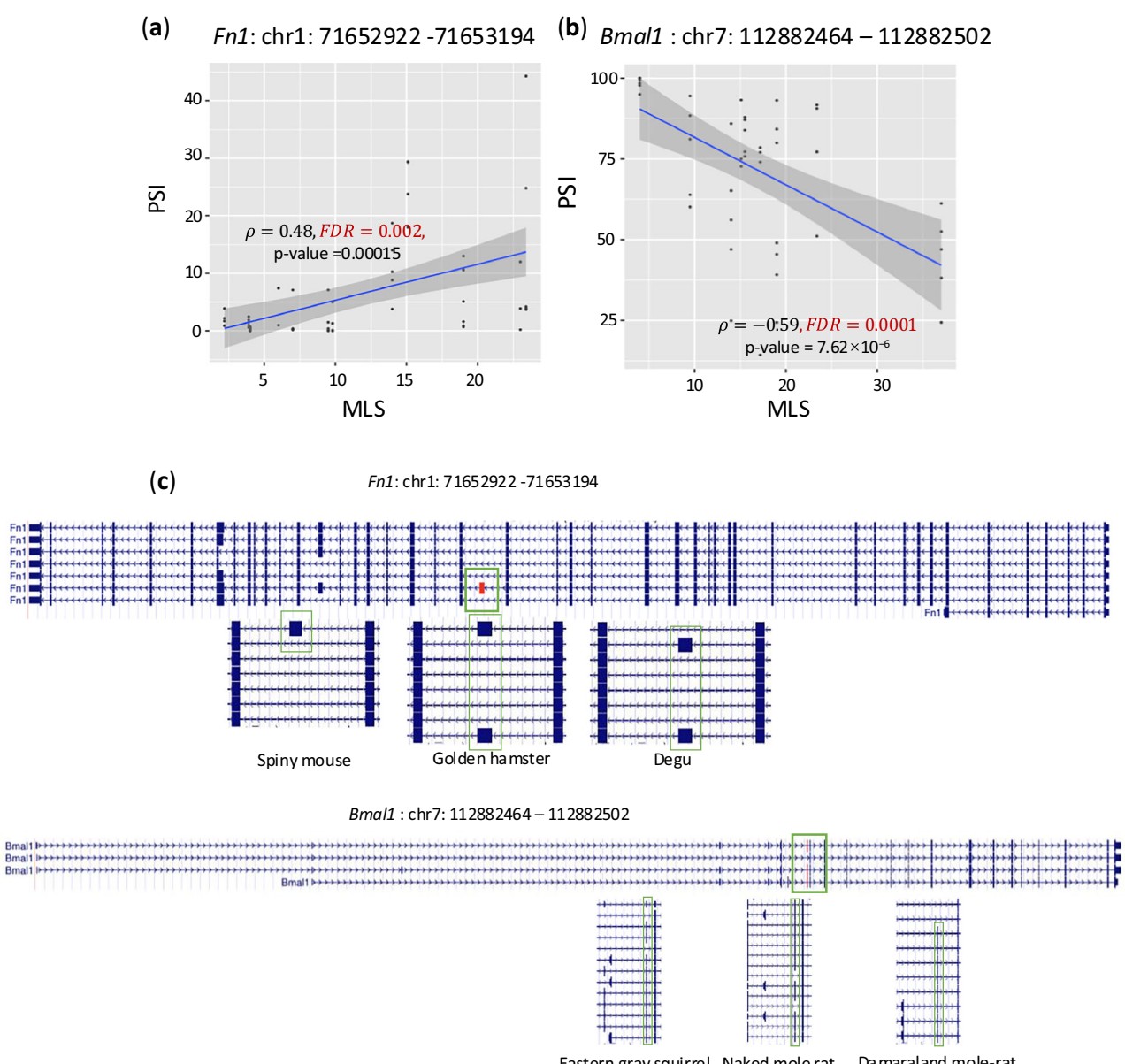

**Fig. 3 | Examples of MLS-associated alternative splicing events. a** An exon-skipping event whose PSI is positively correlated with MLS in the liver. **b** An exon-skipping event whose PSI is negatively correlated with MLS in the kidney. Spearman correlation coefficients ($\rho$) between PSI and MLS and exon coordinates (version GRCm39), two-sided *p*-value, and Benjamini-Hochberg corrected false discovery rate (FDR) are shown in the plot. The blue line shows the linear model regression fit, and the shaded gray area represents the 95% confidence interval for the regression line. **c** Gene structures from mouse (top) and three representative nonhuman mammals (bottom) are shown for each event. The alternatively spliced exon is highlighted in red. For each species, selected transcript isoforms illustrate the presence or absence of the exon, indicating conserved splicing patterns across mammals.

MLS) are lower expressed genes, these identified MLS-AS events could be due to transcriptomic noise. To test this, we compared expression patterns between pos-MLS and neg-MLS genes across six tissues.

The data does not support a universal trend of higher expression in neg-MLS genes or lower expression in pos-MLS genes (Supplementary Fig. 5). The overall distributions of gene expression values between these two groups were very similar in each tissue. In some tissues, Mann-Whitney U-tests comparing group means revealed slight differences (1% to 8%), but the direction of these differences varied. Neg-MLS genes exhibited higher mean expression than pos-MLS genes in the brain and skin ($p < 0.001$) but lower mean expression ($p < 0.05$) in the heart, kidney, and lung. In the liver, there was no significant difference ($p = 0.53$). Further analysis using QQ plots demonstrated these mixed trends among tissues. The heart, kidney, and brain showed a higher proportion of highly expressed genes among neg-MLS events (as illustrated in the QQ plots in Supplementary Fig. 5), while lung, skin, and liver showed the opposite trend for highly expressed genes. Overall, no consistent pattern emerged regarding the gene expression levels of MLS-associated splicing events. While some neg-MLS AS events tend to have higher gene expression, particularly in the brain, this is not consistent across tissues or a widespread pattern. Thus, the relationships between alternative splicing and MLS regulation are largely independent of gene expression levels, and the identified MLS-AS events are not random noise.

**MLS-associated alternative splicing events exhibit distinct patterns in the brain**

Interestingly, the brain tissue stood out as having a much larger number of tissue-specific MLS-associated AS events. Of all MLS-

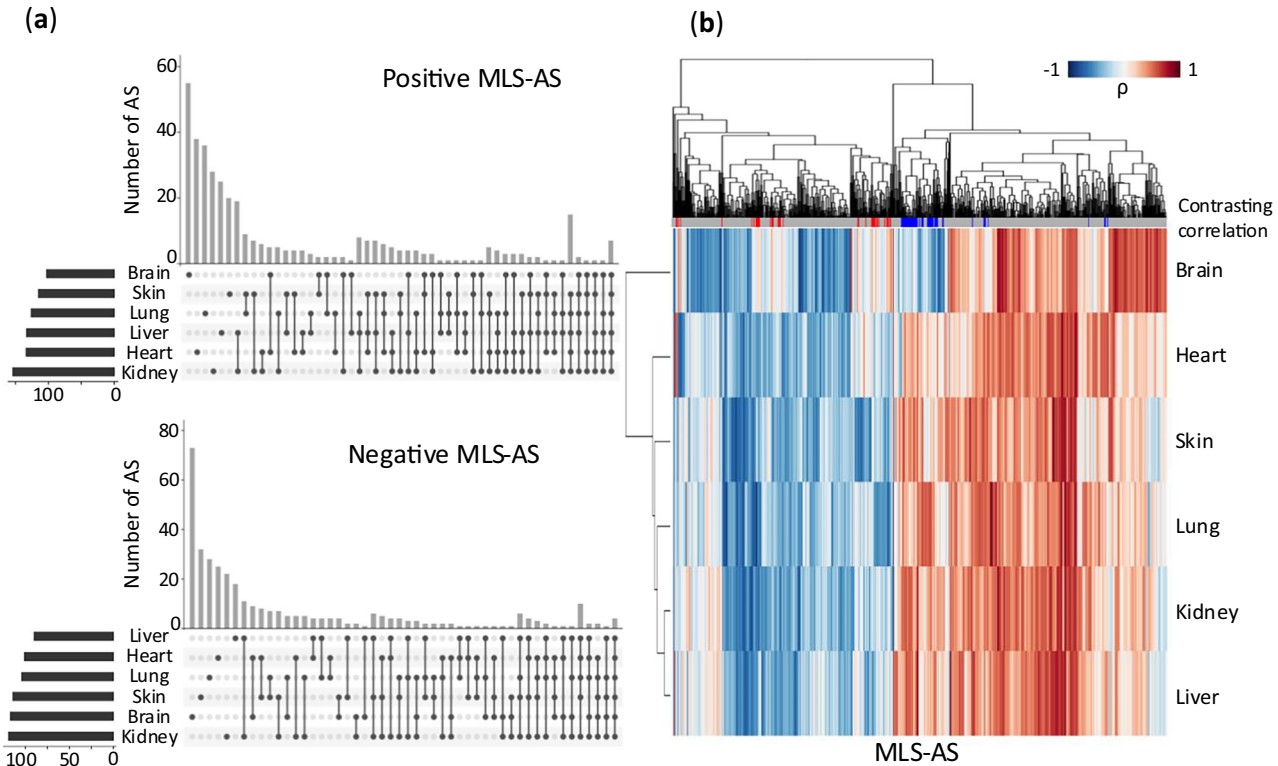

**Fig. 4 | Tissue specificity of MLS-associated alternative splicing events. a** Upset plots showing positive or negative MLS-correlated AS events identified in six tissues. **b** Hierarchical clustering of MLS-correlated AS events across tissues based on their Spearman correlation coefficients with maximum lifespans. Red or blue lines on the top grey bar denote AS events showing contrasting correlations between brains and other tissues. Red lines indicate positive correlations within the brain and negative correlations in the other five tissues. Blue lines indicate negative correlations within the brain and positive correlations in the other five tissues.

associated AS events, 342 out of 731 (46.8%) were associated with MLS in more than one tissue, while the remaining 389 were unique to a single tissue. The brain had two to three times more unique MLS-associated splicing events, both positively and negatively correlated, than any other tissue (Fig. 4a). This suggests that the brain may involve distinct splicing regulation for lifespan regulation that is not observed elsewhere in the body.

The distinctive role of AS in the brain is further highlighted by the hierarchical clustering of the association coefficients across tissues. As shown in Fig. 4b, although the absolute strength of the correlation between AS and MLS varied across tissues, most MLS-associated AS events consistently exhibited the same correlation direction in all tissues. However, for events exhibiting tissue-dependent divergent correlations, the divergence occurs between the brain and peripheral tissues. That is, certain MLS-associated AS events exhibited positive correlations in the brain but negative correlations elsewhere (contrasting correlation highlighted in red on the top bar in Fig. 4b). Conversely, some AS events displayed negative correlations in the brain but positive correlations in the other five tissues (marked with blue on the top bar in Fig. 4b). Figure 5 shows twelve MLS-associated AS events that exhibited opposite correlation directions in the brain versus other tissues. This divergence could be an example of antagonistic pleiotropy, where a biological factor has opposite effects on fitness in different tissues. That is, an AS event promotes the brain's longevity-related functions but incurs a trade-off in other tissues, resulting in opposite effects on MLS correlation. Another possibility is intrinsic negative correlations between the brain and peripheral tissues driven by certain splicing regulators that are active only in the brain. Remarkably, such a contrasting correlation pattern was rare in other tissues, with just one example each in the heart and liver.

## Functional enrichment of MLS-associated alternative splicing

To understand the biological significance of MLS-associated AS (MLS-AS) events, we conducted functional enrichment analysis using Enrichr. We examined genes harboring MLS-associated splicing events for enrichment in the Gene Ontology (GO) biological process terms and KEGG pathways. MLS-AS genes are significantly enriched in functional categories of mRNA processing, stress response, neuronal functions, and epigenetic regulation (Supplementary Data 3). Figure 6 summarizes the significantly enriched terms (FDR < 0.15), organized by functional categories.

Multiple enriched pathways involve cellular stress response, including Stress Granule Assembly, Negative Regulation of Defense Response to Virus, Regulation of PI3K/PKB Signal Transduction, etc. Stress response plays an essential role in safeguarding cellular and organismal integrity[34–36]. When cells encounter stressors such as oxidative damage, heat shock, viral infection, or metabolic fluctuations, they initiate intricate signaling pathways to mitigate harm and restore homeostasis. Efficient stress response mechanisms enable cells to quickly detect and neutralize damage, activate repair systems, and, if necessary, trigger programmed cell death (apoptosis) to remove irreparably compromised cells. Indeed, Regulation of Apoptotic Process (and related GO terms) is significantly enriched (Fig. 6). Apoptosis ensures that tissues maintain their functional capacity and that damaged cells do not accumulate[37,38]. Moreover, the adaptability of the stress response is crucial for long-lived species. Organisms with robust stress response networks can better withstand environmental and physiological fluctuations across their lifespan, delaying the onset of cellular dysfunction and chronic pathologies. Alternative splicing of stress response genes likely modulates organisms' competence for the efficiency and adaptability of stress response.

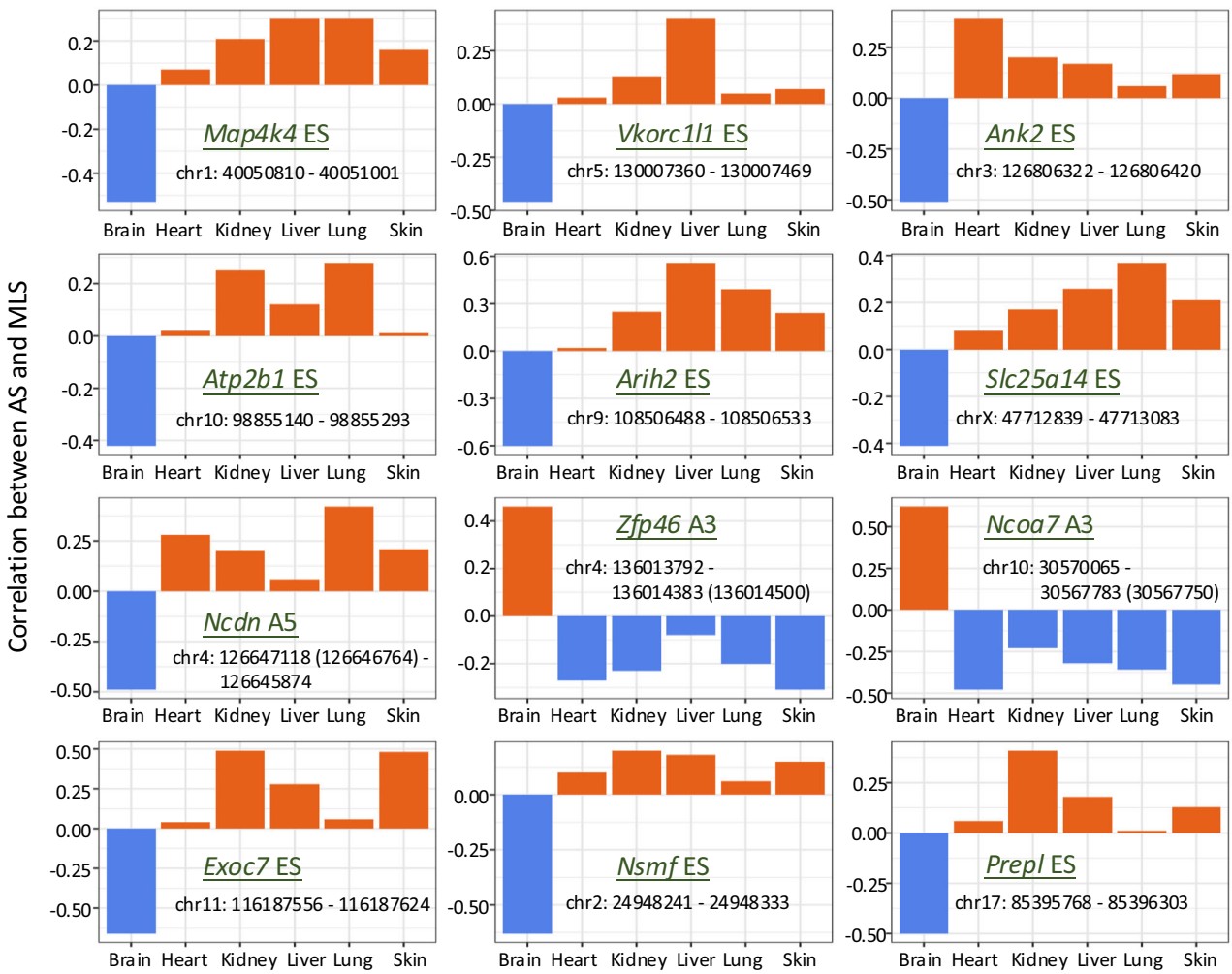

**Fig. 5 | Brain-specific MLS-associated AS events showing opposite correlations in other tissues.** For these AS events, their correlations with MLS are significant in the brain, but their correlations have opposite signs in other tissues. ES exon skipping, A5 alternative 5' splice sites, A3 alternative 3' splice sites. Exon coordinates are in version GRCm39.

Notably, mRNA processing steps and regulatory mechanisms are enriched, including Regulation of Alternative mRNA Splicing (along with many other splicing-related terms not shown in Fig. 6), mRNA Stabilization, Regulation of Translational Initiation, and RNA transport. Collectively, these comprise 17% of all enriched GO terms and encompass many stages of the mRNA life cycle. This underscores the importance of maintaining transcriptome integrity and proper mRNA processing for longevity. Errors or inefficiencies in any of these processes can result in the production of faulty or non-functional proteins, accumulation of aberrant RNA species, and ultimately, cellular dysfunction. For long-lived species, the ability to maintain precise and adaptable RNA processing systems is particularly critical. Accurate splicing ensures that essential gene products are expressed in correct proportions and forms, while alternative splicing provides an additional layer of regulatory flexibility, allowing organisms to rapidly adapt to changing physiological conditions without altering their genomic DNA. mRNA surveillance mechanisms, such as nonsense-mediated decay, also respond to cellular stress and safeguard the transcriptome by identifying and eliminating defective mRNAs that could otherwise give rise to toxic or deleterious proteins[39]. Thus, robust RNA processing and regulation are central to maintaining cellular homeostasis and adaptability, both of which are essential for healthy aging. Their enrichment suggests that longevity in some species may be achieved, at least in part, by optimizing alternative splicing mechanisms that control the transcriptome's quality, diversity, and responsiveness throughout the lifespan.

Additionally, we observed a significant enrichment of biological processes related to neuronal functions, such as Neuron Projection Morphogenesis, Axonogenesis, Regulation of Neuronal Synaptic Plasticity, Axon Guidance, Regulation of Synapse Organization, Dendrite Morphogenesis, Calcium Ion-Regulated Exocytosis of Neurotransmitter, Regulation of Neuron Migration, Synaptic Vesicle Endocytosis, etc. Neuro-related terms represent over 15% of all enriched GO terms, further reinforcing the distinctive role of brain-specific alternative splicing[40–49]. The pivotal role of neuronal function in determining organismal lifespan likely arises from the brain's central control over homeostasis and its orchestration of numerous physiological processes intimately connected to aging and longevity. Learning, memory, and adaptability of neural circuits are intricately regulated by pathways involved in axon formation and synaptic organization. Through these mechanisms, neurons can adapt to developmental cues and environmental stressors. Importantly, the brain influences systemic aging via neuroendocrine and autonomic pathways. For example, the hypothalamus integrates cues related to energy balance, stress, and circadian rhythms, coordinating hormonal outputs that impact metabolism, immune responses, and tissue regeneration. In this way, neuronal function serves as a central determinant of lifespan —supporting cognitive maintenance, regulating systemic physiology,

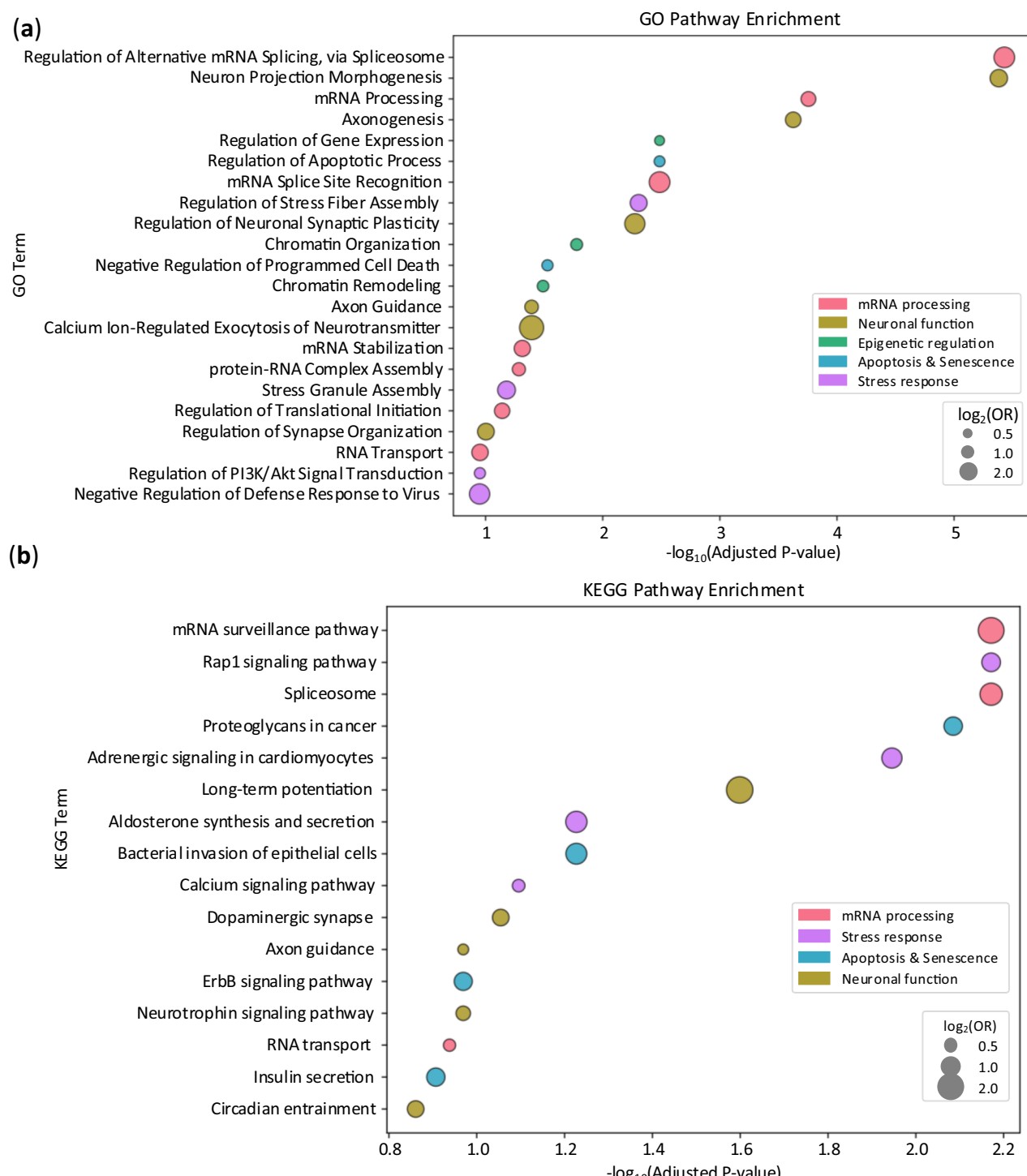

**Fig. 6 | Functional enrichment of genes with MLS-associated AS events. a** GO biological processes and (**b**) KEGG pathways enriched among genes harboring MLS-associated AS events, based on Enrichr analysis. Enrichment significance is determined using a one-sided Fisher's exact test with $p$-values adjusted for multiple comparisons by the Benjamini-Hochberg method, and is represented by the $-\log_{10}$(adjusted $p$-value) on the x-axis. Dot size indicates the magnitude of enrichment ($\log_2$ odds ratio), and color represents the term's functional category.

modulating immune activity, and conferring resilience to stress. In summary, these findings highlight the diverse biological processes modulated by alternative splicing to influence maximum lifespan.

To compare the respective contributions of gene-level and splicing-level regulation to lifespan, we compared the functional enrichment of genes with MLS-associated AS events and those with MLS-associated expression changes (as identified by Lu et al.). First, among the 576 genes with MLS-associated AS events (MLS-AS) and the

6952 genes with MLS-associated expression changes (MLS-gene), only 211 genes overlapped. Using a background of 17,511 multi-exon genes, this overlap yielded a $p$-value of 0.94 (Fisher's Exact test), indicating it is not statistically significant.

Comparing their pathway enrichments reveals a clear distinction. Lu et al. have found MLS-genes are enriched in GO group terms such as DNA replication, mitotic recombination, chromosome segregation, double-strand break repair, protein folding, mitochondrion

organization, cilium movement, and lymphocyte chemotaxis. They are also particularly enriched in various macromolecular metabolism pathways, including acylglycerol metabolic process, amino acid catabolic process, carbohydrate biosynthetic process, ribonucleoside biphosphate biosynthetic process, and hydrogen peroxide catabolic process. These findings align with current theories that emphasize DNA repair[50] and metabolic control in longevity. Proper regulation of macromolecule metabolism is particularly crucial for maintaining cell structure and function, and repairing damaged tissues[51,52]. These GO terms or functional groups are not prominent in MLS-AS genes. Overall, only five out of 192 GO terms from MLS-AS events are enriched in MLS-gene groups (FDR < 0.15 for both groups). When the threshold is tightened to FDR < 0.1, this overlap drops to just one exact shared term, and at FDR < 0.05, there are no overlapping GO terms between the two groups. We therefore used FDR < 0.15 for comparisons. Since different GO terms can still be related under one functional activity, we used REVIGO to organize the GO terms into functional categories. The results still highlighted a clear distinction. For example, among the 424 total GO terms enriched in MLS-genes, only three were related to neural function (0.7%) and ten to RNA-related processes (2.3%). On the other hand, MLS-genes are highly enriched for DNA Repair-related terms, a pattern that is not observed with MLS-AS events. This limited intersection underscores that alternative splicing modulation acts as an independent regulatory layer to gene expression regulation.

## Impact of lifespan-extending intervention on MLS-associated alternative splicing events

To explore if interventions that extend lifespan affect MLS-associated AS events, we analyzed mice treated with a Pregnancy-associated plasma protein-A (PAPP-A) inhibitor[53]. PAPP-A enzymatically cleaves insulin-like growth factor binding proteins (IGFBPs) to liberate IGFs and enhance local IGF signaling. PAPP-A inhibition reduces IGF signaling and has been shown to extend mouse lifespan[54,55]. Specifically, we assessed the change in percent spliced-in values (ΔPSI) between PAPP-A inhibitor treatment and controls for three categories of AS events: MLS positively correlated (pos-MLS), MLS negatively correlated (neg-MLS), and background (non-MLS-associated) AS events.

We found that pos-MLS events were more frequently upregulated (exon inclusion increased) in treated mice compared to control non-MLS-associated exon group, whereas neg-MLS events were the least likely to be upregulated (Supplementary Fig. 6). These trends held across a range of ΔPSI thresholds, based on the proportion of events showing significant PSI increases (FDR < 0.05 by Fisher's Exact Test on median read counts). This result shows that PAPP-A inhibitor, a longevity-enhancing intervention, tends to increase the inclusion of exons that are positively associated with MLS (Supplementary Data 5). In other words, such interventions appear to shift the splicing profile to more closely resemble that of long-lived species.

## Age-associated alternative splicing events across human tissues

Beyond studying non-human mammalian species, we also investigated how AS changes relate to human age across multiple human tissues using GTEx data. To tackle the complex relationships among individuals and better leverage available covariate information, we applied an elastic net regression model, which considers all AS events simultaneously. This method uncovered similar numbers of age-associated AS (age-AS) events in each tissue type or subtype (Fig. 7a). Sub-tissues within the same primary tissue, such as the various brain regions, in general exhibited higher pairwise similarity and tended to cluster closely (Fig. 7b). Still, many age-AS events were specific to a single sub-tissue (Fig. 7c). In total, brain sub-tissues accounted for over 1000 unique age-AS events, substantially more than other tissue types (Fig. 7d), suggesting the complexity and diversity of splicing regulation in the aging human brain.

We further investigated the heterogeneity among brain regions given the large number of age-AS events. We performed pairwise similarity analyses using the Overlap Coefficient (OC). Brain regions shared relatively few age-AS events with each other (lower OC: average of 0.38), indicating high regional specificity. In contrast, more overlap was seen among sub-tissues of the heart (OC: 0.42) or skin (OC: 0.49), suggesting splicing changes with aging are more homogenous within these tissues than in the brain. The Jaccard Index values also confirmed this pattern (brain = 0.22, heart = 0.25, skin = 0.31), further demonstrating that brain sub-tissues have less overlap in AS events compared to other tissues. This suggests that the elevated alternative splicing event count in the brain reflects genuine biological diversity in splicing regulation across brain regions.

Interestingly, several genes with MLS-associated AS events, such as *Hnrnpa2b1, Hnrnpk, Tra2b, FN1, and PTBP2*, have been reported to exhibit splicing changes related to aging or lifespan[16,56], raising the possibility of shared AS regulation between MLS and aging. To rigorously test this potential overlap, we compared MLS-associated AS events with age-AS events. Due to substantial differences in the background sets for these events, we employed a permutation test (10,000 samplings) to calculate an empirical p-value for the observed 131 overlapping events. Surprisingly, we found the 131 overlap was significantly less than expected by chance ($p < 0.0001$), indicating depletion rather than enrichment. We also cross-referenced our MLS-AS genes with GenAge[57], a curated database of aging-related genes, but found no significant enrichment, likely because splicing regulation reveals distinct candidates missed by gene-level analyses. Together, these results highlight alternative splicing as a distinct regulatory layer in lifespan control, complementing transcriptional mechanisms and offering further insights into the biology of longevity.

## Alternative splicing events associated with both MLS and age are enriched for intrinsically disordered proteins

We identified 139 AS events that were associated with both human age and species maximum lifespan across six tissues (i.e., Age-MLS AS events). The distribution of AS types for these overlapping events was similar to MLS-associated splicing, dominated by cassette exons (Supplementary Fig. 7). Further analysis for GO enrichment using DAVID[58] revealed intriguing findings for this group of genes.

The genes containing these overlapping splicing events were highly enriched for the GO term "REGION: Disordered" (FDR: $4.2E^{-56}$ for MLS-AS events, $1.2E^{-47}$ for age-AS events, and $1.2E^{-13}$ for overlapping events). Specifically, 90.6% of the genes with an Age-MLS AS event code for intrinsically disordered proteins (IDPs), which are known to play an essential role in signaling and stress-response[59]. As shown in Fig. 8, the proportion of IDP genes among those with Age-MLS AS events is significantly higher than all spliced genes (90.6% vs. 62.7%, $p = 1.05 \times 10^{-12}$, one-sided Fisher's Exact Test) or among those only associated with age (90.6% vs. 82%, $p = 0.009$, one-sided Fisher's Exact Test). Comparison between MLS-AS and Age-AS sets revealed that MLS-associated events contributed more strongly to this enrichment (88% vs. 82%, $p = 0.0008$). The higher proportion in the MLS-AS set than in the Age-AS (88% vs. 82%, $p$-value = 0.0008, one-sided Fisher's Exact Test) set suggested a potentially stronger contribution from MLS-associated events to this enrichment pattern than age-associated events.

These gene-level results prompted us to investigate whether MLS-associated or Age-associated AS events (or, for simplicity, longevity-associated AS events) preferentially occurred within IDP regions. Using the MOBI database of both curated and predicted IDP sequences, we assessed their overlap with AS regions and found that in both human and mouse, the longevity-associated AS events showed significantly greater overlap with IDP regions than did background AS events (e.g., human curated: $p = 3.16 \times 10^{-4}$, predicted: $p < 10^{-9}$; mouse curated: $p = 0.0023$, predicted: $p < 10^{-9}$; one-sided Fisher's exact test). These

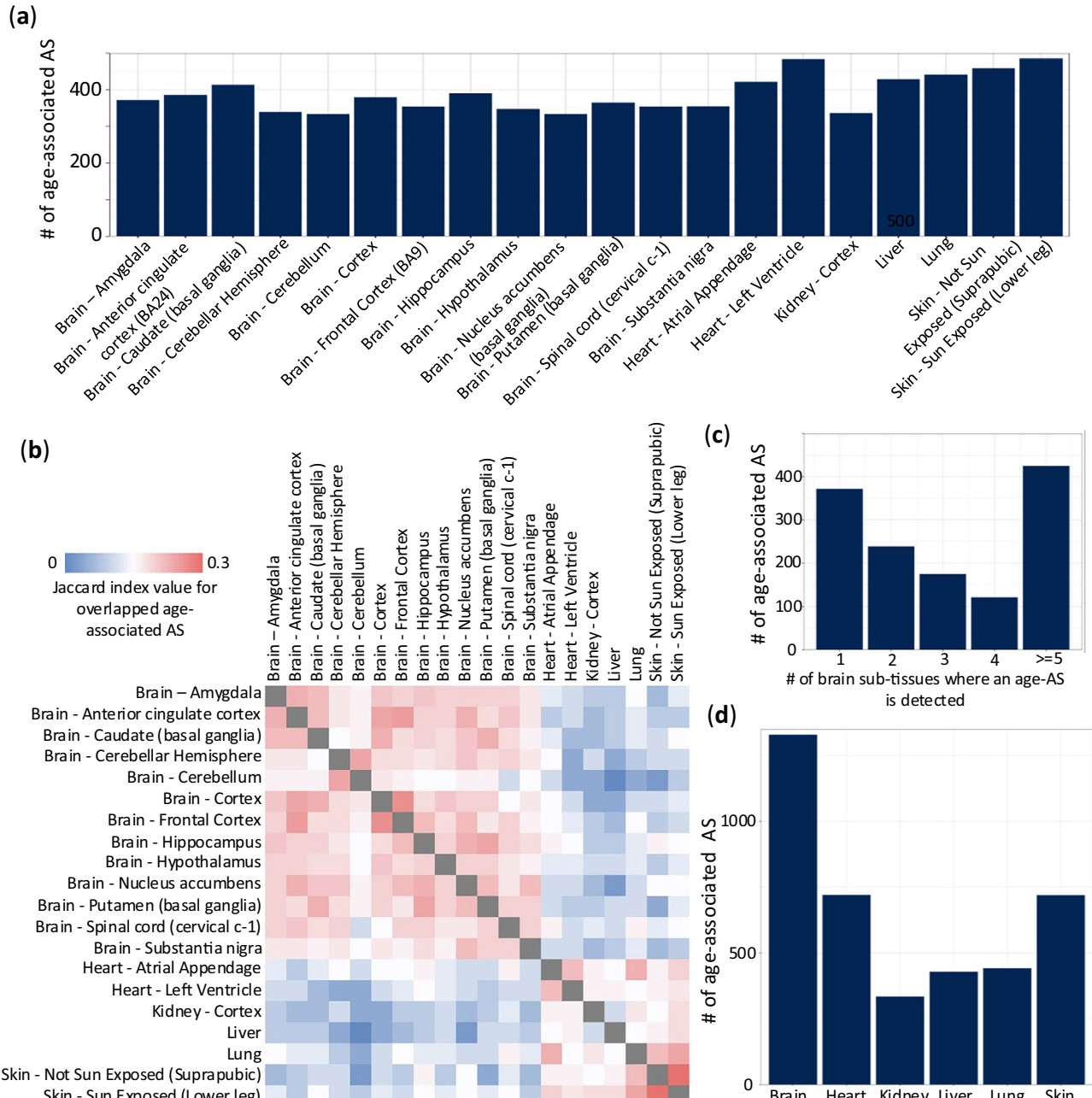

**Fig. 7 | Age-associated alternative splicing events in humans. a** The number of age-associated AS events in individual human tissues. **b** Heatmap showing pairwise (sub)tissue similarity based on Jaccard index values for age-associated AS events. These events were identified by the elastic net regression model. **c** Distribution of age-associated AS events in brain sub-tissues. **d** The number of age-associated AS events in six major tissues. Events from sub-tissues are pooled together.

findings suggest that AS may contribute to lifespan regulation, at least in part, by encoding protein disorder domains, potentially impacting molecular interactions and conferring functional adaptability, which could be essential to stress response and signaling during aging.

We next explored which biological pathways were enriched among these IDP-overlapping genes using Enrichr. In both age and MLS-associated AS gene sets, we observed significant enrichment in RNA processing-related pathways, such as Regulation of mRNA Splicing via the Spliceosome, Regulation of Alternative mRNA Splicing, and mRNA Processing. This suggests a possible feedback mechanism in which splicing factors themselves may be regulated via AS in IDP regions.

Age-associated AS events with IDPs were also enriched in transcriptional regulation pathways, such as Regulation of DNA-templated

Transcription and Regulation of Gene Expression. Interestingly, curated age-related AS events with IDPs were enriched in cardiac-specific developmental processes, such as Regulation of Cardiac Muscle Cell Proliferation and Positive Regulation of Cardiac Muscle Tissue Growth, hinting at potential tissue-specific splicing regulation during aging. In contrast, MLS-associated AS events with IDPs showed enrichment in pathways governing cellular maintenance and quality control, including Ubiquitin-Dependent Protein Catabolic Process, which is essential for proteostasis by eliminating damaged proteins, and Regulation of Apoptotic Process that eliminates dysfunctional cells. These pathways are recognized components of longevity regulation, suggesting that AS events occurring in IDP regions may contribute to lifespan determination by enhancing an organism's ability to manage molecular damage and maintain tissue homeostasis.

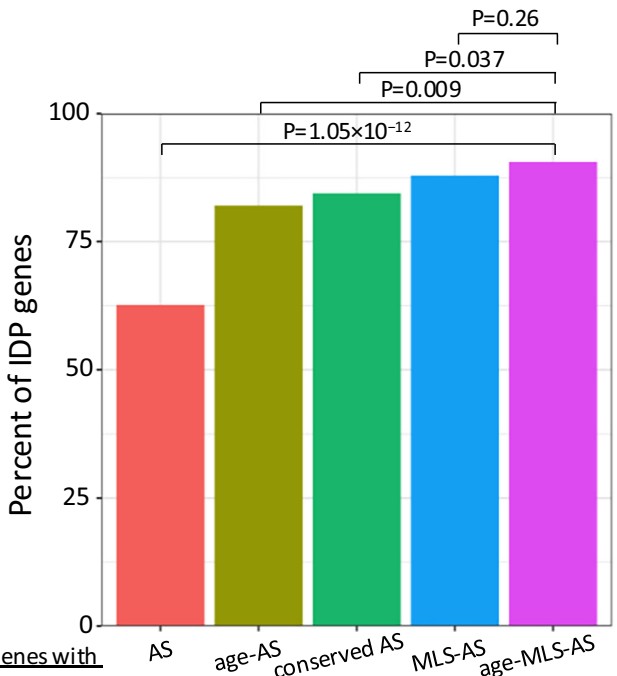

**Fig. 8 | Percentage of IDP genes in each AS event group.** IDP genes are defined by their association with the GO term "REGION: Disordered". One-sided Fisher's exact tests are used for the percentage comparison.

## RNA-binding protein (RBP) regulations of MLS- and age-associated AS events

To further understand the regulation of alternative splicing (AS) events associated with maximum lifespan (MLS) and age, we analyzed RNA-binding protein (RBP) motifs near the associated exons. RBPs play a key role in regulating alternative splicing by binding specific RNA sequences and influencing splice site selection[60]. By interacting with components of the spliceosome or other regulatory factors, RBPs can enhance or repress exon splicing. This regulatory mechanism is crucial for various biological processes, including development, cellular homeostasis, and response to environmental cues[61].

We employed RBPmap[62] to infer RBP binding motifs for each AS event. RBPs exhibiting enriched motif occurrences in a particular AS group were deemed potential "master" upstream regulators of that group. Details of the enrichment analysis are in Methods. Figure 9 shows the enrichment ratios for the top 20 RBPs in the MLS- or the Age-associated AS group, with AS events from various tissues pooled together. The MLS-associated AS group showed substantially higher RBP motif enrichment than the Age-associated AS group ($p = 0.00033$, paired t-test), implying that MLS-associated AS events are subject to a more coordinated regulation by RBPs and seem to be controlled by a more defined set of RBPs.

Many of these top 20 RBPs are known to be involved in certain aging processes, including neurodegeneration, cellular senescence, and stress responses. For example, FUS (fused in sarcoma), a key regulator of RNA processing, is involved in RNA metabolism and stress response. Mutations in *FUS* are linked to neurodegenerative diseases such as amyotrophic lateral sclerosis (ALS) and frontotemporal dementia (FTD)[63,64]. Another example is RNA-binding protein 8a (RBM8A), which is downregulated during aging[65] and plays a role in several age-associated progressive neurological diseases, including Alzheimer's disease[66]. The enrichment in these two RBPs highlighted their potential roles in the age-associated decline in neuronal function. RBPs, such as EIF4G2, SRSF5, and YBX1, have been implicated in cellular senescence, a key mechanism driving the aging process[67–70]. HNRNPK and HNRNPL regulate splicing and mRNA stability during the

stress response and apoptosis process[71–74]. Upregulating PPARG-related coactivator 1 (PPRC1) ameliorates oxidative stress, reduces inflammation, and extends lifespan[75]. IGF2BP2, another enriched RBP, is associated with growth factor regulation and has been linked to aging[76,77]. RBPs like FXR1 and SAMD4A are involved in mRNA stability and stress granule formation, linking them to cellular stress responses during aging[78–81]. Finally, ZC3H10 undergoes age-related epigenetic changes in primates[82], indicating an additional regulation layer. Collectively, these findings suggest an intricate network of RBPs regulating MLS and Age-related AS events.

To investigate whether expression levels of these RBPs are linked to lifespan, we assessed the correlation between their expression and MLS across six tissues based on the prior study[9]. Of the 39 unique key RBPs (the union of the top 20 motif-enriched RBPs from each tissue), only five (*ESRP1*, *FUS*, *MBNL1*, *PCBP2*, *RBM8A*) overlapped with a previously curated list of genes whose expression correlates with MLS[9]. This limited overlap suggests that analyzing splicing regulation can identify a distinct set of potential lifespan regulators missed by studies focusing on gene expression alone. This reinforces the biological relevance of RBP-mediated splicing regulation in mammalian longevity.

In addition to identifying master RBPs enriched for the entire MLS-associated AS event group, we investigated RBP motif enrichment by clustering individual AS events. Motif enrichment was assessed by calculating the rankings (represented by percentiles) of length-normalized motif occurrences for each MLS-associated AS event among all considered AS events. Hierarchical clustering was then used to detect RBP and MLS-AS event clusters.

As shown in Fig. 10, two prominent clusters emerged (details in Supplementary Data 2). Cluster 1 (green box in Fig. 10) was a large cluster consisting of 51 RBPs and 225 MLS-associated splicing events. Notably, the left clade of the RBP branch in Cluster 1 overlaps with Cluster 2 (grey box in Fig. 10), which contained 24 RBPs and 107 MLS-associated splicing events. The overlap suggests shared RBP-dependent regulatory mechanisms between the two clusters. Several RBPs within these clusters have established roles in aging and longevity. For example, CPEB1 and CPEB4 have a direct impact on aging by regulating the mammalian cell cycle, particularly during senescence[83,84]. ELAVL4 and HuR play a role in aging by regulating mRNA stabilization[85,86]. HNRNPC and CPEB2 contribute to aging regulation through their interactions with p53 and other aging-associated factors[87,88]. Interestingly, applying a similar analysis on AGE-related AS events revealed an enriched RBP cluster sharing almost the same set of RBPs as in Cluster 1 (Supplementary Fig. 8).

Examining motif frequencies across the 26 species revealed that certain RBP's motif occurrence correlated strongly with splicing inclusion levels (PSI values) for specific MLS-associated AS events, supporting their regulatory interactions (Supplementary Data 4). The number of RBPs showing significant PSI correlations differed by tissues, with liver showing the highest number (17), followed by lung (16) and heart (15), while skin and brain exhibited fewer associations (3 and 5, respectively). Some RBPs, such as ELAVL4, YBX1, TIA1, QKI, and HNRNPD, were found across multiple tissues. These findings suggest that specific RBPs consistently interact with local motifs of MLS-associated AS events across species, highlighting their potential roles in regulating lifespan-associated splicing.

## Discussion

The regulatory mechanisms underlying the remarkable wide range of maximum lifespans across mammals, with over a hundred-fold variation, are unknown[89]. In this comparative study leveraging the natural variation in species' lifespans, we explored the impact of alternative splicing as a previously overlooked layer of lifespan regulation. Our results suggest alternative splicing as an important factor correlated with both the evolved differences in mammalian lifespan and human

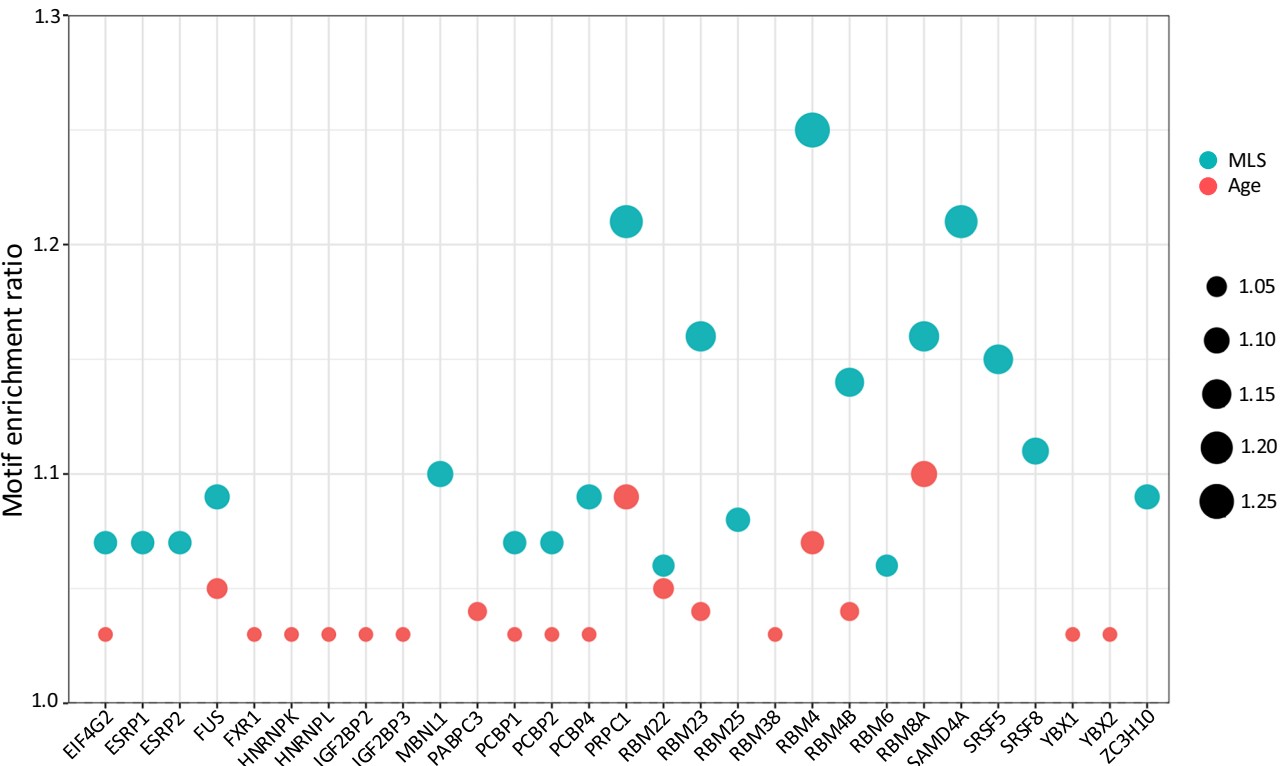

**Fig. 9 | Top 20 most enriched RBPs with motifs in MLS- or age-associated AS events.** AS events from various tissues are pooled. The motif enrichment of an RBP is based on comparison to randomly selected AS event groups with matching AS type compositions. The y-axis indicates the motif enrichment ratio, which is also represented by the dot size.

aging, and show potential molecular mechanisms (e.g., RNA processing, neural regulation, IDPs, RBP regulators) underlying these effects. Remarkably, nearly half of the highly conserved alternative splicing events show a significant association with maximum lifespan in at least one tissue. While many of these splicing associations are consistent across different tissues, the effects in the brain stand out as particularly distinct.

**Brain-specific splicing associated with MLS**

Brain-specific splicing changes were the most prevalent among MLS-associated events, and in some cases, their direction of change contrasted with those seen in other tissues (Figs. 4 and 5), pointing to unique regulatory mechanisms in the nervous system's contribution to longevity. Although the precise functions of many of these brain-specific events are unclear, previous studies have suggested a critical role for AS in regulating neuronal longevity and animal survival. For example, brain-specific alternative splicing of a microexon in pro-apoptotic gene *Bak1* is required to attenuate neuronal apoptosis to enable long-term neuronal survival[90,91]. Remarkably, deleting even one copy of this microexon is sufficient to inflate neuronal cell death and cause premature animal mortality. These findings underscore the unique role of AS in brain-related processes and have potential implications for neurodegenerative diseases[44].

We examined the PSI values of brain-specific MLS-associated AS events in five non-brain tissues. In these other tissues, only 3.84% to 7.85% of PSI values were zero, indicating that most of these events are not exclusive to the brain. Rather, they are broadly expressed across tissues but show significant associations with MLS only in the brain.

To further investigate the unique splicing landscape of the brain, we tested whether brain-specific MLS- or age-associated AS events tend to occur in longer genes, since the brain is known to express many long genes[92]. However, we found no significant difference in gene lengths between brain-specific AS genes and those from other tissues

for either MLS- or age-associated AS events (MLS $p = 0.803$, age $p = 0.515$, t-test).

We also tested whether these AS events preferentially occur in brain-specific genes. Although a greater number of MLS- and age-associated AS events overlapped with brain-specific genes (11 and 38 overlaps, respectively), this reflects the overall abundance of tissue-specific gene numbers in the brain rather than a true enrichment (MLS $p = 0.49$, age $p = 0.1$, Fisher's exact test). Taken together, these findings show that the distinct AS patterns observed in the brain are not primarily explained by differences in tissue expression, gene length, or brain-specific splicing. Instead, they may reflect unique tissue-specific regulatory programs or the specialized functional demands of neurons.

The distinction of brain splicing also emerged from our investigation into the overlap between MLS-associated and BM-associated AS events. While all peripheral tissues show about the same degree of overlap between these two groups of AS events, the overlap in the brain is roughly half that observed in other tissues (Fig. 2a–d), despite similar numbers of events across tissues. This reduced overlap suggests that MLS-associated splicing in the brain is influenced by factors distinct from those associated with body mass. Interestingly, this brain-unique decoupling is not seen at the level of gene expression association (Fig. 2e, f), implying that brain regulation of lifespan may entail tissue-specific alternative splicing regulatory mechanisms[45].

**Analytical strategies for cross-species splicing analyses**

We applied elastic net regression on the GTEx human dataset to identify age-associated AS events. This approach effectively accounts for complex relationships between samples and incorporates measured covariates, setting it apart from the method recently used by Garcia-Pérez et al.[93], which employed a different analytical framework and focused on predicting PSI values from age and covariates. The robustness of our model is demonstrated by high Jaccard index values

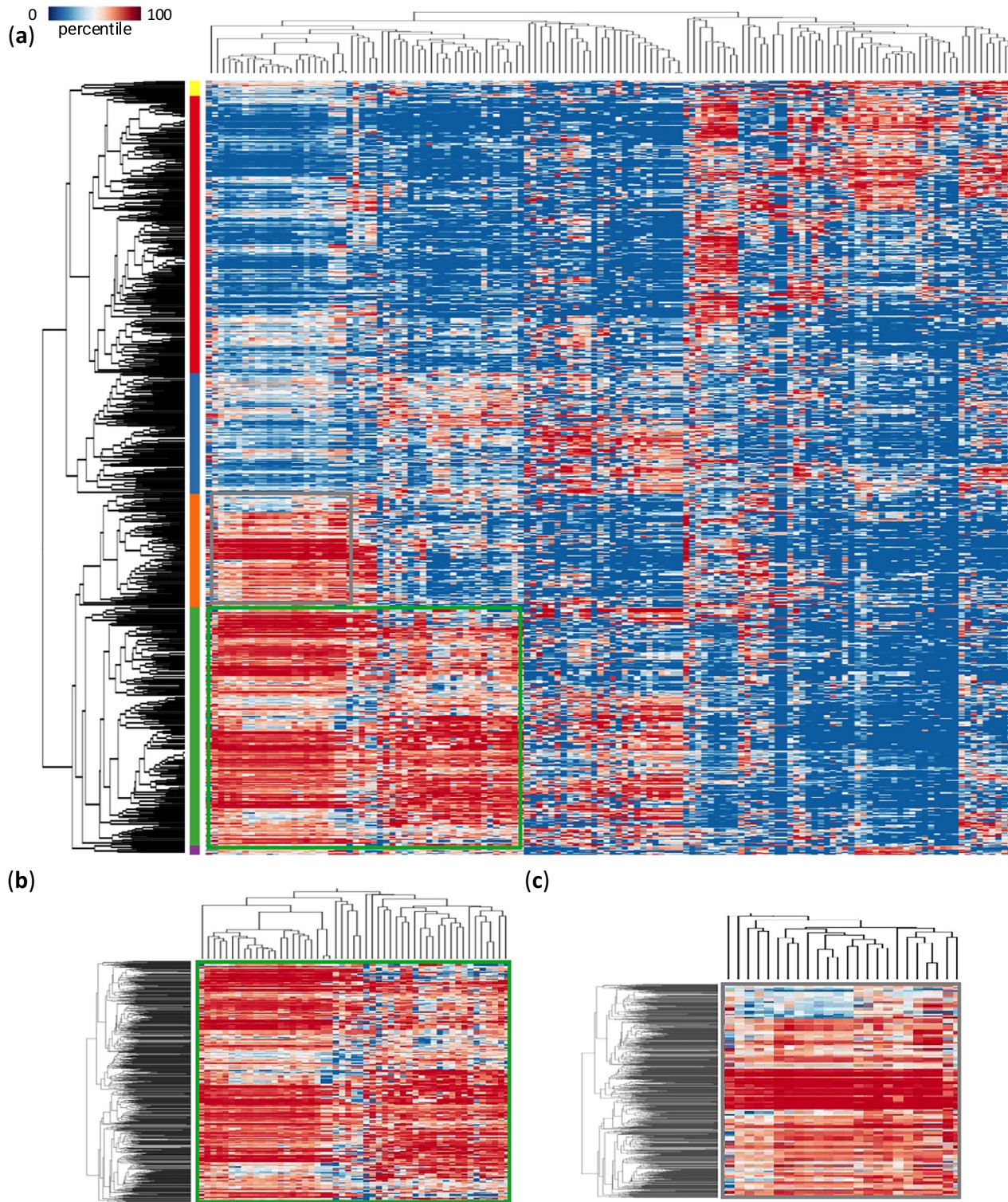

**Fig. 10 | RBP motif enrichment patterns in MLS-associated AS events. a** Heatmap showing hierarchical biclustering of MLS-associated AS events based on their motif enrichment across RBPs. Detailed view of cluster 1 (**b**) and cluster 2 (**c**). Details about the AS events and RBPs in the clusters can be found in Supplementary Data 2.

between similar tissues and sub-tissues (Fig. 7b). In contrast, traditional regression models that analyze each AS event in isolate, as in a previous study[94], yielded lower tissue similarity (Supplementary Fig. 9). This highlights the advantage of elastic net regression in simultaneously modeling multiple AS events.

To rigorously assess the model's effectiveness, we conducted 500 permutation tests by randomly shuffling age labels across samples. Under the null hypothesis, these permutations yielded an average of

approximately 100 significant AS events per iteration, providing an empirical estimate of the false positive rate. By contrast, our actual analysis identified approximately 400 significant AS events, substantially exceeding the permutation-based expectation. Applying a cutoff to the coefficients from elastic net selection can further reduce false positive rates, thereby offering a subset of events with higher confidence. Without coefficient filtering, the median FDR was ~0.25 across tissues; raising the cutoff from 0.1 to 0.7 reduced median FDR

from 0.22 to 0.098. Notably, AS events identified in the permutation tests showed no enrichment for age-related pathways. Together, these results support that many of the identified AS events reflect true biological signals.

While the elastic net model proved effective with human data for identifying Age-associated AS events, it was not suitable for non-human data due to its reliance on complete data matrices. Despite our comprehensive pipelines, variations in reference genome quality and RNA-seq coverage across species limited our ability to detect all AS events in non-human samples. Additionally, alternative spliced regions evolve faster than constitutive regions[95]. Many splicing events in the non-human dataset lack homologous counterparts across the 26 mammalian species analyzed, resulting in incomplete data. Therefore, we opted for the Spearman method for the non-human dataset.

In our cross-species comparative analysis, we applied a stringent pipeline to identify homologous alternative splicing events, uncovering a robust set of conserved events across mammals. Although this approach may underestimate the total number of conserved AS events, it effectively minimizes false positives and prioritizes confident homology. The correlation between evolutionary distance and the number of shared AS events is consistent with evolutionary expectations. Our finding that cassette exons are more frequently conserved than other AS types also matches with results from previous large-scale vertebrate analyses by Barbosa-Morais et al.[96] and Merkin et al.[97]. Notably, we benchmarked our pipeline against Merkin et al.'s dataset and observed a high concordance (99.3%) in homologous mouse–rat events. Furthermore, organ-level transcriptomic studies in mammals[98,99] support that AS conservation follows patterns of evolutionary divergence and reflects functional constraints across tissues.

Future improvements in transcript annotations for humans and other mammals will enhance the detection of homologous AS events in non-human species. These advances will pave the way for the application of more sophisticated models, such as elastic net regression models, on broader datasets of non-human mammals, potentially yielding more robust outcomes and a deeper understanding of the intricate AS landscape in the context of lifespan regulation.

## Distinction between human ages and species lifespan

Our analysis of human age-associated splicing and its intersection with species MLS-associated splicing reveals an interesting pattern: while there is some commonality (notably in genes encoding intrinsically disordered proteins), many splicing events linked to human age do not correspond to those that distinguish long-lived vs short-lived species. This is expected because aging within a species and evolutionary changes in lifespan between species occur over different timescales and under different evolutionary pressures. Nevertheless, the overlap we do observe highlights the presence of potential core mechanisms (like splicing in stress response genes) that influence aging across diverse mammalian lineages.

We also observed that RBP binding motif enrichment is more pronounced in MLS-associated AS events than in those linked to age. Several factors may contribute to this difference. MLS is an intrinsic species-level trait shaped by evolutionary adaptations that sustained physiological functionality and homeostasis, presumably governed by complex, genetically programmed regulatory mechanisms such as RBP activity. In contrast, aging is a gradual, time-dependent process influenced by environmental-organismal interactions, often characterized by progressive deregulation and uncontrolled molecular changes. Therefore, the regulation of MLS-associated AS events may be subject to more precise and evolutionarily conserved controls compared to more variable splicing changes seen with chronological aging. Such nuanced distinctions underscore the importance of considering MLS as a biologically distinct axis in transcriptomic aging research, suggesting the involvement of specialized regulatory mechanisms beyond those conventionally associated with chronological aging.

## Limitations of the study

Cell type composition is a known confounder in transcriptomic analyses, as variations in cellular makeup, even within the same tissue across samples, can affect the inferences of both gene expression and splicing levels. The MLS-AS events we identified exhibit tissue-level differences between long-lived and short-lived species, which may reflect the changes at the level of individual cells or shifts in overall cell type compositions, or both. Because information on cell-type-level splicing or the exact cell type compositions within tissues is unavailable for most species, current research is unable to disentangle these possibilities. Future studies incorporating single-cell RNA-seq in combination with applying cell-type deconvolution methods could provide deeper insights into the cell-type-specific regulation of AS events in the context of lifespan regulation.

While our analysis focused on RBP binding motifs characterized in humans and mice, we acknowledge that different species may have evolved distinct splicing regulatory sequences and RBP binding preferences. The current lack of high-quality RBP motif databases for non-model organisms limits our ability to perform species-specific RBP enrichment. As more RBP binding data become available for a broader range of species, future studies can explore the evolutionary dynamics of RBP-mediated splicing regulation and its contribution to inter-species differences in maximum lifespan.

Our comprehensive across-mammal analysis reveals that alternative splicing is an orthogonal regulatory layer to gene expression regulation in shaping the evolution of maximum lifespan. The distinct splicing programs in long-lived species, especially in the brain, open avenues for future research into the biological foundations of extended lifespans. We identified candidate splicing events and their regulatory RBPs as promising targets for future functional studies to test their roles in aging. Ultimately, these insights into splicing and lifespan regulation could inform the development of strategies to improve health span and address age-related diseases.

## Methods
### Data acquisition
We analyzed RNA-seq data from six tissues: brain, heart, kidney, liver, lung, and skin, comprising 577 samples from 141 individuals, representing 26 Rodentia and Eulipotyphla species. Data was downloaded from the GEO database (GSE181413 and GSE190756)[9]. The MLS of these species (range 2.2–37 years) was obtained from the same study[9]. For the human data, transcript- and gene-level expression, splicing junction read counts, and other sample covariates were acquired from the Genotype-Tissue Expression (GTEx) consortium (https://gtexportal.org/)[17]. Age is the age of the donor at death.

### RNA-seq-based de novo transcriptome assembly
We implemented an analysis pipeline (similar to Lu et al.[9]) to perform the de novo transcriptome assembly (Supplementary Fig. 1). The RNA short sequence reads were trimmed with Trim_Galore v0.6.6 (https://github.com/FelixKrueger/TrimGalore) to remove poor-quality reads and adapter sequences (Phred scores <20). We aligned the processed reads to the reference genome using HISATS2 v2.2.1[100] (parameters: --dta --very-sensitive). If a reference genome was unavailable, we aligned reads to the closest related species' genome (details summarized in Supplementary Data 1). Reference genome sequences were downloaded from the UCSC Genome Browser. For each RNA-seq sample, we used the aligned reads to conduct the de novo transcriptome assembly with StringTie v2.1.5[101] using the default setting. StringTie provides both gene-level and transcript-level expression quantification. The assembled transcripts from all samples for each species were merged using the StringTie merge mode with the parameter -T 0.5 -g 50, yielding a non-redundant set of transcripts for the corresponding species.

## Homologous alternative splicing identification

We used the GENCODE annotations for human (v26) and mouse (vM30), alongside other mammalian annotations obtained from the de novo transcriptome assemblies, to identify seven types of alternative splicing (AS) events: cassette exon (exon skipping or skipped exon), alternative 5' splice sites, alternative 3' splice sites, mutually exclusive exons, retained introns, alternative first exons, and alternative last exons. This extraction was conducted using SUPPA v2.3[102].

To determine homologous AS events, we constructed a BLAST database using mouse exon or intron sequences (mm10) as the reference (anchor). The query sequences consisting of the exonic or intronic regions involved in AS events from other mammalian species, including humans, were blasted against the anchor database. Homologous AS events were defined as those present in both species, with at least 80% alignment of the involved exonic or intronic sequences. For certain AS types involving two distinct exons (e.g., mutually exclusive exons and alternative first and last exons), both needed to exhibit qualified alignments to their homologous counterparts. Specific query regions used for BLAST analysis can be found in Supplementary Fig. 2. If multiple sets of sequences met the criteria, the one with the highest alignment percentage was selected. AS events with homologous counterparts (against mice) in at least ten species were included in subsequent analyses.

## Alternative splicing and gene expression quantification

The transcript-level and gene-level expression for each non-human mammal sample was estimated using the StringTie expression estimation mode. Human transcript-level expression data were directly downloaded from GTEx. Subsequently, SUPPA was employed to quantify the splicing level of each AS event (percent-spliced-in, or PSI), utilizing transcript-level expression data.

## Models for detecting MLS- or age-associated alternative splicing events

To determine MLS-associated AS events, we calculated Spearman's rank correlation coefficient between each event's splicing ratios, PSI (percent-spliced-in), and maximum life spans across the 26 non-human mammalian species. To ensure sufficient statistical power, we considered only AS events whose genes were expressed with TPM (transcripts per million) >0.5 in at least 1/3 of tissue samples. Only AS events with homologs in at least ten non-human mammal species and exhibiting a PSI range ($|\text{max PSI} - \text{min PSI}|$) $\geq 20$ were considered. Events with an absolute correlation >0.4 and FDR < 0.05 were declared significant. This correlation cutoff was chosen to balance stringency and yield; a similar threshold has been used in related cross-species analyses.

To identify human age-associated AS events using the GTEx data, we implemented an elastic net regression model, incorporating both L1 (Lasso) and L2 (Ridge) regularization that simultaneously considers all splicing events and existing covariates to remove potential confounding factors. The covariates include sex, the top five genotyping principal components, 15 PEER (probabilistic estimation of expression residuals) factors, sequencing platform, and protocol. The same set of covariates was used in the GTEx sQTL analysis (GTEx_Analysis_v8_sQTL_covariates, available at the GTEx Portal)[103]. The elastic net model (1) was built as follows:

$$Age_i = \beta_0 + \sum_{k=1}^{n} \beta_k * PSI_{k,i} + \sum_{k=1}^{m} \beta_{n+k} * Cov_{k,i} + \varepsilon_i \quad (1)$$

where $Age_i$ denotes the age of human sample i, $PSI_{k,i}$ is the $k^{th}$ AS event, $Cov_{k,i}$ is the $k^{th}$ covariate provided by GTEx, $\beta s$ are the corresponding coefficients, and $\varepsilon_i$ is the error term. We restricted the analysis to events with gene TPM > 0.5 in at least 1/3 of tissue samples and PSI range $\geq 20$ across individuals, similar to the MLS analysis. An AS

event with a non-zero coefficient in the elastic net model was considered age-associated.

## Phylogenetically corrected correlations

To assess the correlation between AS and MLS while accounting for the evolutionary relationships among species, we calculated the phylogenetically independent contrasts (PIC) using the PSI values obtained from SUPPA as input. The "pic" function from the R package APE was employed to conduct PIC[104]. The phylogenetic tree for the species under consideration was constructed using VertLife[105].

## Functional enrichment analysis of genes containing MLS-associated alternative splicing events

To investigate the functional relevance of genes harboring MLS-associated AS events, we performed enrichment analysis using Enrichr[106], a widely used and up-to-date web-based tool for gene set enrichment analysis. Analyses were conducted using gene sets from the Gene Ontology database (GO)[107] and the Kyoto Encyclopedia of Genes and Genomes (KEGG) database[108]. Enriched terms with FDR < 0.15 were considered statistically significant. We used a custom background for the enrichment analyses, including genes that are alternatively spliced and expressed in at least one studied tissues.

## RNA-binding protein analysis for MLS (or age)-associated alternative splicing

To examine the upstream regulators of MLS- or AGE-associated AS events, we conducted the following analytical approach. The exonic and (or) intronic sequences and flanking sequences (±100 nucleotides for exon and ±50 nucleotides for intron) used in the homologous AS identification were provided to RBPmap[62], a web-based tool for mapping binding sites of RNA-binding proteins (RBPs). Default parameters and consensus RNA binding motifs for both mice (for MLS-AS) and humans (for AGE-AS) were applied in the analysis. For each RBP, we counted the motif occurrences and normalized them based on the region length for each AS event.

To identify enriched RBPs for our AS groups of interest, we compared the MLS- or AGE-associated AS group with randomly selected groups of AS events matching the same composition of AS types observed in the target group (MLS or age group). This process was repeated 500 times to generate multiple random groups. The enrichment ratio was computed by comparing the average length-normalized motif occurrence in the target group with that in the 500 random groups.

At the individual AS event level, we computed the ranking of the length-normalized motif frequency for a specific RBP within each AS event, comparing all considered AS events. Following this, hierarchical clustering was conducted using these rankings to uncover noteworthy RBP and AS event clusters.

To assess the functional relevance of RBP motifs to splicing directionality across species, we expanded our motif analysis to 26 mammals using species-specific genome assemblies and cross-referenced mouse/human RBP motif databases. For each MLS-associated AS event, Spearman's rank correlation was computed between length-normalized motif frequency and PSI values across species, and those with more than 10 data points, an absolute correlation >0.4, and FDR < 0.05 were declared significant.

## Reporting summary

Further information on research design is available in the Nature Portfolio Reporting Summary linked to this article.

## Data availability

The data used in this study are available in Zenodo under accession code https://zenodo.org/records/16042325. The previously published datasets used for this analysis are available in the Gene Expression

Omnibus (GEO) database under accession codes GSE181413 and GSE190756. Source data are provided with this paper.

## Code availability

The code is available in Zenodo under accession code https://zenodo.org/records/16042325.

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

## Acknowledgements

This research was funded by NIH grant R01NS139485 to S.Z.

## Author contributions

L.C. and S.Z. conceptualized the study. W.J., L.C., and S.Z. prepared the original draft. W.J., L.C., and S.Z. reviewed and edited the manuscript. W.J. performed data processing and visualization. L.C. and S.Z. supervised the project.

## Competing interests

The authors declare no competing interests.
