## [Transparent Peer Review file · Nature Communications]

The Implications of Alternative Splicing Regulation for Maximum Lifespan

Corresponding Author: Professor Sika Zheng

Version 0:

Reviewer comments:

Reviewer #1

(Remarks to the Author)

In this study, Jiang et al., addressed the question of whether alternative splicing have impacts on the maximum lifespan (MLS) and aging by conducting comparative analysis using public transcriptome data from 26 mammals. AS events showing positive or negative correlation with MLS were identified in each of 6 major tissues, with brain shown a special pattern. Functional enrichment analysis identified several interesting categories including DNA damage checkpoint signaling. Additionally, they also identified human age-associated AS events using the Genotype-Tissue Expression (GTEx) data and found that AS events associated with both MLS and aging are enriched for intrinsically disordered proteins. Finally, they identified a group of RBPs as the upstream regulators of the AS events associated with MLS and aging.

While the work reports several changes related to MLS and lifespan, it fails to provide the importance of these changes or show any biological relevance besides reporting results. In general, more effort should be devoted to putting the results into the larger frame of the biology of aging, avoiding vague and general statements. The results involving intrinsically disordered proteins are interesting and should have been expanded to make the work more meaningful and impactful. What does it actually mean to differentially spliced IDP ? I suggest to the author to re-think the scope of their work from a mere reporting analysis to a more deep and careful implication of the phenomena they describe.

In summary, while the topic of this study is of interest, the question the authors raised was not well addressed, as most of the analyses lack depth and the evidence is not strong; there are some interesting findings, but they are difficult to integrate together to support the conclusions.

My major concerns are as follows:

Major points:

1, A recent study demonstrated that the species with lower effective population size, which usually longer lived, have a higher genome-wide average alternative splicing rate (AS rate), a lower splicing accuracy and less functional alternative splicing, which mainly driven by low abundance isoforms [PMID: 38470242]. This is consistent with the molecular error hypothesis that the gene product diversity, resulting from alternative splicing, alternative transcription initiation, polyadenylation, or RNA editing, often does not correspond to fine-tuned adaptations but simply to the accumulation of errors [PMID: 35641344; 30883542; 29385526; 29886108; 21151575]. Based on this hypothesis, we expect that long-lived species would have lower AS rate in higher expressed genes, while higher AS rate in lower expressed genes, which could result in lower or higher PSI, respectively, in long-lived species. Therefore, at least for AS events that positively correlated with MLS (pos-MLS), the involvement of these genes in MLS regulation is debatable. This is reflected by the one example AS event shown in Figure 4a: apparently larger variation of PSI in longer-lived species, indicating a nature of randomness for this splicing event and therefore higher variation across samples. For AS events that negatively correlated with MLS (neg-MLS), the authors may need to examine the expression level of these genes, as the highly expressed genes are more likely involved in MLS regulation. However, an AS event correlating with MLS does not necessarily mean this AS event is MLS associated. As there are many factors contributing to the variation of PSI among species, we would expect a fraction of AS events showing MLS correlation under random condition with certain probability. Therefore, the authors can only claim a group of AS events (neg-MLS or pos-MLS) to be associated with MLS when functional enrichment are observed, e.g., significantly over-represented in a specific GO term.

2, The authors highlighted the brain specific MLS-associated splicing events and claimed a distinctive role of AS in the brain. Brain is known to express many very long genes [PMID: 25905808]. Are those brain specific MLS-associated splicing events generally longer than others? And could the differential expression for these genes contribute to brain's distinct pattern of association with lifespan? The authors also shown that brain harbors more age-associated splicing events. Are those genes generally longer than genes showing age associated changes in other tissue?

3, The authors tried to identify the RBPs that regulates the MLS associated AS events. A brief introduction in terms of the roles of RBPs in AS regulations should be added before showing the data. When doing the RBP enrichment analysis, the authors only considered the exon (or intron) involved in a AS event. Would it be more reasonable to consider the target exon/intron plus the flanking intronic/exonic sequences? What is the significance of the multiple RBPs regulating multiple MLS or age AS events at the same time? For the Figure 10, even though it should be okay to state that RBP binding motif enrichment is more pronounced with MLS- associated AS events compared to age-related AS events, I would suggest the authors to test the enrichment ratio for MLS and age group. For those with small ratios, especially the enrichment ratio for age AS events, most of them are not significant (which means that RBPs does not play an important role during aging or in the evolution of MLS). I further suggest the authors to examine the RBPs enrichment in MLS AS events in each species, as different species may have evolved different sequences, hence different binding motifs or splicing sites. Moreover, are the expression of the top20 RBPs correlated with MLS or age?

4, Many interventions are known that extend the lifespan of mammals, including drugs, diets, and genetic manipulations. Are those MLS or age AS events changed in mice treated with those interventions?

5, The description of the procedures for identifying the homologous AS events is not sufficiently detailed. For example, what query sequences were used to perform blast against the exon or intron database. Some AS events are quite sensitive to false positives or negatives. How does the authors guarantee that the alternative first/last exons events have not resulted from the incomplete assembly of the transcripts that missing the first/last exon (especially some genes have very short first exon that could be easily missed)? The intron sequences are quite variable across species. For each species, did the authors searched against their own intron sequences or directly against mouse intron sequences? If the later, will the results keep the same if they use each species' own intron sequences as database? How did the authors define the homologous alternative 5'/3' splice sites? Would the alternative 5'/3' splice sites to be the same between species or the case of different splice sites on an orthologous exon is also included? The authors excluded exon that with alignment rate less than 80%. Is it possible these less aligned exons had resulted from the alternative 5'/3' splicing? I would suggest the authors to describe the procedures clearer and if possible, perform additional filtering steps to improve the accuracy of the AS identification.

Minor points:

1. the p-values should be shown in scatter plots (Fig.4a,4b); to address the multiple testing issue, the adjusted p-value should be provided, for example, in the GSEA enrichment analysis.
2. are all AS events profile identified in this study clustered by species or by organ? The authors may have to show the heatmap of the correlation coefficient matrix to make sure the pipeline works well.
3. for the GSEA dot plots, why did the authors show the gene ratio and counts instead of showing the NES?
4. "Several terms related to metabolic processes were enriched in MLS- [...]. Metabolic changes are hallmarks of aging." In general, the authors should be more specific in their statements. What do they mean by "metabolic changes"? Which processes and pathways are involved? The term "metabolic changes" is too generic and does not help the reader make sense of the results.
5. "A longevity-regulated pathway"—again, this sentence is too generic. To which pathways are the authors referring?
6. "The union of age-associated AS events across all sub-tissues"—This statement, while reflecting an observation from the authors, does not seem relevant or justified, as the authors analyze a substantially higher number of tissues from different brain regions compared to other tissues. The higher number could simply be the result of the increased number of tissues compared.
7. The species considered share an evolutionary history, so the data points used to calculate Spearman's and Pearson's correlations are not independent. A more appropriate approach would be to use phylogenetic least squares regression analysis to account for the natural relationships that organisms share based on their evolutionary history. The authors should perform their analysis using this correction, which would strengthen their results on AS sites and MLS.

Reviewer #2

(Remarks to the Author)

In this paper, the authors analyzed six tissues from 26 non-human mammalian species with varied lifespans to identify hundreds of MLS-associated alternative splicing (AS) events. They find that 50% of these events were shared across tissues, but the brain had twice as many tissue-specific MLS associations. They also identify body-mass-associated AS events and address whether they overlap with MLS-AS events. Finally, they explore age- AS in humans using the GTEx database.

Addressing how alternative splicing associates with maximum lifespan is an interesting question. The paper is well written and easy to follow. However, we find that their methodology could be improved and that some of their conclusions are based on weak evidence. There are many factors that could be confounding their findings and I find the authors do not control for some of them. In several cases, their result should be compared with previous literature. For example, to show that their

method to find AS-MLS associated events works properly it would have been interesting to see that they find what has been previously shown. Finally, the association between AS and aging using GTEx has been addressed in other studies and may not be that novel. Below, we give more specific information about our concerns in their methodology and findings.

There is no code to reproduce the results.

The authors should compare their pipeline to find homologous alternative exons across species with previous studies. I wonder if the amount of conserved AS events they detect is what is expected. Do they detect conserved AS that have been described before?. Finding homologous AS events is a fundamental step in their study. The only "validation" they provide is the statement that "the number of homologous events correlated with the evolutionary distance from mice, spanning from 900 to over 3700". They should statistically test this by performing a proper correlation between evolutionary distance and homologous AS events. Sorting the species in plot 2a by evolutionary distance would also help see the mentioned correlation. Also, this would be an expected result for blasting any type of sequence, not only splicing junctions. So further validations/comparisons with previous work is needed to show that their pipeline finds true homologous AS events. Also, authors should say if the finding that exon skipping events are the most conserved and AF and AL the least is expected based on previous work. It would be interesting to assess if any AS type is more conserved than expected (e.g MX are the least common but seem to be very conserved). Papers from the Kaessmann lab and Blencowe lab among others have relevant work in this regard.

plot in 2c) would benefit to show conservation normalized by the total number of events detected, otherwise it is hard to say whether events are more conserved or less than expected.

If BM and MLF are correlated, they should test both in the same linear model rather than doing a spearman correlation with each and then looking at the overlap. However, if they are extremely correlated (> 0.8) it may actually not even be possible to distinguish between the two. Importantly, modelling PSI with linear models is not straight forward because the distribution of PSI values is between 0 and 1 with most values being very close to zero and one. The authors could perhaps look at how other studies have addressed this. For example, in Garcia-Pérez et al. Cell Genomics 2024 (PMID: 36777183). The number of associated AS events with BM and MLS should be obtained in the same model to identify MLS or BM specific events. In addition, the observation that the brain behaves differently should be tested explicitly (and not just showing that the overlap is smaller, as this could be due to stochastic processes). Finally, the authors should also explain in the methods how the jaccard index shown in Fig 3 was obtained.

The statement "Intriguingly, the brain distinction from other tissues was not observed in the MLS- and BM-association study with gene expression (Figs. 3e and 3f), indicating potential divergence in the regulatory functions of AS and gene expression in brain aging." What gene expression analysis is it referring to? How do they quantify gene expression, I guess adding up the transcript expression values but it is not explained in the methods.

The fact that the authors find the brain having the largest number of AS-associated MLS events in the brain should be put in context with the fact that AS patterns in the brain are also the most distinct compared to other tissues and cite the corresponding literature. I wonder how much of the signal they see could be explained by this fact alone.

Regarding the functional enrichment analysis on the AS-MLS genes. It would have been perhaps interesting to compare whether the enrichments are different between those AS-MLS and differential gene expression-MLS genes previously described using the same dataset? Also, the authors don't specify whether the pathways are enriched among positively or negatively correlated genes with MLS, which is important for their interpretation.

The authors don't correct for some of the technical covariates routinely included in previous GTEx publications (Taru Tukiainen et al., 2018 Nature PMID: 29022598; Pedro G. Ferreira et al., 2018 Nat. comms. PMID: 29440659; Garcia-Pérez et al. Cell Genomics 2023 PMID: 36777183), like Ischemic time and Hardy Scale. They say they use the covariates used in sQTL calling but they do not cite any paper but in previous GTEx publications, technical covariates are always included in the analysis. Thus, the authors should correct for them or demonstrate that the additional included PEER factors are correcting for those technical covariates.

In the RBP enrichment analysis they don't correct for expression levels to control for highly expressed genes. Highly expressed genes would be expected to have more RBP binding sites and could therefore be confounding the results.

Analyzing AS change with age using the GTEx is not novel and it has been done before. For example, Garcia-Pérez et al. Cell Genomics 2023 (PMID: 36777183). It would be nice if the authors compared their findings with previous literature to show that their elastic net regression method works. The authors could also explain the differences between their method and the recently published one, and why this new methodology is better suited for the aim of the paper.

The authors use elastic net regression to identify age-associated splicing events. To show that this method is well calibrated, I would strongly encourage the authors to run the method permuting the labels to show that there are no false positives and the method is well calibrated.

Minor comments::

version of Stringtie (for the alternative splicing quantification)? What is the version of GENCODE used? Where are the reference genomes downloaded from? Where does the MLS information come from?

DAVID is not maintained since a long time ago. Another functional enrichment method should be used. Additionally, the background used for the GO analysis should be specified.

cell type composition is a known confounder on transcriptomic analysis, the authors should maybe explain how this may influence their results.

In the RBP enrichment analysis, the authors use 10 random groups as background. Why not using the whole set of tested AS events and compute a fisher/chi-square test?

The authors state that 37% of AS events are associated with MLS and the majority are exon-skipping. It would be interesting to know if any AS type is more associated with MLS than expected. Also, is this 37% different from the proportion of genes transcriptionally associated with MLS found in the previous study using the same dataset?

When testing multiple hypergeometric tests (Figure 3a-3b), a correction for multiple testing should be used on the p.values.

In Figure 4 it would be useful to see the structure of the gene and highlight the isoforms involved in the AS event. Also indicating the FDR of both correlations would be helpful for the reader.

Is the overlap between age and MLS-associated AS events higher than expected?

Figure 10 shows only the 20 overall enriched RBPs. A supplementary table with all the enrichments, overall, and tissue-specific, should be included.

It may be better not to use abbreviations without introduction in the abstract (RBP).

Reviewer #3

(Remarks to the Author)

Reviewer #4

(Remarks to the Author)

This paper by Jiang et al. investigated the relationship between alternative splicing regulation and longevity and ageing using public RNA-seq data of six tissues. Longevity-dependent splicing was examined in 26 mammalian species (Rodentia/Euarchonta) and ageing-dependent splicing was analyzed in human tissues using GTEx data. In each tissue, hundreds of alternative splicing events were found to be correlated with maximum life span (MLS) and/or age and some of them are shared across tissues. Compared to the other tissues, the brain has the most tissue-specific MLS-associated AS events and the least overlap with body-mass-associated AS events. MLS-associated AS events are enriched in genes related to certain pathways such as metabolism, DNA damage checkpoint signaling, and mRNA surveillance. Overlapping human-age-associated AS events with nonhuman MLS-associated events are enriched in genes encoding intrinsically disordered proteins. Finally, RBP motifs enriched in MLS- and age-associated AS events were identified, and more enrichment was found in MLS-associated events. As the relationship between lifespan and AS has been understudied previously, this study provided interesting insights and the list of MLS-associated AS events can also be valuable to the research community. There are also some technical innovations in the study, such as the elastic net regression method used to identify age-related AS events. Overall, the study is well performed. Some parts can be improved to provide additional support for the reliability of the analysis, as summarized below.

1. Introduction – “The roles of pre-mRNA alternative splicing (AS) in MLS have yet to be explored.” This claim is not exactly true (e.g., see ref. 15 for a review and PMID 27363602).
2. What is the range of ages for the nonhuman and human GTEx samples analyzed in the study? How the age affects identification of longevity and age-associated AS events? e.g., are there aging samples in the non-human data that can complicate the identification of MLS-associated events? are there young individual samples in the GTEx data so that some of the age-associated events reflect development instead of aging?
3. Quantification of AS events. SUPPA was employed to quantify the splicing level of each AS event in terms of percent spliced-in (PSI), utilizing transcript-level expression data. It will be helpful to provide additional details, how PSIs were calculated (e.g., whether it is based on junction reads only or both junction and exonic reads; how transcript-level expression data was utilized and whether this is affected by the accuracy of transcript assembly).
4. The authors built a database of mouse alternative exon/intron sequences against sequences of the other mammalian species. What is exactly the sequence of the other species (i.e., pre-mRNA of the whole gene?). Very few alternative first/last exons were found to be conserved across species. Is this because of technical issues (e.g., longer sequences were used for BLAST search)? The authors have used an 80% blast match to identify conserved alternatively spliced exons. This high stringency yields deeply conserved exons. Nevertheless, exons with vast wobble nucleotide divergence, but conserved amino acid sequences, or UTRs, may be filtered out. Figure 2b for example depicts filtering out of virtually all ALE and AFE,

most likely since these exons contain UTRs which have limited sequence constraints. It will be useful to understand the amino acid conservation of exons that have been included in the analyses, as well as those that have been excluded. In addition, it will be worthwhile to repeat this correlation analyses using exons selected based on 80% (or more suitable) amino acid conservation. This will perhaps expand the set of exons considered for MLS-association and could improve the overall significance of the findings as well.

5. The authors have utilized the 26 species data from Lu et al (2022, Cell Metabolism), a rich resource for studying the connection between AS and MLS. They find a set of AS events that show high correlation between MLS and AS. However, like Lu et al, they find that there is substantial overlap with BM-MLS targets. In Lu et al, the authors were able to apply a BM-correction to their analyses to remove BM-association as a confounder. As there is a clear overlap of genes showing association between AS-MLS and BM, a similar analysis should be conducted to identify BM-corrected MLS-AS genes. Similarly, the authors should consider if there are overlaps between gene influencing MLS and AS genes influencing MLS and seek to remove gene expression as a confounder in their analyses.

6. Also, several previous studies looked for longevity and/or age-related alternative splicing, e.g. PMID: 27363602, 23340839. Are AS-MLS genes previously curated in other collections of aging genes?

7. It will be useful to know if there are multiple AS-MLS exons per gene, and if these exist, whether they have consistent association with MLS or discordant...

8. Given the critical importance of splicing quantification, it will be very helpful to include additional computational and/or experimental validation to confirm the quantifications are reliable, and not complicated by technical issues such as RNA quality used for library preparation and mapping errors. For example, is it possible to validate using RT-PCR for the two examples provided in Fig. 4 (even with human/mouse RNA if the materials are difficult to obtain for the other species)? It will be helpful to depict these examples in the form of syntenic coverage plots as well.

9. Figure 6, the identification of AS events showing the opposite effect on MLS in brain vs other tissues is intriguing, but it is difficult even to imagine how this could happen. Can the authors speculate? The significance should be explored in detail.

10. Lu et al note that the key pathways enriched in MLS-associated genes are energy metabolism, inflammation, circadian genes, DNA damage regulation, immune responses, and RNA biology. A comparison should be performed to understand the overlap in enriched between AS targets and Lu et al gene targets. It is quite interesting that AS genes enrich distinct sets of pathways, can the authors detail whether certain pathways might contribute to MLS via AS, whereas others through expression changes?

11. Figure 9 shows that human-age related AS events involve IDPs. Further information should be provided about the nature of these AS events. Are the alternative exons coding for the IDP regions? What pathways are these genes involved in?

12. The RBP analyses seem interesting, but quite premature as it stands. It seems that only the alternative exons were used for motif analysis. It will be helpful to also examine flanking intronic sequences contributing to AS (e.g., upstream and downstream flanking intronic sequences for skipped exons). How the enrichment of the motif patterns is consistent with the direction of MLS or age-associated splicing and RBP expression in different samples to get a sense whether any of the candidate RBPs is positively or negatively associated with the phenotype? i.e. Is there a relationship between RBP motif and AS-target PSI across the 26 species? That seems a critical aspect to understand RBP regulation. Even in the absence of high-quality genomes, this analysis can be performed for the exonic sequences at least. Furthermore, the authors suggest that RBPs identified influence lifespan: are these genes identified among those in Lu et al?

13. Elastic net model – regularization terms are not specified in the description.

14. The manuscript has 11 figures – some (e.g., figures 1 and 2) might be moved supplemental materials.

15. Fig. 7 – explain gene ratio.

16. The mammalian order Eulipotyphla is repeatedly misspelled in the text

Reviewer #5

(Remarks to the Author)

Version 1:

Reviewer comments:

Reviewer #2

(Remarks to the Author)

The authors made a substantial effort in replying to all of our comments. However, I still have a two minor concerns that were not fully addressed in their rebuttal:

I have methodological concerns in using functional enrichment analysis and using genome wide as background. Using genome-wide background likely is in most cases incorrect because the gene set was detected in an analysis where most likely not all genes were tested i.e. in when doing differential alternative splicing in a particular tissue, not all genes are tested, only those expressed in that particular tissue and often there are other filters. So if we want to test whether the differentially alternatively spliced genes are enriched in a function this should be compared to the genes we initially tested as differentially spliced and not all genes. In this paper some of this issues are explained:

<https://pmc.ncbi.nlm.nih.gov/articles/PMC8936487/>

This applies to all functional enrichments throughout the paper and the backgrounds should be specified in methods.

Having 100 false positive genes in the permutation analysis is a bit concerning to me. The authors claim that their results hold because they identify more genes (400) and those are associated with aging in their functional enrichment analysis. I am aware the authors can not filter by p-value but it would be good if there is anything else the authors could do to reduce false positives?

(Remarks on code availability)

Reviewer #4

(Remarks to the Author)

I thank the authors for making efforts to address most of my critiques. I do not have any new comments.

(Remarks on code availability)

Reviewer #5

(Remarks to the Author)

(Remarks on code availability)

This is not in my wheelhouse of expertise. I am unable to judge if complete.

Version 2:

Reviewer comments:

Reviewer #2

(Remarks to the Author)

the authors have addressed my concerns. I only have a minor comment. The authors should show the decrease of the performance in FDR when increasing the threshold in the manuscript rather than just to me.

(Remarks on code availability)

We are grateful for the reviewers' recognition of the novelty and significance of our findings, with comments such as, "As the relationship between lifespan and AS has been understudied previously, this study provided interesting insights and the list of MLS-associated AS events can also be valuable to the research community", "Addressing how alternative splicing associates with maximum lifespan is an interesting question. The paper is well written and easy to follow", "There are also some technical innovations in the study, such as the elastic net regression method", and "the study is well performed". The insightful suggestions raised by the reviewers are indicative of how carefully they considered the presented data, and we are grateful for the time and effort this took. We apologize for the extended revision period beyond the requested three months due to an unforeseen circumstance that is unrelated to the manuscript itself. We are pleased to report that we have meticulously made extensive revisions, included new figures, and updated existing figures to ensure all the reviewers' concerns are addressed and the revised version provides a clear representation of the data. Their insightful remarks and constructive suggestions have significantly enhanced the quality of our work.

A detailed, point-by-point response to each comment (Reviewers #1, 2, and 4) is provided below. Reviewer #3 and Reviewer #5 provided no additional comments. To reference the revised manuscript, we quote passages (in blue font) from the main text in the response letter wherever appropriate. We hope the reviewers find our revised manuscript appropriate for publication. Revisions in the manuscript are also highlighted in blue font. We thank the reviewers once again for their thoughtful feedback and the opportunity to improve our work.

Reviewer 1 Major Points

1. A recent study demonstrated that the species with lower effective population size, which usually longer lived, have a higher genome-wide average alternative splicing rate (AS rate), a lower splicing accuracy and less functional alternative splicing, which mainly driven by low abundance isoforms [PMID: 38470242]. This is consistent with the molecular error hypothesis that the gene product diversity, resulting from alternative splicing, alternative transcription initiation, polyadenylation, or RNA editing, often does not correspond to fine-tuned adaptations but simply to the accumulation of errors [PMID: 35641344; 30883542; 29385526; 29886108; 21151575]. Based on this hypothesis, we expect that long-lived species would have lower AS rate in higher expressed genes, while higher AS rate in lower expressed genes, which could result in lower or higher PSI, respectively, in long-lived species. Therefore, at least for AS events that positively correlated with MLS (pos-MLS), the involvement of these genes in MLS regulation is debatable. This is reflected by the one example AS event shown in Figure 4a: apparently larger variation of PSI in longer-lived species, indicating a nature of randomness for this splicing event and therefore higher variation across samples. For AS events that negatively correlated with MLS (neg-MLS), the authors may need to examine the expression level of these genes, as the highly expressed genes are more likely involved in MLS regulation. However, an AS event correlating with MLS does not necessarily mean this AS event is MLS associated. As there are many factors contributing to the variation of PSI among species, we would expect a fraction of AS events showing MLS correlation under random condition with certain probability. Therefore, the authors can only claim a group of AS events (neg-MLS or pos-MLS) to be associated with MLS when functional enrichment are observed, e.g., significantly over-represented in a specific GO term.

Reply:

Thank you for the thoughtful suggestion. In response, we have cited these works and added a new Result section "Gene expression levels of MLS-associated alternative splicing events"

(page 11–13) addressing this point. Specifically, we quantified the expression levels of genes containing MLS-correlated AS events and examined “if AS events negatively correlated with maximum lifespan (neg-MLS) are higher expressed genes whereas those positively correlated (pos-MLS) are lower expressed genes. The data does not support a universal trend of higher expression in neg-MLS genes or lower expression in pos-MLS genes (Supplementary Fig. 5). The overall gene expression distributions between these two groups were very similar in each tissue. In some tissues, Mann-Whitney U-tests comparing group means revealed slight differences, but the direction of these differences varied. Neg-MLS genes exhibited higher mean expression than pos-MLS genes in the brain and skin ($p < 0.001$) but lower mean expression ($p < 0.05$) in the heart, kidney, and lung. In the liver, there was no significant difference ($p = 0.53$). Further analysis using QQ plots demonstrated these mixed trends among tissues. The heart, kidney, and brain showed a higher proportion of highly expressed genes among neg-MLS events (as illustrated in the QQ plots in Supplementary Fig. 5), while lung, skin, and liver showed the opposite trend for highly expressed genes. Overall, no consistent pattern emerged regarding the gene expression levels of MLS-associated splicing events. While some neg-MLS AS events tend to have higher gene expression, particularly in the brain, this is not consistent across tissues or a widespread pattern. Thus, the relationships between alternative splicing and MLS regulation are largely independent of gene expression levels and the identified MLS-AS events are not random noise.”

There are multiple explanations to reconcile these observations. First, the correlation between effective population size and longevity is not very strong, as mentioned by the authors, citing their original statement “We note however that these correlations are not very strong. It thus seems likely that none of these proxies provides a perfect estimate of N_e .” PMID: 38470242). This weak correlation can jeopardize inferring further correlations. Second, the authors employed an intron-centric analysis, which intrinsically emphasizes splicing infidelity and identification of low-abundant isoforms as evidenced by the generally low AS rates reported in the study. This approach focused on deviation from constitutive splicing (i.e., basal alternative splicing rate is zero). In contrast, we used an exon-centric analysis with stringent criteria for evolutionary conservation. Our approach focused on conserved alternative splice sites among species (regardless of basal alternative splicing rate). Third, the authors based on the intron splicing status of major isoforms to determine alternative splicing rates and concluded that the splicing rate of rare splicing variants is negatively correlated with gene expression levels. The authors also stated that “this negative correlation between AS rate and gene expression level is not expected for functional SVs”. Fourth, the molecular error theory can also probably be applied to short-lived species. In summary, in the revised manuscript we cite and acknowledge the relevance of the molecular error hypothesis, which posits that increased AS in long-lived species may reflect error accumulation rather than adaptive mechanisms.

Regarding the example in Figure 4a, there is no correlation between *Fn1* gene expression and MLS (as shown to the right).

We agree with the reviewer that correlation does not imply causation. The references mentioned by the Reviewer and most comparative transcriptomic studies, including ours, uncover correlation signals for functional inference. Nevertheless, we emphasize that our current analysis already applies stringent thresholds (Spearman correlation $\rho > 0.4$, FDR < 0.05) to reduce false positives. We appreciate Reviewer's suggestion to further assess functional enrichment and add a new Result section "**Functional enrichment of MLS-associated alternative splicing**" (page 14-17). Furthermore, we analyzed pathway enrichment for genes with MLS-associated AS events versus those with MLS-associated expression changes (as identified by Lu et al.) and found minimal overlap. This limited intersection implies that alternative splicing modulation acts as an independent regulatory layer to lifespan regulation. Importantly, MLS-associated AS events and MLS-associated genes show distinct functional pathways, revealing novel biological insights.

"To understand the biological significance of MLS-associated AS (MLS-AS) events, we conducted functional enrichment analysis using Enrichr. We examined genes harboring MLS-associated splicing events for enrichment in the Gene Ontology (GO) biological process terms and KEGG pathways. MLS-AS genes are significantly enriched in functional categories of mRNA processing, stress response, neuronal functions, and epigenetic regulation (Supplementary Table 3). Figure 6 summarizes the significantly enriched terms (FDR < 0.15), organized by functional categories.

Multiple enriched pathways involve cellular stress response, including Stress Granule Assembly, Regulation of Defense Response to Virus, MAPK Cascade, *Positive Regulation of PI3K-AKT Signal Transduction*, etc. stress response plays an essential role in safeguarding cellular and organismal integrity [44-46]. When cells encounter stressors such as oxidative damage, heat shock, viral infection, or metabolic fluctuations, they initiate intricate signaling pathways to mitigate harm and restore homeostasis. Efficient stress response mechanisms enable cells to quickly detect and neutralize damage, activate repair systems, and, if necessary, trigger programmed cell death (apoptosis) to remove irreparably compromised cells. Indeed, Regulation of Apoptotic Process (and related GO terms) is significantly enriched (Figure 6). Apoptosis ensures that tissues maintain their functional capacity and that damaged cells do not accumulate [47, 48]. Moreover, the adaptability of the stress response is crucial for long-lived species. Organisms with robust stress response networks can better withstand environmental and physiological fluctuations across their lifespan, delaying the onset of cellular dysfunction and chronic pathologies. Alternative splicing of stress response genes likely modulates organisms' competence for the efficiency and adaptability of stress response.

Notably, mRNA processing steps and regulatory mechanisms are enriched, including spliceosomal complex assembly (along with many other splicing-related terms), cleavage and polyadenylation, regulation of translation, mRNA surveillance, and RNA transport. Collectively, these comprise 18% of all enriched GO terms and encompass nearly every stage of the mRNA life cycle. This underscores the importance of maintaining transcriptome integrity and proper mRNA processing for longevity. Errors or inefficiencies in any of these processes can result in the production of faulty or non-functional proteins, accumulation of aberrant RNA species, and ultimately, cellular dysfunction. For long-lived species, the ability to maintain precise and adaptable RNA processing systems is particularly critical. Accurate splicing ensures that essential gene products are expressed in correct proportions and forms, while alternative splicing provides an additional layer of regulatory flexibility, allowing organisms to rapidly adapt to changing physiological conditions without altering their genomic DNA. mRNA surveillance mechanisms, such as nonsense-mediated decay, also respond to cellular stress and safeguard the transcriptome by identifying and eliminating defective mRNAs that could otherwise give rise to toxic or deleterious proteins [49]. Thus, robust RNA processing and regulation are central to maintaining cellular homeostasis and adaptability, both of which are essential for healthy aging. Their enrichment suggests that longevity in some species may be achieved, at least in part, by

optimizing alternative splicing mechanisms that control the transcriptome's quality, diversity, and responsiveness throughout the lifespan.

Additionally, we observed a significant enrichment of biological processes related to neuronal functions, such as Regulation of Neuronal Synaptic Plasticity, Axon Guidance, Synapse Organization, Brain Development, Neuron Migration, Long-term potentiation, Neurotrophin Signaling Pathway, Neurotransmitter Secretion, etc. Neuro-related terms represent over 22% of all enriched GO terms, further reinforcing the distinctive role of brain-specific [50-59]. The pivotal role of neuronal function in determining organismal lifespan likely arises from the brain's central control over homeostasis and its orchestration of numerous physiological processes intimately connected to aging and longevity. Learning, memory, and adaptability of neural circuits are intricately regulated by pathways involved in axon guidance, synaptic organization, and neurotrophin signaling. Through these mechanisms, neurons can adapt to developmental cues and environmental stressors. Importantly, the brain influences systemic aging via neuroendocrine and autonomic pathways. For example, the hypothalamus integrates cues related to energy balance, stress, and circadian rhythms, coordinating hormonal outputs that impact metabolism, immune responses, and tissue regeneration. In this way, neuronal function serves as a central determinant of lifespan—supporting cognitive maintenance, regulating systemic physiology, modulating immune activity, and conferring resilience to stress. In summary, these findings highlight the diverse biological processes modulated by alternative splicing to influence maximum lifespan.

To compare the respective contributions of gene-level and splicing-level regulation to lifespan, we compared the functional enrichment of genes with MLS-associated AS events and those with MLS-associated expression changes (as identified by Lu et al.). First, Among the 576 genes with MLS-associated AS events (MLS-AS) and the 6,952 genes with MLS-associated expression changes (MLS-gene), only 211 genes overlapped. Using a background of 17,511 multi-exon genes, this overlap yielded a p-value of 0.94 (Fisher's Exact test), indicating it is not statistically significant.

Comparing their pathway enrichments reveals clear distinction. MLS-genes are enriched in GO group terms such as DNA replication, mitotic recombination, chromosome segregation, double-strand break repair, protein folding, mitochondrion organization, cilium movement, and lymphocyte chemotaxis. They are also particularly enriched in various macromolecular metabolism pathways, including acylglycerol metabolic process, amino acid catabolic process, carbohydrate biosynthetic process, ribonucleoside biphosphate biosynthetic process, hydrogen peroxide catabolic process. These findings align with current theories that emphasize DNA repair [60] and metabolic control in longevity. Proper regulation of macromolecule metabolism is particularly crucial for maintaining cell structure and function, and repairing damaged tissues [61, 62]. These GO terms or functional groups are not prominent in MLS-AS genes. Overall, only seven out of 265 GO terms from MLS-AS events are enriched in MLS-gene groups (FDR <0.15 for both groups). When the threshold is tightened to FDR < 0.1, this overlap drops to just one exact shared term, and at FDR < 0.05, there are no overlapping GO terms between the two groups. We therefore used FDR <0.15 for comparisons. Since different GO terms can still be related under one functional activity, we used REVIGO to organize the GO terms into functional categories. The results still highlighted a clear distinction. For example, among the 424 total GO terms enriched in MLS-genes, only three were related to neural function (0.7%) and ten to RNA-related processes (2.3%). Similarly for MLS-AS genes, DNA Repair is only minorly represented with two specific terms, "Regulation of Nucleotide-Excision Repair" (FDR: 0.056) and "Positive Regulation of DNA Repair" (FDR: 0.082). This limited intersection underscores that alternative splicing modulation acts as an independent regulatory layer to gene expression regulation."

In summary, thanks to reviewers' suggestions, we have significantly revised the manuscript with a novel scope and incorporated in-depth analyses to provide insights into the relationship between splicing regulation and MLS as a species trait:

- MLS associations are independent of gene expression levels and the associated AS events are largely distinct from genes exhibiting expression association with MLS, showing that alternative splicing captures additional lifespan-related signals beyond gene expression.
- The MLS-associated AS events are enriched in pathways related to mRNA processing, neuronal functions, and stress response. There are different from the pathways enriched with MLS-associated gene expression changes: DNA replication, protein folding, mitochondrion organization, cilium movement, lymphocyte chemotaxis, and various biosynthetic and catabolic processes.
- Most tissues have a similar number of total MLS-associated AS events, with ~50% shared among tissues. However, the brain stands out by having twice as many tissue-specific MLS associations compared to other tissues, suggesting unique splicing programs underlying neural contributions to longevity.
- Many AS events associated with MLS overlap with those associated with body mass in peripheral tissues, but this overlap is reduced by half in the brain. In contrast, gene expression associations with MLS and body mass show similar degrees of gene overlap across all tissues. This implies brain-specific splicing regulatory mechanisms in lifespan determination.
- While MLS-associated AS events have limited overlap with age-associated AS events, the shared events are disproportionately enriched in intrinsically disordered protein (IDP) regions, implicating splicing in modulating protein flexibility and stress adaptability. These IDP genes are enriched in RNA processing-related pathways, suggesting a feedback mechanism in which splicing factors regulated via AS in IDP regions are linked to lifespan regulation.
- MLS-associated AS events show stronger RNA-binding protein (RBP) motif coordination than age-associated ones, supporting a more genetically programmed adaptation for lifespan determination and contrasting with the more variable splicing changes linked to chronological aging.

2. The authors highlighted the brain specific MLS-associated splicing events and claimed a distinctive role of AS in the brain. Brain is known to express many very long genes [PMID: 25905808]. Are those brain specific MLS-associated splicing events generally longer than others? And could the differential expression for these genes contribute to brain's distinct pattern of association with lifespan? The authors also shown that brain harbors more age-associated splicing events. Are those genes generally longer than genes showing age associated changes in other tissue?

Reply:

We appreciate the reviewer's insightful suggestion to examine whether brain-specific MLS- and age-associated splicing events are linked to longer genes, given the known enrichment of long genes in the brain.

To investigate this, we compared the gene lengths of brain-specific MLS- and age-associated splicing events to those from other tissues. Statistical testing (t-test) revealed no significant differences in gene length (MLS: $p = 0.803$; age: $p = 0.515$), suggesting that brain-specific splicing events are not generally associated with longer genes.

We also evaluated whether these splicing events are more prevalent in brain-specific genes, defined based on tissue-specific gene expression profiles. While the number of overlapping genes was generally high in the brain (MLS: 11 overlaps; age: 38), this is likely due to the larger number

of brain-specific genes overall, rather than a specific enrichment. Fisher's exact test did not indicate significant enrichment (MLS: $p = 0.49$; age: $p = 0.1$).

Additionally, we examined the PSI values of brain-specific MLS-associated AS events across five non-brain tissues. We found that only 3.84 to 7.85 % of PSI values in these tissues were zero, indicating that most of these events are not strictly brain-exclusive. Rather, they are broadly expressed across tissues but show significant associations with MLS only in the brain. This supports the idea that the observed brain specificity arises from functional relevance rather than being solely the result of restricted splicing.

These findings have been incorporated into the revised manuscript (page 24–25) under a new section “**Brain-specific splicing associated with MLS**”. We thank the reviewer for raising this point, which helped us more clearly delineate the distinctiveness of brain-associated splicing in the context of gene length and tissue specificity.

3. The authors tried to identify the RBPs that regulates the MLS associated AS events. A brief introduction in terms of the roles of RBPs in AS regulations should be added before showing the data. When doing the RBP enrichment analysis, the authors only considered the exon (or intron) involved in a AS event. Would it be more reasonable to consider the target exon/intron plus the flanking intronic/exonic sequences? What is the significance of the multiple RBPs regulating multiple MLS or age AS events at the same time? For the Figure 10, even though it should be okay to state that RBP binding motif enrichment is more pronounced with MLS-associated AS events compared to age-related AS events, I would suggest the authors to test the enrichment ratio for MLS and age group. For those with small ratios, especially the enrichment ratio for age AS events, most of them are not significant (which means that RBPs does not play an important role during aging or in the evolution of MLS). I further suggest the authors to examine the RBPs enrichment in MLS AS events in each species, as different species may have evolved different sequences, hence different binding motifs or splicing sites. Moreover, are the expression of the top20 RBPs correlated with MLS or age?

Reply:

We thank the reviewer for these valuable suggestions, which have led to several important additions and clarifications in the revised manuscript.

First, we have added a brief introduction outlining the role of RNA-binding proteins (RBPs) in regulating alternative splicing (AS), particularly their binding to specific RNA motifs that can enhance or repress splice site recognition. This provides necessary context before presenting our enrichment analysis results.

“To further understand the regulation of alternative splicing (AS) events associated with maximum lifespan (MLS) and age, we analyzed RNA-binding protein (RBP) motifs near the associated exons. RBPs play a key role in regulating alternative splicing by binding specific RNA sequences and influencing splice site selection [70]. By interacting with components of the spliceosome or other regulatory factors, RBPs can enhance or repress exon splicing. This regulatory mechanism is crucial for various biological processes, including development, cellular homeostasis, and response to environmental cues [71].” (page 21)

The simultaneous regulation of multiple MLS or age-associated alternative splicing events by multiple RBPs suggests a highly coordinated system-level mechanism and a modular layer of post-transcriptional control separated from transcriptional regulation. This pattern implies that RBPs do not act in isolation or randomly but rather as part of regulatory networks that fine-tune splicing programs in a context-dependent manner. Each RBP can target multiple splicing targets,

and each alternative splicing events are targeted by different RBPs thanks to multiple cis-regulatory elements. Such coordination is particularly significant in the brain, where precise splicing is essential for maintaining complex functions in post-mitotic neurons, which live through the whole lifespan. Moreover, the fact that these RBPs target distinct sets of AS events associated with either MLS or aging indicates that splicing regulation may contribute to the divergence between lifespan determination and chronological aging. This supports the idea that RBPs help orchestrate specialized gene expression programs that are evolutionarily conserved and biologically distinct.

Regarding the RBP binding site analysis, we appreciate the reviewer's point about including flanking intronic and exonic regions, as RBP binding often occurs near splice junctions. In our initial analysis, we focused on the core exonic or intronic regions directly involved in each AS event. In the revised version, we have expanded the analysis to incorporate flanking sequences (± 100 nt for introns, ± 50 nt for exons). This broader scope has increased the sensitivity of motif detection and identified additional RBPs potentially involved in AS regulation.

"The exonic and (or) intronic sequences and flanking sequences (± 100 nucleotides for exon and ± 50 nucleotides for intron) used in the homologous AS identification were provided to RBPmap [27], a web-based tool for mapping binding sites of RNA-binding proteins (RBPs). Default parameters and consensus RNA binding motifs for both mice (for MLS-AS) and humans (for AGEAS) were applied in the analysis. For each RBP, we counted the motif occurrences and normalized them based on the region length for each AS event." (page 7)

In response to the suggestion to compare enrichment between MLS- and age-associated AS events, we computed enrichment ratios for both groups and performed statistical testing. A paired t-test comparing enrichment ratios of matched RBPs revealed a significant difference ($p = 0.00033$), confirming that RBP motif enrichment is significantly stronger in MLS-associated AS events. These updates are reflected in the revised manuscript on page 21.

We also appreciate the suggestion to examine species-specific RBP enrichment. While we agree that this is a valuable direction, current limitations in species-specific RBP motif annotation, especially outside of human and mouse, pose significant challenges. Using human or mouse motifs across other species could introduce biases due to sequence divergence and species-specific splicing regulation. We acknowledge this in a new Discussion section "**Limitations of the study**" (page 28) and highlight it as a promising area for future research, as more high-quality cross-species RBP data become available.

"While our analysis focused on RBP binding motifs characterized in humans and mice, we acknowledge that different species may have evolved distinct splicing regulatory sequences and RBP binding preferences. The current lack of high-quality RBP motif databases for non-model organisms limits our ability to perform species-specific RBP enrichment. As more RBP binding data become available for a broader range of species, future studies can explore the evolutionary dynamics of RBP-mediated splicing regulation and its contribution to interspecies differences in maximum lifespan."

Finally, to evaluate whether RBP expression levels are associated with MLS or age, we examined the overlap between the top 20 enriched RBPs identified in our motif analysis and the set of genes previously reported by Lu et al. as significantly correlated with MLS. The five overlapped ones out of 39 genes examined may contribute to lifespan regulation both through changes in their own expression and through their downstream effects on splicing regulation. This new analysis is included in page 22.

"To investigate whether expression levels of these RBPs are linked to lifespan, we assessed the correlation between their expression and MLS across six tissues. Of the 39 unique

key RBPs (the union of the top 20 motif-enriched RBPs from each tissue), only five (*ESRP1*, *FUS*, *MBNL1*, *PCBP2*, *RBM8A*) overlapped with a previously curated list of genes whose expression correlates with MLS⁹. This limited overlap suggests that analyzing splicing regulation can identify a distinct set of potential lifespan regulators missed by studies focusing on gene expression alone. This reinforces the biological relevance of RBP-mediated splicing regulation in mammalian longevity.”

4. Many interventions are known that extend the lifespan of mammals, including drugs, diets, and genetic manipulations. Are those MLS or age AS events changed in mice treated with those interventions?

Reply:

We appreciate the reviewer’s insightful question. To address this, we expanded our analysis to include a dataset involving a known lifespan-extending intervention: inhibition of PAPP-A (Pregnancy-Associated Plasma Protein A), which has been shown to increase lifespan in mice. We examined whether this intervention affected the inclusion levels of AS events identified as MLS-associated. Our analysis revealed that PAPP-A inhibition significantly impacted AS events positively correlated with MLS, suggesting that such interventions can modulate splicing patterns linked to lifespan regulation. This finding provides preliminary evidence that MLS-associated AS events are responsive to pro-longevity treatments. This is presented in a new Result section **“Impact of lifespan-extending intervention on MLS-associated alternative splicing events”** (page 17-18).

“To explore if interventions that extend lifespan affect MLS-associated AS events, we analyzed mice treated with a Pregnancy-associated plasma protein-A (PAPP-A) inhibitor [63]. PAPP-A enzymatically cleaves insulin-like growth factor binding proteins (IGFBPs) to liberate IGFs and enhance local IGF signaling. PAPP-A inhibition reduces IGF signaling and has been shown to extend mouse lifespan [64, 65]. Specifically, we assessed the change in percent spliced-in values (Δ PSI) between PAPP-A inhibitor treatment and controls for three categories of AS events: MLS positively correlated (pos-MLS), MLS negatively correlated (neg-MLS), and background (non-MLS-associated) AS events.

We found that pos-MLS events were more frequently upregulated (exon inclusion increased) in treated mice compared to control non-MLS-associated exon group, whereas neg-MLS events were the least likely to be upregulated (Supplementary Fig. 6). These trends held across a range of Δ PSI thresholds, based on the proportion of events showing significant PSI increases (FDR < 0.05 by Fisher’s Exact Test on median read counts). This result shows that PAPP-A inhibitor, a longevity-enhancing intervention, tends to increase the inclusion of exons that are positively associated with MLS (Supplementary Table 5). In other words, such interventions appear to shift the splicing profile to more closely resemble that of long-lived species”

However, we acknowledge that this is a single intervention in one species with sufficient RNA-seq coverage for splicing analyses. Further research incorporating a broader range of lifespan-extending strategies, including pharmacological, dietary, and genetic approaches, is needed to evaluate whether these splicing changes are a generalizable feature of lifespan modulation.

5. The description of the procedures for identifying the homologous AS events is not sufficiently detailed. For example, what query sequences were used to perform blast against the exon or intron database. Some AS events are quite sensitive to false positives or negatives. How does the authors guarantee that the alternative first/last exons events have not resulted from the incomplete assembly of the transcripts that missing the first/last exon (especially some genes have very short first exon that could be easily missed)? The intron sequences are quite variable

across species. For each species, did the authors searched against their own intron sequences or directly against mouse intron sequences? If the later, will the results keep the same if they use each species' own intron sequences as database? How did the authors define the homologous alternative 5'/3' splice sites? Would the alternative 5'/3' splice sites to be the same between species or the case of different splice sites on an orthologous exon is also included? The authors excluded exon that with alignment rate less than 80%. Is it possible these less aligned exons had resulted from the alternative 5'/3' splicing? I would suggest the authors to describe the procedures clearer and if possible, perform additional filtering steps to improve the accuracy of the AS identification.

Reply:

Thank you for these thoughtful and detailed comments. We have revised the manuscript and used Supplementary Fig. 2 to clarify the procedures used for identifying homologous alternative splicing (AS) events across species, and we have added further methodological details as suggested. These clarifications and supporting analyses have been added to the revised manuscript (page 4-5, 8, 26-27) and explained below.

To begin, we used SUPPA to identify alternative splicing events based on assembled transcripts within each species. Splice sites are defined at the RNA instead of DNA level. Then, for cross-species comparison, we performed BLAST alignments of the alternatively spliced regions against the corresponding sequences in mouse. As illustrated in Supplementary Fig. 2, we used exonic sequences for most event types, except for retained introns, where the intronic sequences were used. No other intronic sequences were used in the alignment process, so variability in intron sequences across species does not affect our analysis.

For alternative first exon (AFE) and alternative last exon (ALE) events, we acknowledge the concern that incomplete transcript assembly, especially for short exons, could lead to false positives. However, in our analysis, we require that both exons involved in the event (e.g., the first and second exons for AFE) in species 1 have homologs in species 2. This dual requirement significantly reduces the likelihood that both exons are artifacts of incomplete assembly, making such false positives relatively rare. Note that no event of these two types was conserved in at least 10 species, which may be attributed to the higher variability of UTR regions compared to coding sequences and makes it more challenging to confidently establish homology.

For alternative 5' and 3' splice site events (A5SS/A3SS), homologous events were defined based on the alignment of both the alternatively spliced region (i.e., the segment between the two alternative splice sites) and the adjacent upstream/downstream exon. If splice site variation leads to a lower alignment rate between exons, the event is still retained as homologous, provided that both the alternatively spliced region and the flanking exon meet the alignment criteria. This approach ensures that genuine A5SS/A3SS variations are preserved.

To assess the robustness of our method, we compared our homologous AS event predictions to those from Barbosa-Morais et al. [PMID: 23258891], who reported 404 mouse–rat homologous skipped exon (SE) events. Using our pipeline, we identified 294 of the same mouse SE events, with 292 matching the corresponding rat event, yielding a 99.3% concordance (page26-27). As their method using orthologous exon mapping followed by PSI filtering is different from ours using SUPPA-defined AS events and sequence alignment, the low mismatch rate demonstrates the stringency of our approach. The partial difference in total matched events (73%) is due to differences in event definitions.

To further support the biological validity of our method, we tested whether the number of homologous AS events declines with evolutionary divergence. Using 100 phylogenetic trees generated by PhyloT, we performed Spearman correlation analysis and observed a median correlation of -0.51 (page8). This significant negative correlation is consistent with the expected trend that AS conservation decreases with evolutionary distance, lending further support to the robustness of our approach.

Again, we thank the reviewer for the insightful suggestions that helped us improve the rigor and transparency of our methodology.

Minor points:

1. the p-values should be shown in scatter plots (Fig.4a,4b); to address the multiple testing issue, the adjusted p-value should be provided, for example, in the GSEA enrichment analysis.

Reply:

We appreciate the reviewer's suggestion regarding the display of p-values in scatter plots and the importance of correcting for multiple testing. To address this, we have incorporated p-values and FDR into the relevant scatter plots as suggested (the former Figure 4 is now Figure 3 due to the addition and reorganization of figures).

In response to Reviewer 2's recommendation to use a more updated tool for GSEA analysis, we have replaced the original GSEA analysis with Enrichr (PMID: 23586463), which performs overrepresentation analysis and reports Benjamini–Hochberg adjusted p-values by default. The adjusted p-values are provided in Figure 6 and Supplementary Table 3. The results are updated in the Result section "Functional enrichment of MLS-associated alternative splicing" on page 14-17.

2. are all AS events profile identified in this study clustered by species or by organ? The authors may have to show the heatmap of the correlation coefficient matrix to make sure the pipeline works well.

Reply:

The heatmap of the correlation coefficients, as suggested by the reviewer, is now presented in Fig. 4B, which visually highlights the tissue-specific correlation patterns of MLS-AS events. In our study, MLS-associated AS (MLS-AS) events were clustered based on their correlation with maximum lifespan (MLS) across multiple organs, with the goal of identifying tissue-specific splicing patterns. Since the correlation was calculated between PSI and MLS within each tissue across 26 species, each event has only one correlation value per tissue, making it not feasible to cluster by species. Combined with the UpSet plot in Fig. 4A, these results show a considerable number of consistent MLS-AS events across tissues, while also highlighting the distinctiveness of the brain, where many events show unique correlation patterns.

3. for the GSEA dot plots, why did the authors show the gene ratio and counts instead of showing the NES?

Reply:

We thank the reviewer for the helpful comment. As we have now switched to using Enrichr for functional enrichment analysis (as requested by Reviewer 2), normalized enrichment scores (NES) are not applicable in this context. Instead, we report odds ratios and Benjamini–Hochberg

adjusted p-values, which are the standard metrics for interpreting results from overrepresentation-based analyses.

4. "Several terms related to metabolic processes were enriched in MLS- [...]. Metabolic changes are hallmarks of aging." In general, the authors should be more specific in their statements. What do they mean by "metabolic changes"? Which processes and pathways are involved? The term "metabolic changes" is too generic and does not help the reader make sense of the results.

Reply:

We appreciate the reviewer's suggestion to be more specific regarding the metabolic processes enriched in MLS-associated AS events. In the revised manuscript, we have replaced the broad term "metabolic changes" with specific pathways identified in our GO and KEGG enrichment analyses, including PI3K-AKT signaling, and MAPK cascade (page 14). We have also listed other specific terms under specific functional categories.

5. "A longevity-regulated pathway"—again, this sentence is too generic. To which pathways are the authors referring?

Reply:

Thank you for pointing this out. In the revised manuscript, we have replaced the generic reference to the "longevity-regulating pathway" (KEGG pathway hsa04211) with more specific and well-characterized aging-related pathways identified in our Enrichr analysis. These include terms related to stress response pathways, apoptosis, and neuronal function. We have elaborated on their relationship to longevity controls (page 14-17). This revision improves the biological specificity and aligns more closely with the updated enrichment results.

6. "The union of age-associated AS events across all sub-tissues"—This statement, while reflecting an observation from the authors, does not seem relevant or justified, as the authors analyze a substantially higher number of tissues from different brain regions compared to other tissues. The higher number could simply be the result of the increased number of tissues compared.

Reply:

Reviewer 1 has a good point that the higher number of brain sub-tissues analyzed could increase the total count of age-associated AS events, and we agree. We want to emphasize that our intent is to highlight the pronounced **heterogeneity** among brain regions, which is the other variable contributing to the higher number of age-associated AS events observed in the brain. To demonstrate this, we conducted pairwise similarity analyses using the Overlap Coefficient (OC), which measures the proportion of shared AS events between sub-tissues. Interestingly, brain sub-tissues showed a lower OC (average of 0.38) compared to the two sub-tissues of heart (0.42) and skin (0.49), indicating greater heterogeneity in age-associated AS events across brain regions.

To further validate this observation, we also calculated the Jaccard Index, which accounts for both shared and unique events. Again, the brain exhibited the lowest Jaccard Index (average of 0.22) relative to heart (0.25) and skin (0.31), suggesting that brain sub-tissues share fewer common AS events overall.

These findings support the conclusion that the elevated number of AS events observed in the brain reflects true biological diversity in splicing regulation across distinct brain regions in addition

to more sub-tissues analyzed. The updated analyses and discussion have been incorporated into the revised manuscript (page. 18).

“We further investigated the heterogeneity among brain regions given this large number of age-AS events. We performed pairwise similarity analyses using the Overlap Coefficient (OC). Brain regions shared relatively few age-AS events with each other (lower OC: average of 0.38), indicating high regional specificity. In contrast, more overlap was seen among sub-tissues of the heart (OC: 0.42) or skin (OC: 0.49), suggesting splicing changes with aging are more homogenous within these tissues than in the brain. The Jaccard Index values also confirmed this pattern (brain = 0.22, heart = 0.25, skin = 0.31), further demonstrating that brain sub-tissues have less overlap in AS events compared to other tissues. This suggests that the elevated alternative splicing event count in the brain reflects genuine biological diversity in splicing regulation across brain regions”.

7. The species considered share an evolutionary history, so the data points used to calculate Spearman’s and Pearson’s correlations are not independent. A more appropriate approach would be to use phylogenetic least squares regression analysis to account for the natural relationships that organisms share based on their evolutionary history. The authors should perform their analysis using this correction, which would strengthen their results on AS sites and MLS.

Reply:

We thank the reviewer for the thoughtful suggestion regarding phylogenetic non-independence and agree that accounting for shared evolutionary history is important when analyzing traits across species. To address this, we applied phylogenetically independent contrasts (PIC) to correct for this non-independence. Given that our analysis involves a single predictor variable (PSI vs. MLS), PIC is a well-established and appropriate method that yields results equivalent to those from phylogenetic generalized least squares (PGLS, more applicable in multi-variable regression contexts).

The use of PIC appropriately addresses the phylogenetic structure of the data and provides a robust correction for shared evolutionary history. This clarification has been added to the revised manuscript (page 6, 9-10).

“Because closely related species might have similar lifespans and splicing patterns due to common ancestry, we conducted an analysis of phylogenetically independent contrasts (PICs) on the original PSI values to ensure correlations were not driven by phylogenetic relationships. The evolutionary relatedness among species can introduce statistical dependencies, potentially leading to biased conclusions. PICs adjust for these dependencies by statistically accounting for phylogenetic relationships among species. After applying a threshold of absolute correlation greater than 0.2 to the corrected data, we observed that 83% of associations remained significant after correction. This underscores the robustness of our discovered MLS-AS associations.”

Reviewer 2

1. There is no code to reproduce the results.

Reply:

We thank the reviewer for pointing this out. We have deposited the code in Zenodo as documented in the Data Availability section of the manuscript.

<https://zenodo.org/records/16042325>

2. The authors should compare their pipeline to find homologous alternative exons across species with previous studies. I wonder if the amount of conserved AS events they detect is what is expected. Do they detect conserved AS that have been described before?. Finding homologous AS events is a fundamental step in their study. The only “validation” they provide is the statement that “the number of homologous events correlated with the evolutionary distance from mice, spanning from 900 to over 3700”. They should statistically test this by performing a proper correlation between evolutionary distance and homologous AS events. Sorting the species in plot 2a by evolutionary distance would also help see the mentioned correlation. Also, this would be an expected result for blasting any type of sequence, not only splicing junctions. So further validations/comparisons with previous work is needed to show that their pipeline finds true homologous AS events. Also, authors should say if the finding that exon skipping events are the most conserved and AF and AL the least is expected based on previous work. It would be interesting to assess if any AS type is more conserved than expected (e.g MX are the least common but seem to be very conserved). Papers from the Kaessmann lab and Blencowe lab among others have relevant work in this regard.

Reply:

Thank you for this valuable suggestion. We agree that identifying homologous AS events is a fundamental component of our study and appreciate the opportunity to clarify and strengthen this aspect of our analysis.

To assess whether the number of conserved AS events we identified is consistent with expectations, we note that our pipeline applies stringent criteria for defining homologous events. As a result, we anticipate detecting fewer events (also fewer false positives) than might be found if less stringent criteria were applied. To statistically test the relationship between evolutionary distance and the number of homologous AS events, we performed a Spearman correlation analysis using 100 phylogenetic trees generated by PhyloT. The median correlation coefficient was -0.51 , demonstrating a strong negative relationship, consistent with the expectation that the conservation of alternative splicing events declines as evolutionary divergence increases. This result is now incorporated in the main text (page 8).

To further validate our pipeline, we compared our results to those of Merkin et al. (PMID: 23258891), which identified 404 conserved AS events between mouse and rat. Their approach identified orthologous exons followed by PSI filtering, while ours is based on SUPPA-defined splicing events followed by sequence comparison. The differences should result in variations of event identification. Nevertheless, we recovered 294 of the 404 AS events. Importantly, only 2 corresponding events differed from theirs—indicating a 99.3% concordance. Despite the methodological differences, our extremely low error rate (0.7%) among the matched pairs highlights the accuracy of our homologous AS identification. This result is now incorporated in the main text (page 26-27).

We also appreciate the recommendation to place our findings in the context of previous studies. Indeed, prior comparative work by Barbosa-Morais et al. from Blencowe's lab (PMID: 23258890), those from Burge's lab (PMID: 23258891), and Mazin et al. from the Kaessmann's lab (PMID: 33941934, 31243369) have shown that exon skipping is the most prevalent AS type and reported some degree of conservation, which is consistent with our observation. These studies have not comprehensively determined the conservations of other AS types.

The poor conservation of alternative first exon and alternative last exon events is probably due to the presence of untranslated regions in these areas, which have less evolutionary pressure. Interestingly, although mutually exclusive exons (MX) are rare, they appear to be highly conserved in our dataset. The application of the stringent criteria on four splice junctions for their identification may be one factor.

A recent comprehensive catalog (PMID: 29242366) found 855 MX events in humans using 515 RNA-seq datasets. This is a tiny fraction compared to tens of thousands of cassette exons. Thus, MX events are less frequent by far than AF, AL, or cassette exon events. They also reported high conservation of MX clusters. Therefore, our observations about the frequency and conservation of MX align with the prior study. We agree that this pattern merits further exploration, ideally using more complete annotations of non-model organisms when they are available. We have also expanded the discussion of these findings and relevant literature on page 26-27.

3. plot in 2c) would benefit to show conservation normalized by the total number of events detected, otherwise it is hard to say whether events are more conserved or less than expected.

Reply:

Thank you for highlighting the importance of normalization in interpreting conservation levels. In response, we have updated the plot to display conservation as a percentage for each AS type, calculated as the number of conserved homologs found in ≥ 10 species divided by the total number of events identified for that AS type. This normalization allows for a more meaningful comparison across AS types. Please note that the former Figure 2 is now Figure 1 due to the addition and reorganization of figures.

The updated analysis reveals that exon skipping (ES) events are significantly more conserved than expected (17.2% vs. an overall average of 5.9%), while retained introns (RI) show markedly lower conservation (0.8%). We appreciate the reviewer's helpful suggestion, which improves the interpretability of our findings by accounting for differences in AS type abundance. We have expanded the text describing these results (page 8-9).

4. If BM and MLF are correlated, they should test both in the same linear model rather than doing a spearman correlation with each and then looking at the overlap. However, If they are extremely correlated (> 0.8) it may actually not even be possible to distinguish between the two. Importantly, modelling PSI with linear models is not straight forward because the distribution of PSI values is between 0 and 1 with most values being very close to zero and one. The authors could perhaps look at how other studies have addressed this. For example, in Garcia-Pérez et al. Cell Genomics 2024 (PMID: 36777183). The number of associated AS events with BM and MLS should be obtained in the same model to identify MLS or BM specific events. In addition, the observation that the brain behaves differently should be tested explicitly (and not just showing that the overlap is smaller, as this could be due to stochastic processes). Finally, the authors should also explain in the methods how the jaccard index shown in Fig 3 was obtained.

Reply:

Thank you for these thoughtful and important suggestions.

We first examined the relationship between BM and MLS and found that they are only moderately correlated (Spearman's $\rho = 0.44$), suggesting that while related, they are not redundant and can be considered separately in modeling.

While García-Pérez et al. (PMID: 36777183) modeled PSI as the response variable, our framework treats MLS as the response variable, using PSI as a predictor to assess its association with lifespan.

In response to the reviewer's recommendation, we implemented a multiple linear regression model of the form: $MLS = \beta_0 + \beta_1 \cdot BM + \beta_2 \cdot PSI$ for each of our identified MLS-AS events. We then categorized the results based on the significance of each predictor (threshold: $p < 0.05$). These new results are incorporated into the main text (page 11).

“Although BM and MLS are moderately correlated (Spearman correlation = 0.44), they represent distinct biological traits and are not redundant. To disentangle their individual effects, we applied the model $MLS = \beta_0 + \beta_1 \cdot BM + \beta_2 \cdot PSI$ for each identified MLS-AS event. Our results showed that PSI contributes independently to MLS in 59% of cases (where p value for $\beta_2 < 0.05$ and p value for $\beta_1 > 0.05$), while both PSI and BM jointly contribute to MLS in 35% of cases (p values for both β_2 and $\beta_1 < 0.05$). Altogether, in 94% of MLS-associated AS events, PSI was a significant predictor even after accounting for BM, supporting the idea that splicing contributes independently to lifespan regulation.”

We agree with the reviewer that the distinct behavior of brain tissues warrants explicit testing. Using a hypergeometric test, the enrichment for the overlap between MLS-PSI association and BM-PSI association in the brain exhibited the weakest enrichment among all tissues (adjusted $p = 2.87 \times 10^{-5}$ vs. adjusted $p < 10^{-24}$), despite still being statistically significant. This suggests that while the brain also shows overlap, the degree is substantially lower than in other tissues. These new results are incorporated into the main text (page 10-11).

Lastly, we have clarified in the legend of Figure 2 (former Fig 3) that the Jaccard Index was calculated as the size of the intersection divided by the size of the union of BM-associated and MLS-associated AS event sets.

5. The statement “Intriguingly, the brain distinction from other tissues was not observed in the MLS- and BM-association study with gene expression (Figs. 3e and 3f), indicating potential divergence in the regulatory functions of AS and gene expression in brain aging.” What gene expression analysis is it referring to? How do they quantify gene expression, I guess adding up the transcript expression values but it is not explained in the methods.

Reply:

Thank you for pointing this out. The gene expression analyses referenced in Fig. 2e and 2f (the former Fig 3) were conducted using the same modeling framework as the splicing analysis, but with gene expression values substituted for PSI. We have rephrased the sentence to “**Intriguingly, when analyzing gene expression (instead of splicing) associated with MLS and BM, the brain did not show this distinction (Fig. 2e and 2f)**” (Page 11)

Gene expression quantification was performed using StringTie in expression estimation mode (i.e., adding up the transcript expression), which provides gene-level expression estimates in Transcripts Per Million (TPM). These TPM values were then used in the downstream analyses.

We have clarified this procedure in the Methods section of the revised manuscript (page 4, 5) to ensure reproducibility and transparency.

6. The fact that the authors find the brain having the largest number of AS-associated MLS events in the brain should be put in context with the fact that AS patterns in the brain are also the most distinct compared to other tissues and cite the corresponding literature. I wonder how much of the signal they see could be explained by this fact alone.

Reply:

We thank the reviewer for this insightful comment. To clarify, when considering the total number of MLS-associated AS events (including both tissue-specific and shared events), the count in the brain (220 events, or the sum of positive-association events in Fig.2a and negative-association events in Fig. 2b) is **comparable** to that of other tissues (mean: 235; median: 229.5). The brain stands out for exhibiting the highest number of tissue-specific MLS-associated AS events, or events whose association with MLS is significant in only one tissue. We have revised the text to explicitly distinguish between tissue-specific and total MLS-associated AS events to improve clarity for readers.

We agree with the reviewer that the brain's distinct splicing landscape, as reported in prior studies (e.g., Barbosa-Morais et al., 2012; Raj et al., 2014), may contribute to this pattern. To address whether the enrichment of MLS-associated AS events in the brain simply reflects its unique splicing profile, we analyzed the PSI values of these events across other tissues and species. We found that the majority of these brain-specific MLS-AS events are not exclusive to the brain: only ~3.84–7.85% of PSI values were zero across the other five tissues. This indicates that while these events are broadly present, their association with MLS is specific to the brain, suggesting tissue-specific regulatory relevance rather than tissue-restricted splicing. We have incorporated this explanation and relevant citations into the revised manuscript on page 24-25.

7. Regarding the functional enrichment analysis on the AS-MLS genes. It would have been perhaps interesting to compare whether the enrichments are different between those AS-MLS and differential gene expression-MLS genes previously described using the same dataset? Also, the authors don't specify whether the pathways are enriched among positively or negatively correlated genes with MLS, which is important for their interpretation.

Reply:

We appreciate the reviewer's insightful suggestion regarding the comparison of functional enrichment between MLS-associated AS events and previously reported MLS-associated gene expression changes using the same dataset.

To explore this, we performed a comparative analysis of enriched pathways identified from MLS-associated AS genes and those identified from MLS-associated gene expression. The overlap between the two sets of enriched terms was minimal. MLS-AS genes are overwhelmingly enriched in pathways related to multiple steps of mRNA processing, stress response, neuronal functions, and epigenetic regulation. Therefore, certain pathways contribute to MLS predominantly through splicing regulation, while others are more affected by gene expression modulation. These findings are updated in the Result section "**Functional enrichment of MLS-associated alternative splicing**" on page 14-17.

"To understand the biological significance of MLS-associated AS (MLS-AS) events, we conducted functional enrichment analysis using Enrichr. We examined genes harboring MLS-associated splicing events for enrichment in the Gene Ontology (GO) biological process terms

and KEGG pathways. MLS-AS genes are significantly enriched in functional categories of mRNA processing, stress response, neuronal functions, and epigenetic regulation (Supplementary Table 3). Figure 6 summarizes the significantly enriched terms (FDR < 0.15), organized by functional categories.

Multiple enriched pathways involve cellular stress response, including Stress Granule Assembly, Regulation of Defense Response to Virus, MAPK Cascade, *Positive Regulation of PI3K-AKT Signal Transduction*, etc. stress response plays an essential role in safeguarding cellular and organismal integrity [44-46]. When cells encounter stressors such as oxidative damage, heat shock, viral infection, or metabolic fluctuations, they initiate intricate signaling pathways to mitigate harm and restore homeostasis. Efficient stress response mechanisms enable cells to quickly detect and neutralize damage, activate repair systems, and, if necessary, trigger programmed cell death (apoptosis) to remove irreparably compromised cells. Indeed, Regulation of Apoptotic Process (and related GO terms) is significantly enriched (Figure 6). Apoptosis ensures that tissues maintain their functional capacity and that damaged cells do not accumulate [47, 48]. Moreover, the adaptability of the stress response is crucial for long-lived species. Organisms with robust stress response networks can better withstand environmental and physiological fluctuations across their lifespan, delaying the onset of cellular dysfunction and chronic pathologies. Alternative splicing of stress response genes likely modulates organisms' competence for the efficiency and adaptability of stress response.

Notably, mRNA processing steps and regulatory mechanisms are enriched, including spliceosomal complex assembly (along with many other splicing-related terms), cleavage and polyadenylation, regulation of translation, mRNA surveillance, and RNA transport. Collectively, these comprise 18% of all enriched GO terms and encompass nearly every stage of the mRNA life cycle. This underscores the importance of maintaining transcriptome integrity and proper mRNA processing for longevity. Errors or inefficiencies in any of these processes can result in the production of faulty or non-functional proteins, accumulation of aberrant RNA species, and ultimately, cellular dysfunction. For long-lived species, the ability to maintain precise and adaptable RNA processing systems is particularly critical. Accurate splicing ensures that essential gene products are expressed in correct proportions and forms, while alternative splicing provides an additional layer of regulatory flexibility, allowing organisms to rapidly adapt to changing physiological conditions without altering their genomic DNA. mRNA surveillance mechanisms, such as nonsense-mediated decay, also respond to cellular stress and safeguard the transcriptome by identifying and eliminating defective mRNAs that could otherwise give rise to toxic or deleterious proteins [49]. Thus, robust RNA processing and regulation are central to maintaining cellular homeostasis and adaptability, both of which are essential for healthy aging. Their enrichment suggests that longevity in some species may be achieved, at least in part, by optimizing alternative splicing mechanisms that control the transcriptome's quality, diversity, and responsiveness throughout the lifespan.

Additionally, we observed a significant enrichment of biological processes related to neuronal functions, such as Regulation of Neuronal Synaptic Plasticity, Axon Guidance, Synapse Organization, Brain Development, Neuron Migration, Long-term potentiation, Neurotrophin Signaling Pathway, Neurotransmitter Secretion, etc. Neuro-related terms represent over 22% of all enriched GO terms, further reinforcing the distinctive role of brain-specific [50-59]. The pivotal role of neuronal function in determining organismal lifespan likely arises from the brain's central control over homeostasis and its orchestration of numerous physiological processes intimately connected to aging and longevity. Learning, memory, and adaptability of neural circuits are intricately regulated by pathways involved in axon guidance, synaptic organization, and neurotrophin signaling. Through these mechanisms, neurons can adapt to developmental cues and environmental stressors. Importantly, the brain influences systemic aging via neuroendocrine and autonomic pathways. For example, the hypothalamus integrates cues related to energy balance, stress, and circadian rhythms, coordinating hormonal outputs that impact metabolism,

immune responses, and tissue regeneration. In this way, neuronal function serves as a central determinant of lifespan—supporting cognitive maintenance, regulating systemic physiology, modulating immune activity, and conferring resilience to stress. In summary, these findings highlight the diverse biological processes modulated by alternative splicing to influence maximum lifespan.

To compare the respective contributions of gene-level and splicing-level regulation to lifespan, we compared the functional enrichment of genes with MLS-associated AS events and those with MLS-associated expression changes (as identified by Lu et al.). First, Among the 576 genes with MLS-associated AS events (MLS-AS) and the 6,952 genes with MLS-associated expression changes (MLS-gene), only 211 genes overlapped. Using a background of 17,511 multi-exon genes, this overlap yielded a p-value of 0.94 (Fisher's Exact test), indicating it is not statistically significant.

Comparing their pathway enrichments reveals clear distinction. MLS-genes are enriched in GO group terms such as DNA replication, mitotic recombination, chromosome segregation, double-strand break repair, protein folding, mitochondrion organization, cilium movement, and lymphocyte chemotaxis. They are also particularly enriched in various macromolecular metabolism pathways, including acylglycerol metabolic process, amino acid catabolic process, carbohydrate biosynthetic process, ribonucleoside biphosphate biosynthetic process, hydrogen peroxide catabolic process. These findings align with current theories that emphasize DNA repair [60] and metabolic control in longevity. Proper regulation of macromolecule metabolism is particularly crucial for maintaining cell structure and function, and repairing damaged tissues [61, 62]. These GO terms or functional groups are not prominent in MLS-AS genes. Overall, only seven out of 265 GO terms from MLS-AS events are enriched in MLS-gene groups (FDR <0.15 for both groups). When the threshold is tightened to FDR < 0.1, this overlap drops to just one exact shared term, and at FDR < 0.05, there are no overlapping GO terms between the two groups. We therefore used FDR <0.15 for comparisons. Since different GO terms can still be related under one functional activity, we used REVIGO to organize the GO terms into functional categories. The results still highlighted a clear distinction. For example, among the 424 total GO terms enriched in MLS-genes, only three were related to neural function (0.7%) and ten to RNA-related processes (2.3%). Similarly for MLS-AS genes, DNA Repair is only minorly represented with two specific terms, "Regulation of Nucleotide-Excision Repair" (FDR: 0.056) and "Positive Regulation of DNA Repair" (FDR: 0.082). This limited intersection underscores that alternative splicing modulation acts as an independent regulatory layer to gene expression regulation. ”

Regarding the directionality of MLS associations, we acknowledge that in gene expression studies, the direction (e.g., up- or downregulation) provides clear biological insight. However, in the context of alternative splicing, the interpretation is less straightforward. AS is quantified by PSI values, reflecting splice-in inclusion levels rather than expression abundance. An observed change in PSI could indicate an opposite trend in the expression of the inclusion vs exclusion isoforms. Therefore, labeling events as "positively" or "negatively" associated with MLS does not directly translate to biological activation or repression of a gene or pathway. Due to this complexity, we chose not to emphasize directionality in our functional enrichment of MLS-AS genes.

8. The authors don't correct for some of the technical covariates routinely included in previous GTEx publications (Taru Tukiainen et al., 2018 Nature PMID: 29022598; Pedro G. Ferreira et al., 2018 Nat. comms. PMID: 29440659; Garcia-Pérez et al. Cell Genomics 2023 PMID: 36777183), like Ischemic time and Hardy Scale. They say they use the covariates used in sQTL calling but they do not cite any paper but in previous GTEx publications, technical covariates are always included in the analysis. Thus, the authors should correct for them or demonstrate that the additional included PEER factors are correcting for those technical covariates.

Reply:

Thank you for pointing this out, and we apologize for the lack of clarity in our initial description. We followed the covariate framework used in the GTEx sQTL analysis, which includes sex, the top five genotype principal components, 15 PEER factors, as well as sequencing platform and protocol. We have now updated the manuscript to explicitly cite the relevant GTEx dataset (e.g., PMID: 23715323, 32913098).

9. In the RBP enrichment analysis they don't correct for expression levels to control for highly expressed genes. Highly expressed genes would be expected to have more RBP binding sites and could therefore be confounding the results.

Reply:

Thank you for this important comment.

We would like to clarify that the RBP enrichment analysis is based on motif enrichment within MLS-associated AS events and higher expression does not contribute more to the enrichment analysis than lower expression. The MLS-associated AS events were derived from PSI (Percent-Spliced-In) values. PSI, by definition, reflects relative splicing inclusion and is not influenced by absolute gene expression levels.

On the other hand, since the enrichment analysis is applied to the sequence regions relevant to an AS event (not the whole gene), we accounted for sequence length, specifically the length of regions involved in the alternative splicing event, including exonic and flanking intronic sequences. Since length is correlated with the number of potential RBP binding sites, this normalization helps to control for a major confounding factor.

10. Analyzing AS change with age using the GTEx is not novel and it has been done before. For example, Garcia-Pérez et al. *Cell Genomics* 2023 (PMID: 36777183). It would be nice if the authors compared their findings with previous literature to show that their elastic net regression method works. The authors could also explain the differences between their method and the recently published one, and why this new methodology is better suited for the aim of the paper.

Reply:

We thank the reviewer for highlighting the study by Garcia-Pérez et al. (*Cell Genomics*, 2023). While both studies leverage GTEx data to investigate alternative splicing, our approaches differ significantly in both methodology and research focus.

Garcia-Pérez et al. modeled PSI as the dependent variable, with age, sex, and tissue type as predictors, focusing on predicting PSI values from age and covariates. In contrast, our study models age as the dependent variable, using PSI values from many AS events as features. This inversion allows us to identify splicing events that affect aging combinatorially.

Our use of elastic net regression is particularly well-suited for this task, as it handles high-dimensional datasets with correlated variables (such as PSI across many splicing events), performs variable selection, and reduces the risk of overfitting, making it more robust for identifying biologically meaningful splicing predictors of lifespan. We have also added a comparison with traditional regression models, which further demonstrates the robustness of the elastic net regression model (see below).

We attempted to directly compare our results with those of Garcia-Pérez et al., but the absence of detailed lists of age-associated splicing events in their publication limited our ability to do so.

We have added a discussion of this prior work in the revised manuscript under a new section “**Analytical strategies for cross-species splicing analyses**” to provide context and highlight the novelty of our approach (page 25-26).

“We applied elastic net regression on the GTEx human dataset to identify age-associated AS events. This approach effectively accounts for complex relationships between samples and incorporates measured covariates, setting it apart from the method recently used by Garcia-Pérez et al. [109], which employed a different analytical framework and focused on predicting PSI values from age and covariates. The robustness of our model is demonstrated by high Jaccard index values between similar tissues and sub-tissues (Fig. 7b). In contrast, traditional regression models that analyze each AS event in isolate, as in a previous study [110], yielded lower tissue similarity (Supplementary Fig. 9). This highlights the advantage of elastic net regression in simultaneously modeling multiple AS events.”

11. The authors use elastic net regression to identify age-associated splicing events. To show that this method is well calibrated, I would strongly encourage the authors to run the method permuting the labels to show that there are no false positives and the method is well calibrated.

Reply:

We thank the reviewer for the helpful suggestion regarding permutation testing to assess the calibration of our elastic net regression model. We have performed this analysis according to the reviewer’s request and added the results to the revised manuscript (page 26).

“To rigorously assess the model’s effectiveness, we conducted 500 permutation tests by randomly shuffling age labels across samples. Under the null hypothesis, these permutations yielded an average of approximately 100 significant AS events per iteration, providing an empirical estimate of the false positive rate. By contrast, our actual analysis identified approximately 400 significant AS events, substantially exceeding the permutation-based expectation. Notably, AS events identified in the permutation tests showed no enrichment for age-related pathways. Together, these results support that many of the identified AS events reflect true biological signals.”

We also want to clarify that feature selection in our model is determined by non-zero coefficients in the elastic net output. Unlike traditional hypothesis testing, this approach does not produce p-values, and thus false positives cannot be simply discarded by thresholding the smallest coefficients. To address this, we took an additional step to assess biological relevance by analyzing functional enrichment. The results show that AS events identified in the permutation tests did not show enrichment in age-related pathways, whereas those from the original analysis did. This functional difference further supports the validity of our results.

Given these findings, we chose to retain all significant AS events from the elastic net regression model, in order to preserve analytical consistency. The current approach offers a strong and well-supported foundation for identifying age-associated splicing events.

Minor comments:

1. version of Stringtie (for the alternative splicing quantification)? What is the version of GENCODE used? Where are the reference genomes downloaded from? Where does the MLS information come from?

Reply:

Thank you for pointing out the missing methodological details. For alternative splicing quantification, we used SUPPA v2.3. The gene annotation files were based on GENCODE Release vM30 for mouse and v26 for human. Reference genome sequences were downloaded

from the UCSC Genome Browser. Maximum lifespan (MLS) data for the 26 non-human mammalian species were obtained from Lu, J. Y. et al. 2022 (Reference #9).

We have updated the manuscript to include these details (page 3-4) for clarity and reproducibility.

2. DAVID is not maintained since a long time ago. Another functional enrichment method should be used. Additionally, the background used for the GO analysis should be specified.

Reply:

Thank you for the valuable feedback. To address this, we have re-analyzed our gene sets using Enrichr, a well-maintained and widely used enrichment analysis tool. The updated results are now included in the revised manuscript (Fig 6, page 14-17).

For the analysis related to intrinsically disordered proteins (IDPs), we have retained DAVID, as it provides specific annotations relevant to IDPs that are not directly available through Enrichr.

Regarding background gene sets, DAVID and Enrichr both use the full genome-wide annotated genes as the default background.

3. cell type composition is a known confounder on transcriptomic analysis, the authors should maybe explain how this may influence their results.

Reply:

Thank you for raising this important point. We acknowledge that cell type composition is a well-known confounder in transcriptomic analyses and may influence splicing patterns across tissues and species. In our revised manuscript, we have added relevant discussion under the section “**Limitations of the study**” (page 28).

“Cell type composition is a known confounder in transcriptomic analyses, as variations in cellular makeup, even within the same tissue across samples, can affect the inferences of both gene expression and splicing levels. The MLS-AS events we identified exhibit tissue-level differences between long-lived and short-lived species, which may reflect the changes at the level of individual cells or shifts in overall cell type compositions, or both. Because information of cell-type-level splicing or the exact cell type compositions within tissues is unavailable for most species, current research is unable to disentangle these possibilities. Future studies incorporating single-cell RNA-seq in combination with applying cell-type deconvolution methods could provide deeper insights into the cell-type-specific regulation of AS events in the context of lifespan regulation.”

4. In the RBP enrichment analysis, the authors use 10 random groups as background. Why not using the whole set of tested AS events and compute a fisher/chi-square test?

Reply:

Thank you for this thoughtful comment regarding our RBP enrichment analysis. We chose to use randomized control groups, which were matched for splicing event type composition, to avoid compositional bias, as the overall pool of AS events has a markedly different distribution of event types between MLS-associated set and age-associated set. Using the full set directly as background could inflate enrichment signals due to this imbalance.

In response to your suggestion, we have increased the number of random samplings from 10 to 500 iterations to enhance statistical robustness (page 7).

5. The authors state that 37% of AS events are associated with MLS and the majority are exon-skipping. It would be interesting to know if any AS type is more associated with MLS than expected. Also, is this 37% different from the proportion of genes transcriptionally associated with MLS found in the previous study using the same dataset?

Reply:

We thank the reviewer for their insightful questions. No specific AS type was significantly more associated with MLS than expected. This suggests that MLS-associated splicing regulation is not biased toward a particular AS type but rather reflects a broad influence of splicing on longevity. In contrast to the 37% of AS events associated with MLS, only ~20% of genes showed transcriptional associations with MLS in the same dataset. This difference suggests that alternative splicing may play a more widespread role in MLS regulation than gene expression alone, further supporting the complementary nature of AS- and expression-based regulatory mechanisms in lifespan evolution. This has been added on page 9.

“In total, 731 out of 1974 conserved AS events (37%) showed an MLS association in at least one of the six tissues, indicating that a substantial fraction of splicing events may be linked to lifespan differences. For comparison, only about 20% of genes showed expression levels correlated with MLS [9], suggesting that alternative splicing captures additional lifespan-related signals beyond gene expression alone. The majority of these AS events were cassette exon, and the distribution of AS types closely mirrored that of all homologous AS events considered, indicating that no specific AS type is disproportionately enriched among the MLS-associated events (Supplementary Fig. 4).”

6. When testing multiple hypergeometric tests (Figure 3a-3b), a correction for multiple testing should be used on the p.values.

Reply:

Thank you for pointing this out. We have now applied FDR correction to the p-values from the hypergeometric tests for the overlap of MLS-AS and MLS-BM shown in Figure 2a–2b (former Figure 3). After adjustment, the enrichment for the overlap between MLS-PSI association and BM-PSI association in the brain exhibited the weakest enrichment among all tissues (adjusted $p = 2.87 \times 10^{-5}$). While the brain also shows some overlap, the degree is substantially lower than in other tissues (adjusted $p < 10^{-24}$). We have updated the manuscript accordingly (page 10-11). “The enrichment for shared splicing events related to both MLS and BM is much weaker in the brain (adjusted $p = 2.87 \times 10^{-5}$) than in any other tissue (adjusted $p < 10^{-24}$ for each of the other tissues).”

7. In Figure 4 it would be useful to see the structure of the gene and highlight the isoforms involved in the AS event. Also indicating the FDR of both correlations would be helpful for the reader.

Reply:

Thank you for the helpful suggestion. We have updated Figure 3 (former Figure 4) to include gene structure diagrams, highlighting the isoforms involved in the alternative splicing events. Additionally, we now display both the p-values and FDRs for the correlations, providing better clarity and context for the reader.

8. Is the overlap between age and MLS-associated AS events higher than expected?

Reply:

Thank you for the insightful question. To assess whether the overlap between age- and MLS-associated AS events is higher than expected by chance, we performed a permutation-based sampling analysis. Using the set of 1,976 homologous candidate AS events (for MLS study) and

3,233 human–mouse homologous AS events (for age study) as the background, we randomly sampled 731 events for the MLS study and 1,756 events for the age study (matching the observed set sizes) across 10,000 iterations. In each iteration, we recorded the number of overlapping events. Interestingly, in all 10,000 permutations, the overlap was equal to or greater than the observed overlap, or 131 in the actual data (empirical p-value < 1e-4), indicating that the observed overlap is significantly lower than expected by chance. This suggests that age- and MLS-associated AS events may represent largely distinct splicing programs. We have revised the manuscript accordingly (page 17).

“Interestingly, several genes with MLS-associated AS events, such as *Hnrnpa2b1*, *Hnrnpk*, *Tra2b*, *FN1*, and *PTBP2*, have been reported to exhibit splicing changes related to aging or lifespan [16, 66], raising the possibility of shared AS regulation between MLS and aging. To rigorously test this potential overlap, we compared MLS-associated AS events with age-AS events. Due to substantial differences in the background sets for these events, we employed a permutation test (10,000 samplings) to calculate an empirical p-value for the observed 131 overlapping events. Surprisingly, we found the 131 overlap was significantly less than expected by chance ($p < 0.0001$), indicating depletion rather than enrichment. We also cross-referenced our MLS-AS genes with GenAge [67], a curated database of aging-related genes, but found no significant enrichment, likely because splicing regulation reveals new candidates missed by gene-level analyses”.

9. Figure 10 shows only the 20 overall enriched RBPs. A supplementary table with all the enrichments, overall, and tissue-specific, should be included.

Reply:

Thank you for pointing this out. As suggested, we have now included a multi-tabbed table (Supplementary Table 2) listing all RBP enrichment results.

10. It may be better not to use abbreviations without introduction in the abstract (RBP).

Reply:

We thank the reviewer for pointing this out. We have revised the abstract accordingly.

Reviewer 4

1. Introduction – “The roles of pre-mRNA alternative splicing (AS) in MLS have yet to be explored.” This claim is not exactly true (e.g., see ref. 15 for a review and PMID 27363602).

Reply:

We thank the reviewer for pointing out this oversight and for suggesting relevant references. We have revised the introduction to include PMID 27363602, which examined AS in relation to lifespan in mouse strains (ranging from 2.1 to 3.3 years, or about 1.5-fold differences) and humans but not any other species. They reported 8 splicing factors whose expressions are associated with lifespan in mouse spleen and to a lesser extent in muscle tissues. Reference 15 reviews RBP dysregulation and splicing errors (intron retention) during the aging process and highlights specific examples of dysregulated splicing events, some of which we reference throughout the main text in supporting or discussing the functional enrichment of alternative splicing events we have identified. While previous publications have provided valuable insights, their studies were often limited to one or two model organisms and focused on the chronological aging process instead of **maximum lifespan as a species trait**.

To our knowledge, our study is the first to systematically investigate MLS-associated AS events across multiple mammalian species spanning a broad range of maximum lifespans (from 2.2 to 37 years, or >16-fold differences), filling a significant gap to uncover potential splicing-mediated regulation that may contribute to lifespan evolution.

2. What is the range of ages for the nonhuman and human GTEx samples analyzed in the study? How the age affects identification of longevity and age-associated AS events? e.g., are there aging samples in the non-human data that can complicate the identification of MLS-associated events? are there young individual samples in the GTEx data so that some of the age-associated events reflect development instead of aging?

Reply:

We appreciate the reviewer’s thoughtful question. The GTEx dataset includes adult individuals ranging from 20 to 79 years old. Since all samples come from post-developmental stages, the identified age-associated AS events are more likely to reflect the aging process rather than early developmental changes. Age information is unavailable for samples of non-human species in the original publication, because the samples were collected from randomly wild-caught animals. Below is from the *Cell Metabolism* publication (PMID: 35580607) that generated the RNA-seq datasets:

“Since most of the species used in this study were caught in the wild, it is difficult to evaluate the precise age of these individuals. We presume that the samples used in this study were from young adults. We excluded juveniles based on size and appearance. Most individuals in wild rodent populations die before they reach an advanced age due to predation and pathogens. Consistently, age structure analysis of wild rodents and shrews revealed that <5% of individuals survive for more than 1 year (Bishop and Hartley, 1976; Blair, 1953). Therefore, we believe that the samples used in this study were from young adult individuals at a comparable physiological stage across the species.”

Our analysis of maximum lifespan (MLS) focuses on differences between species (interspecies analysis), treating MLS as a fixed trait, rather than examining age-related changes within a single species (intraspecies analysis). Given that interspecies differences tend to be much greater than differences within a species, the impact of individual sample age on MLS-associated findings is

relatively smaller. Furthermore, each species in the RNA-seq dataset is represented by 5-6 biological replicates, and the samples were prepared by experienced MLS researchers, further reducing the risk of sampling bias toward older individuals. Therefore, although exact ages for non-human samples are unavailable, this does not substantially affect our ability to detect MLS-associated AS events. The diverse sample set is not skewed toward aged animals, helping to minimize potential confounding from age-related changes. The selection of samples follows established best practices for cross-species studies of maximum lifespan that compare multiple species and treat MLS as an inherent species characteristic.

3. Quantification of AS events. SUPPA was employed to quantify the splicing level of each AS event in terms of percent spliced-in (PSI), utilizing transcript-level expression data. It will be helpful to provide additional details, how PSIs were calculated (e.g., whether it is based on junction reads only or both junction and exonic reads; how transcript-level expression data was utilized and whether this is affected by the accuracy of transcript assembly).

Reply:

We thank the reviewer for the opportunity to clarify this point. SUPPA estimates PSI (Percent Spliced-In) using transcript-level expression values rather than relying solely on either junction or exonic reads. Specifically, PSI is calculated as the ratio of expression of transcript isoforms that include the alternative splicing region relative to the total expression of all transcript isoforms for that gene.

As the reviewer noted, the accuracy of PSI estimates is dependent on the quality of transcript annotation and assembly. To mitigate this, we used high-quality, well-annotated reference transcriptomes (GENCODE v26 for human and vM30 for mouse) and performed *de novo* transcriptome assembly for other species using StringTie. Additionally, we implemented stringent filtering steps to include only AS events with sufficient expression levels and coverage. We considered only AS events whose genes were expressed with TPM (transcripts per million) >0.5 in at least 1/3 of tissue samples. Only AS events with homologs in at least ten non-human mammal species and exhibiting a PSI range ($|\max \text{PSI} - \min \text{PSI}| \geq 20$) were considered. These steps help ensure the reliability of the PSI estimates across species. These details have been clarified in the revised manuscript (page 4-5).

4. The authors built a database of mouse alternative exon/intron sequences against sequences of the other mammalian species. What is exactly the sequence of the other species (i.e., pre-mRNA of the whole gene?). Very few alternative first/last exons were found to be conserved across species. Is this because of technical issues (e.g., longer sequences were used for BLAST search)? The authors have used an 80% blast match to identify conserved alternatively spliced exons. This high stringency yields deeply conserved exons. Nevertheless, exons with vast wobble nucleotide divergence, but conserved amino acid sequences, or UTRs, may be filtered out. Figure 2b for example depicts filtering out of virtually all ALE and AFE, most likely since these exons contain UTRs which have limited sequence constraints. It will be useful to understand the amino acid conservation of exons that have been included in the analyses, as well as those that have been excluded. In addition, it will be worthwhile to repeat this correlation analyses using exons selected based on 80% (or more suitable) amino acid conservation. This will perhaps expand the set of exons considered for MLS-association and could improve the overall significance of the findings as well.

Reply:

Thank you for the thoughtful suggestion. We have revised the manuscript to provide more clarity on the identification of homologous AS events across species.

To begin, we used SUPPA to identify alternative splicing events based on assembled transcripts within each species. For cross-species comparison, we performed BLAST alignments of alternatively spliced regions (not the whole gene) against corresponding sequences in the mouse genome. We include Supplementary Fig. 2 to illustrate this. Specifically, we primarily used alternative exonic sequences for most event types, except for retained introns (Supplementary Fig. 2e), where we used retained intronic sequences.

For alternative first exon (AFE) and alternative last exon (ALE) events (Supplementary Fig. 2f & g), we required that both exons involved in the event (e.g., the first and second exons for AFE) in species 1 have homologous exons in species 2. This dual requirement reduces the likelihood that both exons are artifacts due to incomplete assembly, thus minimizing the occurrence of false positives. The AFE and ALE events identified in species other than mice means that they already meet the 80% coverage threshold and are conserved by a traditional definition. By applying this first conservation criterion, the total numbers of AFE and ALE events are comparable to those of other AS types, with AFE being the second most prevalent AS type. This high prevalence, despite the dual requirement selection, means that the 80% threshold alone is not a constraint for the identification of conserved AFE and ALE events (additional evidence below).

These AFE and ALE events are then subject to a second criterion, i.e., occurring in 10 species or more to qualify for further analyses (or be classified as “conserved across species” in this study). It is this second criterion (≥ 10 species) that filters out most AFE and ALE, likely because of the inherently higher variability of UTR regions across species. This high variability complicates the establishment of confident homology for these regions across species. We could relax the threshold. However, fewer than 10 data points per event would significantly reduce statistical power. Therefore, we have chosen to prioritize specificity and reliability in our cross-species comparisons, even if this means erring on the side of caution.

We agreed with the reviewer’s suggestion to test the effects of relaxing our homology threshold. When we lowered the threshold from 80% to 70%, the total number of conserved events increased from 1,974 to 2,049, an increase of only 3.8%. Dropping the threshold further to 60% led to 2,102 conserved events, a modest additional increase of 2.5%. These relatively small gains support the earlier observation that the 80% threshold does not significantly limit the number of identified exons. Instead, the primary filter is the requirement that conserved events be present in at least 10 species.

At the end, we chose to keep the 80% threshold, for several reasons. Relaxing this criterion brought only modest gains while introducing drawbacks. Lower thresholds allow for more sequence variation, which complicates downstream analyses, especially for RNA-binding protein (RBP) regulation, since RBP motifs are short and sensitive to sequence changes. Additionally, a lower threshold increases the likelihood that the observed associations with maximum lifespan could result from AA changes, rather than differences in splicing (PSI) across species.

We agree with the reviewer that the exclusion of AFE and ALE events is largely due to the high variability of 3' UTR regions. Accordingly, we have reduced emphasis on exon skipping as the most conserved event type and have removed this claim from the abstract.

5. The authors have utilized the 26 species data from Lu et al (2022, Cell Metabolism), a rich resource for studying the connection between AS and MLS. They find a set of AS events that show high correlation between MLS and AS. However, like Lu et al, they find that there is

substantial overlap with BM-MLS targets. In Lu et al, the authors were able to apply a BM-correction to their analyses to remove BM-association as a confounder. As there is a clear overlap of genes showing association between AS-MLS and BM, a similar analysis should be conducted to identify BM-corrected MLS-AS genes. Similarly, the authors should consider if there are overlaps between gene influencing MLS and AS genes influencing MLS and seek to remove gene expression as a confounder in their analyses.

Reply:

Thank you for these thoughtful and important suggestions. We first examined the relationship between BM and MLS and found that they are only moderately correlated (Spearman's $\rho = 0.44$), suggesting that while related, they are not redundant and can be considered separately in modeling. In response to the reviewer's recommendation, we implemented a multiple linear regression model of the form: $MLS = \beta_0 + \beta_1 \cdot BM + \beta_2 \cdot PSI$ for each of our identified MLS-AS events. We then categorized the results based on the significance of each predictor (threshold: $p < 0.05$). These new results are incorporated into the main text (page 11).

“Although BM and MLS are moderately correlated (Spearman correlation = 0.44), they represent distinct biological traits and are not redundant. To disentangle their individual effects, we applied the model $MLS = \beta_0 + \beta_1 \cdot BM + \beta_2 \cdot PSI$ for each identified MLS-AS event. Our results showed that PSI contributes independently to MLS in 59% of cases (where p value for $\beta_2 < 0.05$ and p value for $\beta_1 > 0.05$), while both PSI and BM jointly contribute to MLS in 35% of cases (p values for both β_2 and $\beta_1 < 0.05$). Altogether, in 94% of MLS-associated AS events, PSI was a significant predictor even after accounting for BM, supporting the idea that splicing contributes independently to lifespan regulation.”

We also follow reviewer's recommendation to test if there are overlaps between gene influencing MLS and AS genes influencing MLS. “among the 576 genes with MLS-associated AS events (MLS-AS) and the 6,952 genes with MLS-associated expression changes (MLS-gene), only 211 genes overlapped. Using a background of 17,511 multi-exon genes, this overlap yielded a p -value of 0.94 (Fisher's Exact test), indicating it is not statistically significant.” (page 16)

We also compared functional enrichment between MLS-associated AS events and previously reported MLS-associated gene expression changes from Lu et al. They show clear distinction (see response to point #10, and results on page 14-17).

To further test if gene expression could be a confounder, we compared expression patterns of genes containing positively-associated or negatively-associated alternative splicing events or no associated AS events across six tissues. The results show MLS-AS is largely independent of gene expression. The following is incorporated into the main text (page 12-13).

“The data does not support a universal trend of higher expression in neg-MLS genes or lower expression in pos-MLS genes (Supplementary Fig. 5). The overall gene expression distributions between these two groups were very similar in each tissue. In some tissues, Mann-Whitney U-tests comparing group means revealed slight differences, but the direction of these differences varied. Neg-MLS genes exhibited higher mean expression than pos-MLS genes in the brain and skin ($p < 0.001$) but lower mean expression ($p < 0.05$) in the heart, kidney, and lung. In the liver, there was no significant difference ($p = 0.53$). Further analysis using QQ plots demonstrated these mixed trends among tissues. The heart, kidney, and brain showed a higher proportion of highly expressed genes among neg-MLS events (as illustrated in the QQ plots in Supplementary Fig. 5), while lung, skin, and liver showed the opposite trend for highly expressed genes. Overall, no consistent pattern emerged regarding the gene expression levels of MLS-associated splicing events. While some neg-MLS AS events tend to have higher gene expression, particularly in the brain, this is not consistent across tissues or a widespread pattern. Thus, the

relationships between alternative splicing and MLS regulation are largely independent of gene expression levels and the identified MLS-AS events are not random noise.”

6. Also, several previous studies looked for longevity and/or age-related alternative splicing, e.g. PMID: 27363602, 23340839. Are AS-MLS genes previously curated in other collections of aging genes?

Reply:

We thank the reviewer for highlighting the importance of comparing our MLS-AS genes with previously curated aging-related splicing events. We have included these references. However, neither study has published its complete curated list, so we are not able to make a direct comparison. PMID 27363602 reported splicing changes of 8 genes in the spleen and 5 genes in the muscle. Among these, Fn1 and Vcan are on our MLS-AS list. Mazin et al (PMID: 23340839) studied aging-associated patterns rather than lifespan. They reported 23 genes showing splicing changes. Among these, 9 genes are in our MLS-AS list.

Additionally, we cross-referenced our MLS-AS genes with the aging gene database genAge. However, we did not observe significant enrichment, which is expected since genAge is focused on aging-related genes rather than splicing events. This lack of enrichment further highlights the unique contribution of our study in identifying splicing-specific signals associated with longevity. We have included this result in the revised manuscript (page 19).

7. It will be useful to know if there are multiple AS-MLS exons per gene, and if these exist, whether they have consistent association with MLS or discordant...

Reply:

Thank you for the suggestion. We examined genes that harbor multiple MLS-AS events to determine whether these events exhibit consistent or discordant associations with MLS. We found that some genes do indeed harbor multiple MLS-AS events, with the following counts across tissues:

- Heart: 34 genes (3 discordant)
- Kidney: 30 genes (6 discordant)
- Lung: 24 genes (1 discordant)
- Brain: 23 genes (2 discordant)
- Skin: 22 genes (5 discordant)
- Liver: 21 genes (2 discordant)

The majority of genes exhibit consistent associations with MLS, while a smaller proportion show discordant associations (mean = 11%, median = 9%). We have updated the manuscript to include these results (page.10).

8. Given the critical importance of splicing quantification, it will be very helpful to include additional computational and/or experimental validation to confirm the quantifications are reliable, and not complicated by technical issues such as RNA quality used for library preparation and mapping errors. For example, is it possible to validate using RT-PCR for the two examples provided in Fig. 4 (even with human/mouse RNA if the materials are difficult to obtain for the other species)? It will be helpful to depict these examples in the form of syntenic coverage plots as well.

Reply:

We appreciate the reviewer’s suggestion and recognize the importance of experimental validation. We have been studying the biological functions and regulation of alternative splicing for a long time and always perform experimental validation whenever possible. Unfortunately, the RNA

samples from the wild-caught animals are not available to us, and we do not have the resources, capability, or institutional approval to capture these animals in the field. Figure 3 (former Figure 4) presents data from 26 non-human mammalian species and does not contain human datapoints. Without comparisons with other species, RT-PCR on single mouse species would provide limited values. Nevertheless, to test the concerns about RNA quality and library preparations, we compared the PSI values derived from Lu et al's mouse RNA-seq and our in-house mouse RNA-seq datasets (using high -quality RNAs) and found high consistency, showing that technical factors do not compromise the quantification. The reliability of splicing quantification is also indirectly supported by results that align well with findings in the literature, since these downstream analyses depend on accurate PSI values.

Importantly, the cross-species datasets include 5-6 biological replicates per species, significantly increasing the overall rigor and reliability of splicing quantification. We also applied strict filtering criteria to select only alternative splicing events with adequate expression and read coverage, ensuring the robustness of PSI estimates across all species. Furthermore, the SUPPA tool used for splicing quantification has been validated in its original publication and in subsequent published research.

Synteny coverage plot is a great tool to visualize genome similarity, including gene order and the conservation of gene content. The splice site information, which is most relevant to alternative splicing, does not necessarily stand out. Synteny coverage is asymmetrical between reference and query genomes and can be complicated in multi-species comparisons. Because alternative splicing patterns are hard to show with synteny coverage plots, we updated Figure 3 (former Figure 4) with plots that depict the transcript isoform structures across multiple species, illustrating the presence or absence of the alternative exon in transcript isoforms and highlighting conserved splicing patterns across mammals.

9. Figure 6, the identification of AS events showing the opposite effect on MLS in brain vs other tissues is intriguing, but it is difficult even to imagine how this could happen. Can the authors speculate? The significance should be explored in detail.

Reply:

We thank the reviewer for their interest in this intriguing brain-specific pattern. The observation that certain AS events show opposing correlations with MLS in brain versus other tissues may reflect the unique functional and regulatory landscape of the brain. We added the following to the text (page 13-14):

“This divergence could be an example of antagonistic pleiotropy, where a biological factor has opposite effects on fitness in different tissues. That is, an AS event promotes the brain's longevity-related functions but incurs a trade-off in other tissues, resulting in opposite effects on MLS correlation. Another possibility is intrinsic negative correlations between the brain and peripheral tissues driven by certain splicing regulators that are active only in the brain. Remarkably, such a contrasting correlation pattern was rare in other tissues, with just one example each in the heart and liver.”

We have also included a dedicated section in Discussion, titled “**Brain-specific splicing associated with MLS**”, where we explore the topic in additional detail. (page 24-25)

10. Lu et al note that the key pathways enriched in MLS-associated genes are energy metabolism, inflammation, circadian genes, DNA damage regulation, immune responses, and RNA biology. A comparison should be performed to understand the overlap in enriched between AS targets and Lu et al gene targets. It is quite interesting that AS genes enrich distinct sets of

pathways, can the authors detail whether certain pathways might contribute to MLS via AS, whereas others through expression changes?

Reply:

We appreciate the reviewer's suggestion to compare pathway enrichments between MLS-associated AS events and expression-based gene targets identified in Lu et al. To explore this, we compared the enriched biological processes from our MLS-AS analysis with those reported for MLS-associated gene expression. In response to Reviewer 2's recommendation to use a more updated tool for GSEA analysis, we have replaced the original GSEA analysis with Enrichr (PMID: 23586463), which performs overrepresentation analysis and reports Benjamini–Hochberg adjusted p-values by default. The adjusted p-values are provided in Figure 6 and Supplementary Table 3.

The overlap between the two sets of enriched terms was minimal. MLS-AS genes are overwhelmingly enriched in pathways related to multiple steps of mRNA processing, stress response, neuronal functions, and epigenetic regulation. It is worth noting that in Lu et al., only RNA transport showed enrichment among their gene set, while other RNA processing pathways were not reported as enriched. Therefore, certain pathways contribute to MLS predominantly through splicing regulation, while others are more affected by gene expression modulation. These findings are updated in the Result section "**Functional enrichment of MLS-associated alternative splicing**" on page 14-17.

"To understand the biological significance of MLS-associated AS (MLS-AS) events, we conducted functional enrichment analysis using Enrichr. We examined genes harboring MLS-associated splicing events for enrichment in the Gene Ontology (GO) biological process terms and KEGG pathways. MLS-AS genes are significantly enriched in functional categories of mRNA processing, stress response, neuronal functions, and epigenetic regulation (Supplementary Table 3). Figure 6 summarizes the significantly enriched terms (FDR < 0.15), organized by functional categories.

Multiple enriched pathways involve cellular stress response, including Stress Granule Assembly, Regulation of Defense Response to Virus, MAPK Cascade, *Positive Regulation of PI3K-AKT Signal Transduction*, etc. stress response plays an essential role in safeguarding cellular and organismal integrity^{44–46}. When cells encounter stressors such as oxidative damage, heat shock, viral infection, or metabolic fluctuations, they initiate intricate signaling pathways to mitigate harm and restore homeostasis. Efficient stress response mechanisms enable cells to quickly detect and neutralize damage, activate repair systems, and, if necessary, trigger programmed cell death (apoptosis) to remove irreparably compromised cells. Indeed, Regulation of Apoptotic Process (and related GO terms) is significantly enriched (Figure 6). Apoptosis ensures that tissues maintain their functional capacity and that damaged cells do not accumulate^{47,48}. Moreover, the adaptability of the stress response is crucial for long-lived species. Organisms with robust stress response networks can better withstand environmental and physiological fluctuations across their lifespan, delaying the onset of cellular dysfunction and chronic pathologies. Alternative splicing of stress response genes likely modulates organisms' competence for the efficiency and adaptability of stress response.

Notably, mRNA processing steps and regulatory mechanisms are enriched, including spliceosomal complex assembly (along with many other splicing-related terms), cleavage and polyadenylation, regulation of translation, mRNA surveillance, and RNA transport. Collectively, these comprise 18% of all enriched GO terms and encompass nearly every stage of the mRNA life cycle. This underscores the importance of maintaining transcriptome integrity and proper mRNA processing for longevity. Errors or inefficiencies in any of these processes can result in the production of faulty or non-functional proteins, accumulation of aberrant RNA species, and

ultimately, cellular dysfunction. For long-lived species, the ability to maintain precise and adaptable RNA processing systems is particularly critical. Accurate splicing ensures that essential gene products are expressed in correct proportions and forms, while alternative splicing provides an additional layer of regulatory flexibility, allowing organisms to rapidly adapt to changing physiological conditions without altering their genomic DNA. mRNA surveillance mechanisms, such as nonsense-mediated decay, also respond to cellular stress and safeguard the transcriptome by identifying and eliminating defective mRNAs that could otherwise give rise to toxic or deleterious proteins⁴⁹. Thus, robust RNA processing and regulation are central to maintaining cellular homeostasis and adaptability, both of which are essential for healthy aging. Their enrichment suggests that longevity in some species may be achieved, at least in part, by optimizing alternative splicing mechanisms that control the transcriptome's quality, diversity, and responsiveness throughout the lifespan.

Additionally, we observed a significant enrichment of biological processes related to neuronal functions, such as Regulation of Neuronal Synaptic Plasticity, Axon Guidance, Synapse Organization, Brain Development, Neuron Migration, Long-term potentiation, Neurotrophin Signaling Pathway, Neurotransmitter Secretion, etc. Neuro-related terms represent over 22% of all enriched GO terms, further reinforcing the distinctive role of brain-specific^{50–59}. The pivotal role of neuronal function in determining organismal lifespan likely arises from the brain's central control over homeostasis and its orchestration of numerous physiological processes intimately connected to aging and longevity. Learning, memory, and adaptability of neural circuits are intricately regulated by pathways involved in axon guidance, synaptic organization, and neurotrophin signaling. Through these mechanisms, neurons can adapt to developmental cues and environmental stressors. Importantly, the brain influences systemic aging via neuroendocrine and autonomic pathways. For example, the hypothalamus integrates cues related to energy balance, stress, and circadian rhythms, coordinating hormonal outputs that impact metabolism, immune responses, and tissue regeneration. In this way, neuronal function serves as a central determinant of lifespan—supporting cognitive maintenance, regulating systemic physiology, modulating immune activity, and conferring resilience to stress. In summary, these findings highlight the diverse biological processes modulated by alternative splicing to influence maximum lifespan.

To compare the respective contributions of gene-level and splicing-level regulation to lifespan, we compared the functional enrichment of genes with MLS-associated AS events and those with MLS-associated expression changes (as identified by Lu et al.). First, among the 576 genes with MLS-associated AS events (MLS-AS) and the 6,952 genes with MLS-associated expression changes (MLS-gene), only 211 genes overlapped. Using a background of 17,511 multi-exon genes, this overlap yielded a p-value of 0.94 (Fisher's Exact test), indicating it is not statistically significant.

Comparing their pathway enrichments reveals clear distinction. MLS-genes are enriched in GO group terms such as DNA replication, mitotic recombination, chromosome segregation, double-strand break repair, protein folding, mitochondrion organization, cilium movement, and lymphocyte chemotaxis. They are also particularly enriched in various macromolecular metabolism pathways, including acylglycerol metabolic process, amino acid catabolic process, carbohydrate biosynthetic process, ribonucleoside biphosphate biosynthetic process, hydrogen peroxide catabolic process. These findings align with current theories that emphasize DNA repair⁶⁰ and metabolic control in longevity. Proper regulation of macromolecule metabolism is particularly crucial for maintaining cell structure and function, and repairing damaged tissues^{61,62}. These GO terms or functional groups are not prominent in MLS-AS genes. Overall, only seven out of 265 GO terms from MLS-AS events are enriched in MLS-gene groups (FDR <0.15 for both groups). When the threshold is tightened to FDR < 0.1, this overlap drops to just one exact shared term, and at FDR < 0.05, there are no overlapping GO terms between the two groups. We therefore used FDR <0.15 for comparisons. Since different GO terms can still be related under one functional activity, we used REVIGO to organize the GO terms into functional categories. The

results still highlighted a clear distinction. For example, among the 424 total GO terms enriched in MLS-genes, only three were related to neural function (0.7%) and ten to RNA-related processes (2.3%). Similarly for MLS-AS genes, DNA Repair is only minorly represented with two specific terms, “Regulation of Nucleotide-Excision Repair” (FDR: 0.056) and “Positive Regulation of DNA Repair” (FDR: 0.082). This limited intersection underscores that alternative splicing modulation acts as an independent regulatory layer to gene expression regulation.”

11. Figure 9 shows that human-age related AS events involve IDPs. Further information should be provided about the nature of these AS events. Are the alternative exons coding for the IDP regions? What pathways are these genes involved in?

Reply:

We appreciate the reviewer’s suggestion to further characterize the nature of the IDP-related AS events. To investigate whether these splicing events preferentially encode intrinsically disordered protein (IDP) regions, we examined their overlap with both curated and predicted IDP sequences from the MOBI database. We also explored what pathways they are involved in. The results are now reported in the revised manuscript (page 20-21).

“These gene-level results prompted us to investigate whether MLS-associated or Age-associated AS events (or, for simplicity, longevity-associated AS events) preferentially occurred within IDP regions. Using the MOBI database of both curated and predicted IDP sequences, we assessed their overlap with AS regions and found that in both human and mouse, the longevity-associated AS events showed significantly greater overlap with IDP regions than did background AS events (e.g., human curated: $p = 3.16 \times 10^{-4}$, predicted: $p < 10^{-9}$; mouse curated: $p = 0.0023$, predicted: $p < 10^{-9}$; one-sided Fisher’s exact test). These findings suggest that AS may contribute to lifespan regulation, at least in part, by encoding protein disorder domains, potentially impacting molecular interactions and conferring functional adaptability, which could be essential to stress response and signaling during aging.

We next explored which biological pathways were enriched among these IDP-overlapping genes using Enrichr. In both age and MLS-associated AS gene sets, we observed significant enrichment in RNA processing-related pathways, such as Regulation of mRNA Splicing via the Spliceosome, Regulation of Alternative mRNA Splicing, and mRNA Processing. This suggests a possible feedback mechanism in which splicing factors themselves may be regulated via AS in IDP regions.

Age-associated AS events with IDPs were also enriched in transcriptional regulation pathways, such as Regulation of DNA-templated Transcription and Regulation of Gene Expression. Interestingly, curated age related AS events with IDPs were enriched in cardiac-specific developmental processes, such as Regulation of Cardiac Muscle Cell Proliferation and Positive Regulation of Cardiac Muscle Tissue Growth, hinting at potential tissue-specific splicing regulation during aging. In contrast, MLS-associated AS events with IDPs showed enrichment in stress-response pathways, including Protein Localization to Cytoplasmic Stress Granule, Endoplasmic Reticulum Stress-induced Apoptotic Signaling. These pathways are well-recognized components of longevity regulation, suggesting that AS events occurring in IDP regions may contribute to lifespan determination by enhancing an organism’s ability to respond to stress and adapt”

12. The RBP analyses seem interesting, but quite premature as it stands. It seems that only the alternative exons were used for motif analysis. It will be helpful to also examine flanking intronic sequences contributing to AS (e.g., upstream and downstream flanking intronic sequences for skipped exons). How the enrichment of the motif patterns is consistent with the direction of MLS or age-associated splicing and RBP expression in different samples to get a sense whether any of the candidate RBPs is positively or negatively associated with the

phenotype? i.e. Is there a relationship between RBP motif and AS-target PSI across the 26 species? That seems a critical aspect to understand RBP regulation. Even in the absence of high-quality genomes, this analysis can be performed for the exonic sequences at least. Furthermore, the authors suggest that RBPs identified influence lifespan: are these genes identified among those in Lu et al?

Reply:

We thank the reviewer for raising these important points. In our initial RBP motif enrichment analysis, we focused only on the alternatively spliced exon and retained intron sequences. In response to the suggestion, we have expanded the analysis to include flanking intronic and exonic regions—specifically ± 100 nucleotides into introns and ± 50 nucleotides into exons from the splice sites (page 7). This more comprehensive approach increased our sensitivity and uncovered additional enriched RBP motifs potentially involved in regulating these events.

To investigate whether RNA-binding protein (RBP) motifs are associated with alternative splicing in the context of lifespan, we examined the correlation between motif occurrence frequency and the percent spliced in (PSI) values of MLS-associated AS events across 26 species. We have updated the manuscript on page 23 and listed the identified RBPs in Supplementary Table 4.

“Examining motif frequencies across the 26 species revealed that certain RBP’s motif occurrence correlated strongly with splicing inclusion levels (PSI values) for specific MLS-associated AS events, supporting their regulatory interactions (Supplementary Table 4). The number of RBPs showing significant PSI correlations differed by tissues, with liver showing the highest number (17), followed by lung (16) and heart (15), while skin and brain exhibited fewer associations (3 and 5, respectively). Some RBPs, such as ELAVL4, YBX1, TIA1, QKI, and HNRNPD, were found across multiple tissues. These findings suggest that specific RBPs consistently interact with local motifs of MLS-associated AS events across species, highlighting their potential roles in regulating lifespan-associated splicing.”

To assess whether RBP expression is functionally associated with MLS, we cross-referenced our RBP list with the set of genes previously identified as significantly correlated with MLS by Lu et al., which uses the same dataset. These results are included in the revised manuscript (page 22).

“To investigate whether expression levels of these RBPs are linked to lifespan, we assessed the correlation between their expression and MLS across six tissues based on the prior study⁹. Of the 39 unique key RBPs (the union of the top 20 motif-enriched RBPs from each tissue), only five (*ESRP1*, *FUS*, *MBNL1*, *PCBP2*, *RBM8A*) overlapped with a previously curated list of genes whose expression correlates with MLS⁹. This limited overlap suggests that analyzing splicing regulation can identify a distinct set of potential lifespan regulators missed by studies focusing on gene expression alone. This reinforces the biological relevance of RBP-mediated splicing regulation in mammalian longevity.”

13. Elastic net model – regularization terms are not specified in the description.

Reply:

Thank you for pointing it out. We have now specified the regularization terms used in our elastic net model: “incorporating both L1 (Lasso) and L2 (Ridge) regularization that simultaneously considers all splicing events and existing covariates to remove potential confounding factors” (page 5-6).

14. The manuscript has 11 figures – some (e.g., figures 1 and 2) might be moved supplemental materials.

Reply:

Thank you for the suggestion. We have moved Figure 1 to the supplementary materials as recommended. However, we have retained Figure 2 in the main text, as it now includes additional subfigures based on suggestions from other reviewers, which enhance the clarity and relevance of the presented results.

15. Fig. 7 – explain gene ratio.

Reply:

Thank you for pointing this out. In the revised version of the manuscript, we have replaced the previous GSEA-based enrichment analysis with Enrichr based on suggestions from another reviewer. Unlike GSEA, Enrichr does not report a “gene ratio” (i.e., the number of overlapping genes divided by the size of the input gene set). Instead, it uses an odds ratio to indicate the strength of enrichment and provides adjusted p-values to assess statistical significance. These metrics are now displayed in Figure 6 (former Figure 7), and we have updated the figure legend to clarify this.

16. The mammalian order Eulipotyphla is repeatedly misspelled in the text

Reply:

We thank the reviewer for pointing this out. We have made corrections.

We appreciate Reviewer #2's valuable feedback. We have done additional analyses and revised the manuscript to ensure all concerns are addressed. A detailed, point-by-point response to each comment is provided below.

The authors made a substantial effort in replying to all of our comments. However, I still have a two minor concerns that were not fully addressed in their rebuttal:

I have methodological concerns in using functional enrichment analysis and using genome wide as background. Using genome-wide background likely is in most cases incorrect because the gene set was detected in an analysis where most likely not all genes were tested i.e. in when doing differential alternative splicing in a particular tissue, not all genes are tested, only those expressed in that particular tissue and often there are other filters. So if we want to test whether the differentially alternatively spliced genes are enriched in a function this should be compared to the genes we initially tested as differentially spliced and not all genes. In this paper some of this issues are explained: <https://pmc.ncbi.nlm.nih.gov/articles/PMC8936487/>

This applies to all functional enrichments throughout the paper and the backgrounds should be specified in methods.

Response: We thank the reviewer for pointing this out. In our original analysis, we used all annotated genes (~20k for mouse) as the background. Following the reviewer's suggestion, we have now limited the background to those genes that **BOTH** exhibit AS events and are expressed (TPM > 0.5 in at least one tissue; ~12k genes in mouse). This is a more relevant reference set, as it accounts for expression and restricts the analysis to genes with detectable AS events. Mouse expression was used because the GO analysis for MLS is based on annotated mouse GO terms and because the mouse genome and transcriptomes served as references for defining conserved events.

With the new background gene set, most GO terms and KEGG Pathway terms remain significant. However, we noted that the number of GO terms with FDR <0.15 reduced from 265 to 192, and KEGG terms from 30 to 29. The adjusted p-values and Odds Ratio have been updated in Supplementary Table 3.

The major categories of functional terms remain consistent as shown in the revised Figure 6. However, the order of GO terms by their adjusted p-values have changed. We have updated the main text (page 14-17) accordingly to focus on the new highly significant terms and removed those determined to be insignificant or weakly significant. For example, "MAPK Cascade" now has a higher adjusted p-value in the new analysis so is removed from Figure 6. Most mRNA processing steps remain highly enriched, particularly for regulation of alternative splicing. However, regulation of cleavage and polyadenylation is no longer significant and has been removed from the text (page 15). Similarly, terms like "Brain Development" are no longer significant and are removed.

We have also compared GO enrichment between MLS-AS events and MLS-associated genes identified by Lu et al. (PMID 35580607, Cell Metab. 2022). MLS-genes are enriched in GO group terms such as DNA replication, mitotic recombination, chromosome segregation, double-strand break repair, protein folding, mitochondrion organization, cilium movement, and lymphocyte chemotaxis. This comparison highlights a clear distinction. For example, among the 424 total GO terms enriched in MLS-genes, only three were related to neural function (0.7%) and ten to RNA-related processes (2.3%). On the other hand, MLS-genes are highly enriched for DNA Repair related terms, a pattern that is not observed with MLS-AS events. This limited intersection underscores that alternative splicing modulation acts as an independent regulatory layer to gene expression regulation.

For functional enrichment analysis of the human age-associated-AS events, we have updated the analyses using expressed human genes containing AS events as the background. The results remain consistent with those obtained using the original genome-wide background.

We also update the background gene set to determine the enrichment of MLS-AS and age-AS events in genes containing IDP. The FDR values are $4.2E^{-56}$ for MLS-AS events, $1.2E^{-47}$ for age-AS events, and $1.2E^{-13}$ for overlapping events. They are highly significant.

Regarding the enrichment analysis for IDP-containing genes with MLS-AS and age-AS events, we updated the background gene sets. Most terms remain significant, and we have revised the main text to reflect these new results with terms showing stronger enrichment (page 21).

Having 100 false positive genes in the permutation analysis is a bit concerning to me. The authors claim that their results hold because they identify more genes (400) and those are associated with aging in their functional enrichment analysis. I am aware the authors can not filter by p-value but it would be good if there is anything else the authors could do to reduce false positives?

Response: Thank you for raising this important point. Since elastic net models do not provide p-values, we explored further filtering based on the absolute values of regression coefficients per reviewer #2's request. Because PSI values for AS events range between 0 and 1, the coefficient magnitude provides a natural criterion. In our initial analysis, the median/mean FDR without coefficient filtering was ~0.25 across tissues. Progressively increasing the coefficient cutoff from 0.1 to 0.7 led to a decrease in median FDR from 0.22 to 0.098 and mean FDR from 0.23 to 0.11. This consistent trend suggests that applying stricter coefficient thresholds can effectively reduce the false positive rate while preserving robust signals. We hope this addresses reviewer #2's concern.

Cutoff	Median # of age-AS events	Mean # of age-AS events	Median FDR	Mean FDR
0.1	331	338	22.30%	23.11%
0.3	233	239	17.96%	18.04%
0.5	162	169	14.21%	14.76%
0.7	113	118	9.79%	10.95%

Reviewer #2 (Remarks to the Author):

the authors have addressed my concerns. I only have a minor comment. The authors should show the decrease of the performance in FDR when increasing the threshold in the manuscript rather than just to me.

Response: We appreciate Reviewer #2's feedback and have added the results to the manuscript on page 22 (blue font).